# Dynamical and combinatorial coding by MAPK p38 and NFκB in the inflammatory response of macrophages

Stefanie Luecke[1,2], Xiaolu Guo [1,2,7], Katherine M Sheu [1,2,7], Apeksha Singh [1,2,7], Sarina C Lowe [1,3], Minhao Han [1,2], Jessica Diaz[1,2], Francisco Lopes [2,4], Roy Wollman [2,5,6] & Alexander Hoffmann [1,2 ✉]

## Abstract

**Macrophages sense pathogens and orchestrate specific immune responses. Stimulus specificity is thought to be achieved through combinatorial and dynamical coding by signaling pathways. While NFκB dynamics are known to encode stimulus information, dynamical coding in other signaling pathways and their combinatorial coordination remain unclear. Here, we established live-cell microscopy to investigate how NFκB and p38 dynamics interface in stimulated macrophages. Information theory and machine learning revealed that p38 dynamics distinguish cytokine TNF from pathogen-associated molecular patterns and high doses from low, but contributed little to information-rich NFκB dynamics when both pathways are considered. This suggests that immune response genes benefit from decoding immune signaling dynamics or combinatorics, but not both. We found that the heterogeneity of the two pathways is surprisingly uncorrelated. Mathematical modeling revealed potential sources of uncorrelated heterogeneity in the branched pathway network topology and predicted it to drive gene expression variability. Indeed, genes dependent on both p38 and NFκB showed high scRNAseq variability and bimodality. These results identify combinatorial signaling as a mechanism to restrict NFκB-AND-p38-responsive inflammatory cytokine expression to few cells.**

**Keywords** Combinatorial Signaling Dynamics; Cytokine Control; Inflammation; Innate Immunity; Signal Encoding
**Subject Categories** Immunology; Signal Transduction

## Introduction

Macrophages are ubiquitous sentinel cells of the innate immune system. They maintain tissue homeostasis and orchestrate local and systemic immune responses via the secretion of cytokines and chemokines (Wynn et al, 2013; Sheu and Hoffmann, 2022). They must balance high sensitivity to pathogens to generate effective responses against the risk of tissue damage. Indeed, dysregulated macrophage responses are associated with numerous pathologies (Murray and Wynn, 2011; Luecke et al, 2021). The stimulus-specificity of responses is a means to ensure appropriate but not unnecessary immune activity and is, therefore, a hallmark of healthy macrophages (Sheu and Hoffmann, 2022). Macrophages sense pathogen-associated molecular patterns (PAMPs) and host cytokines through more than a dozen receptors (including toll-like receptors (TLRs)), which converge on a limited number of immune response signaling pathways, including the IKK/NFκB and TBK1/IRF pathways, and the MAPKs, which include p38, JNK, and ERK (Kawai and Akira, 2011; Ablasser and Hur, 2020; Luecke et al, 2021).

How do these pathways effect stimulus-specific gene expression programs? Prior studies have provided evidence for both dynamical and combinatorial coding of stimulus information (Hoffmann, 2016; Sheu et al, 2019; Luecke et al, 2021). Through dynamical coding, a single signaling pathway can convey information about distinct stimuli to the nucleus via stimulus-specific dynamics, i.e., variations over time, of signaling activity. Target genes can then distinguish the different dynamic profiles to produce stimulus-specific responses. For example, the activity dynamics of the transcription factor NFκB were shown to convey information about extracellular immune threats to the nucleus via informative dynamical features, termed "NFκB signaling codons", such as speed, amplitude, and duration of signaling (Hoffmann et al, 2002; Werner et al, 2008; Tay et al, 2010; Turner et al, 2010; Cheong et al, 2011; Lee et al, 2014; Adelaja et al, 2021; Covert et al, 2005; Werner et al, 2005). NFκB target genes and enhancers have been shown to distinguish the differential deployment of these signaling codons, in

---

[1]Department of Microbiology, Immunology, and Molecular Genetics (MIMG), University of California Los Angeles, Los Angeles, CA 90095, USA. [2]Institute for Quantitative and Computational Biosciences, University of California Los Angeles, Los Angeles, CA 90095, USA. [3]Vatche and Tamar Manoukian Division of Digestive Diseases, Department of Medicine, David Geffen School of Medicine, University of California Los Angeles, Los Angeles, CA 90095, USA. [4]Grupo de Biologia do Desenvolvimento e Sistemas Dinamicos, Campus Duque de Caxias Professor Geraldo Cidade, Universidade Federal do Rio de Janeiro, Duque de Caxias 25240-005, Brazil. [5]Department of Chemistry and Biochemistry, University of California Los Angeles, Los Angeles, CA 90095, USA. [6]Department of Integrative Biology and Physiology, University of California Los Angeles, Los Angeles, CA 90095, USA. [7]These authors contributed equally: Xiaolu Guo, Katherine Sheu, Apeksha Singh. ✉E-mail: ahoffmann@ucla.edu

a gene-specific manner (Lee et al, 2014; Lane et al, 2017; Sen et al, 2020; Cheng et al, 2021; Werner et al, 2005). While the information content of NFκB dynamics has been thoroughly elucidated, the information content of dynamical signaling activities of other innate immune response pathways has not been quantified or characterized.

Combinatorial coding involves two or more pathways that are activated in stimulus-specific combinations; this allows immune response genes to be expressed stimulus-specifically by responding to specific pathway combinations, often through highly gene-specific regulatory mechanisms (Hoffmann, 2016; Sheu et al, 2019; Luecke et al, 2021; Tong et al, 2016; Sen et al, 2020). Biochemical studies that average cell variable activities have established that PRRs and cytokine receptors activate different subsets of the key immune response signaling pathways, e.g., NFκB + JNK in response to the host cytokine TNF (tumor necrosis factor), NFκB + JNK+p38 in response to MyD88-activating PAMPs, and NFκB + JNK + IRF in response to TRIF-activating PAMPs (Amit et al, 2009; Cheng et al, 2017; Luecke et al, 2021). In single RAW264.7 cells, a macrophage-like immortal cell line, combinatorial activation of JNK and NFκB activity was shown to allow distinction of bacterially-infected cells and uninfected bystander cells as well as exposure dose (Lane et al, 2019). MAPK p38 was also reported to require higher doses of TLR4-ligand for activation than NFκB, suggesting that it may contribute to ligand dose distinction (Gottschalk et al, 2016; Regot et al, 2014). However, how MAPK p38 is regulated dynamically in primary macrophages, whether the dynamics contain information, and how they are coordinated with NFκB dynamics to encode information about the dose and molecular identity of the stimulus remains unclear.

Prior biochemical studies of population averages indicate that MAPK p38 is a good candidate for both dynamical and combinatorial coding with NFκB. Studies of signaling mechanisms have reported two distinct signaling pathways activating MAPK p38. Whereas p38 activation by TNF relies almost entirely on MKK3/6 downstream of IKK and Tpl2, the PAMP LPS (lipopolysaccharide) can alternatively activate p38 via MKK4 downstream of TAK1 (Pattison et al, 2016), suggesting the possibility that different activation dynamics may result. Further, studies of gene regulatory mechanisms showed that the combination of NFκB and MAPK p38 controls important immune response genes, such as inflammatory cytokines (Cheng et al, 2017). In addition to activating transcription factors such as CREB (Park et al, 2005; Arthur and Ley, 2013), MAPK p38 is an important regulator of post-transcriptional and post-translational regulation of pro-inflammatory cytokines by controlling mRNA processing and half-life, pro-protein processing, and secretion (Caldwell et al, 2014; Mahtani et al, 2001; Luecke et al, 2021; Andersson and Sundler, 2006; Xu and Derynck, 2010; Scott et al, 2011). Through these mechanisms, p38 activity has been described to form a "sequential AND gate" with transcription-activating NFκB (Cheng et al, 2017), meaning that although they act on sequential biochemical steps, both their activities are required for proper production of cytokines regulated in this manner. This suggests a role for NFκB-p38 combinatorial coding to ensure the stimulus-specificity of gene expression.

While there is good evidence that NFκB-p38 combinatorial coding plays a role in the stimulus-specificity of macrophage responses, how it relates to the dynamical coding within either pathway is not known. Quantifying coding capacities in NFκB-p38 combinatorial signaling requires examination at the single-cell level, given the substantial cell-cell-heterogeneity within innate immune responses. Stimulus-specificity may be quantified using machine learning classification or information-theoretic analyses developed in signal theory, which determines how well observed signaling features are correlated with the stimulus (Rhee et al, 2012; Mitchell and Hoffmann, 2018; Shannon, 1948).

To undertake single-cell quantitative studies of dynamical and combinatorial coding through p38 and NFκB in primary macrophages, we leveraged two recently developed technologies: First, we employed kinase translocation reporters (KTRs), which are engineered proteins that contain a fluorescent protein coupled to the substrate recognition motif of the kinase of interest and phosphorylation-dependent nuclear localization and export signals (Kudo et al, 2018; Regot et al, 2014). Phosphorylation of the KTR typically increases nuclear export and decreases nuclear import. This allows monitoring of the dynamic activity, both activation and deactivation, of the kinase in real time using live cell microscopy. Second, we employed the HoxB4-transduced myeloid precursor system (hMPs) (Ruedl et al, 2008) to generate primary hMP-derived macrophages (hMPDMs) using M-CSF. These closely resemble primary bone marrow-derived macrophages (BMDMs) in terms of PAMP-responsive transcriptomic responses and NFκB signaling dynamics (Sheu et al, 2023; Singh et al, 2024). Since the precursors can be maintained in culture, they can be genetically engineered with the aforementioned KTRs before differentiation into macrophages.

We report here that p38 activity shows stimulus- and dose-specific dynamics that contain less information than those of NFκB but allow precise distinction of TNF from PAMPs. However, in combination with the information-rich NFκB dynamic features, p38 dynamics contribute, contrary to expectation, only little to ligand- or dose-distinction, despite the differential dose-response behavior of the two pathways. This suggests that immune response genes gain stimulus-specificity when evolving the ability for decoding NFκB dynamics or decoding NFκB&p38 combinatorics, but gain little further from evolving both. Due to noisy signaling inherent to the branched pathway architecture, p38 and NFκB signaling dynamics were found to be poorly correlated and heterogeneous. Our results suggest that AND gate gene regulatory mechanisms have a role in generating cell-variable, even bimodal expression responses of its target genes.

## Results

### Generation of dual reporter macrophages for live cell imaging of NFκB and p38 MAPK activities

To study the single cell activity dynamics of NFκB and p38 MAPK simultaneously in macrophages using live cell imaging, we generated dual reporter cells expressing fluorescently labeled RELA and a p38-specific fluorescent kinase translocation reporter (KTR) (Fig. 1A). Using bone marrow from a previously established knock-in mouse line, which expresses mVenus-RelA from the endogenous *Rela* locus (Adelaja et al, 2021) (thus avoiding potentially artefactual NFκB dynamics caused by overexpression-based reporter systems (Barken et al, 2005)), we generated hMPs (Ruedl

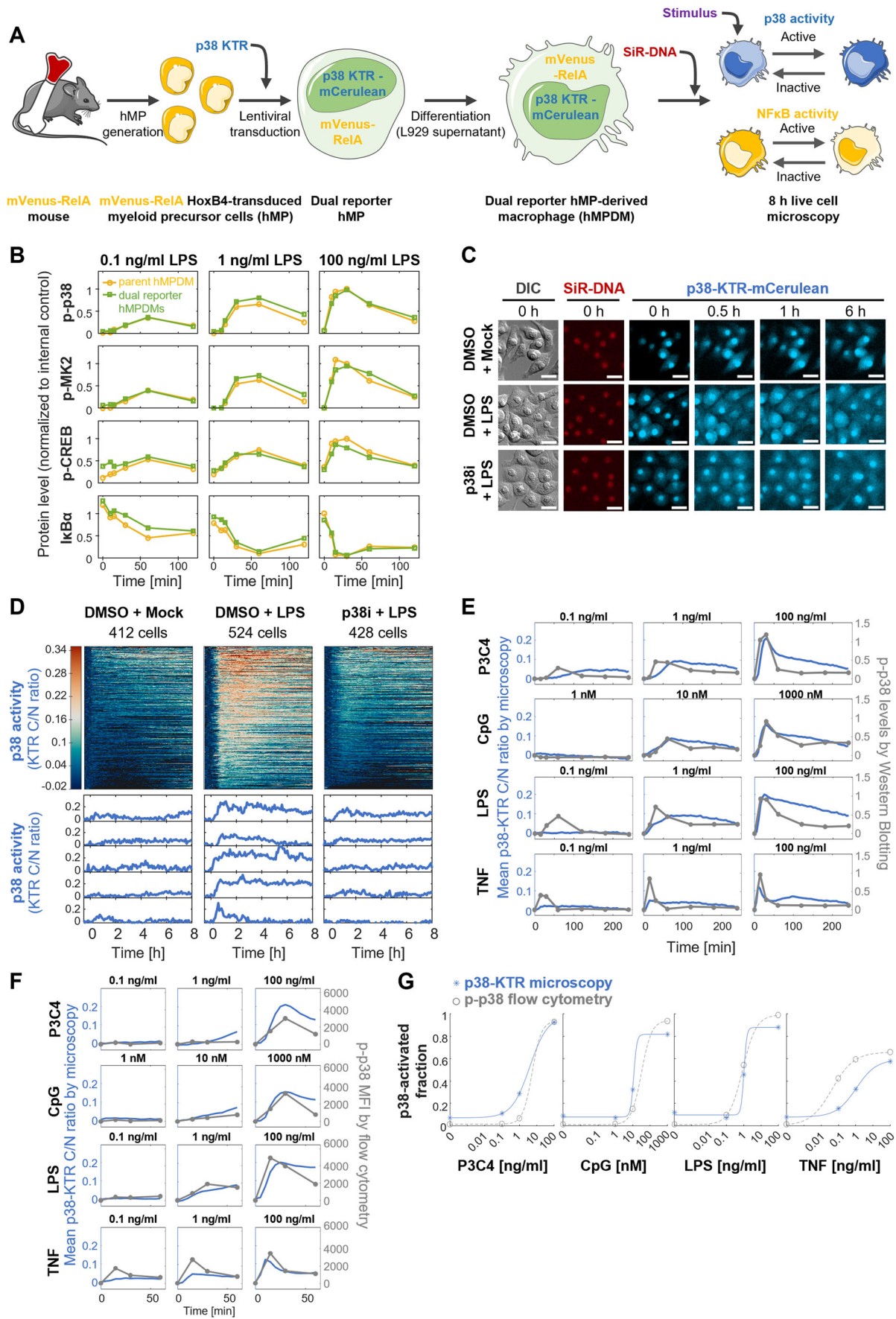

**Figure 1.  Experimental system for quantifying p38 and NFκB activity dynamics in murine macrophages.**

(A) Schematic of approach to generate NFκB Rela and p38 MAPK activity dual fluorescence reporter macrophages by lentiviral transduction of hMPs. (B) Signaling responses to three LPS doses over 2 h in mVenus-Rela p38-KTR-mCerulean hMPDMs compared to parent cell line by Western Blotting for phospho-p38, phospho-MK2, phospho-CREB, and IκBα protein levels. Band intensities were background corrected, normalized to tubulin control, and normalized across multiple membranes using an internal control sample. Data from one experiment are depicted (Western Blot membrane shown in Fig. EV1C). (C) Fluorescence microscopy images of p38-KTR localization in hMPDMs upon stimulation with 100 ng/ml LPS over 6 h with and without p38 inhibitor pre-treatment, as well as nuclear dye (SiR-DNA) fluorescence and DIC image at baseline. Scale bar: 20 μm. (D) p38 activity dynamics in response to 100 ng/ml LPS with and without p38 inhibitor measured over 8 h by fluorescence microscopy, expressed as baseline-corrected p38-KTR fluorescence cytoplasmic/nuclear ratio, quantified by automated image analysis. Each row of the heatmap represents the p38 signaling trajectory of one cell. Trajectories are sorted by maximum amplitude. Example trajectories are shown below. Data from one experiment are depicted. (E) Comparison of p38 activity over 4 h in hMPDMs measured by p38-KTR microscopy (mean of trajectories) and by bulk phospho-p38 levels measured by Western Blotting in response to indicated doses of P3C4, CpG, LPS, or TNF. Western Blotting quantification: band intensities were background corrected, normalized to tubulin control, normalized across multiple membranes using an internal control sample, and baseline-deducted; depicting data from a single experiment (Western Blot membrane shown in Fig. EV1D). For microscopy, the mean of means of trajectories from two biological replicates is shown (Total # of cells: 923, 1171, 970, and 1055 cells for P3C4, CpG, LPS, and TNF). (F) Comparison of p38 activity over 1 h in hMPDMs measured by p38-KTR microscopy (mean of trajectories, as in Panel E) and as intracellular p-p38 levels by flow cytometry in response to indicated doses of P3C4, CpG, LPS, or TNF. MFIs were baseline-deducted. Data from one flow cytometry experiment is shown. (G) Comparison of the fractions of cells with p38 activity in the hMPDM population as measured by p38-KTR microscopy or by intracellular p-p38 flow cytometry upon stimulation with P3C4, CpG, LPS, or TNF. Microscopy: A cell is considered p38 active if its KTR measurement passes a threshold of 3x STDV of baseline for 3 consecutive timepoints within 1 h of stimulation. Data from two pooled biological replicates are used. Flow cytometry: The fluorescence signal of an unstimulated, unstained sample is used to define a cutoff between p-38[+] and p-p38[−] cells (Fig. EV1F). For each dose, the fraction of cells passing the threshold at 30 min post stimulation (Fig. EV1G) is plotted. Data from one experiment are displayed. DIC differential interference contrast, p38i p38 inhibitor, MFI mean fluorescence intensity. Source data are available online for this figure.

et al, 2008). These cells can be differentiated into hMP-derived macrophages (hMPDMs) that very closely resemble primary bone marrow-derived macrophages (BMDMs) in terms of PAMP-responsive transcriptomic responses and NFκB signaling dynamics (Sheu et al, 2023; Singh et al, 2024). A p38-specific KTR coupled to mCerulean (Regot et al, 2014) was then introduced into the hMPs by lentiviral transduction using a vector which ensures silencing-resistant expression of the transgene following differentiation (Müller-Kuller et al, 2015). Cells that were mVenus[+] (100% of single cell gate) and mCerulean[+] (17% of single cell gate) were collected by fluorescence-activated cell sorting (Fig. EV1A,B).

Given that KTRs are substrates of kinase activity but not signaling effectors, their ectopic expression is not expected to alter the signaling system parameters significantly. However, the dose responses of substrate phosphorylation can be sensitive to the introduction of additional kinase substrates (Rowland et al, 2012). We characterized the dual reporter cells by comparing phosphorylation of the endogenous p38-substrate MK2 and its downstream target CREB in the dual reporter hMPDMs and mVenus-Rela hMDPMs upon stimulation with three LPS doses over 2 h by Western Blotting and did not observe any appreciable differences (Figs. 1B and EV1C). We also confirmed that activation of p38, measured by phosphorylation, and of the NFκB pathway, measured as IκBα degradation, was unaffected (Figs. 1B and EV1C).

To test the expression, function, and specificity of the p38 reporter in differentiated hMPDMs, we imaged p38-KTR-expressing hMPDMs over 8 h with and without LPS stimulation. Before stimulation, the fluorescence intensity of the KTR was higher in the nucleus than in the cytoplasm; this ratio was heterogeneous between cells, as expected (Regot et al, 2014). While very little translocation of the KTR was observed upon mock stimulation, LPS stimulation induced translocation of a portion of nuclear KTR from the nucleus to the cytoplasm, visible at 30 min and increasing to 1 h (Fig. 1C). By 6 h post stimulation, the nuclear fluorescence intensity of the KTR had increased again, although not to baseline levels. Pre-incubation with a pharmacological p38 inhibitor decreased the observed translocation (Fig. 1C).

Adaptation of a previously established automated image analysis pipeline (Adelaja et al, 2021) allowed for quantification of the C/N ratio of p38-KTR fluorescence in hundreds of cells per stimulation condition over 8 h (Fig. 1D). Since the C/N ratio is used, intrinsic variability in reporter expression is controlled for (Regot et al, 2014). 30–60 min of baseline fluorescence was measured before in situ stimulation, allowing a per-cell baseline correction of the trajectories. The quantified trajectories confirmed the lack of translocation seen upon mock stimulation, the translocation out of and back into the nucleus upon LPS stimulation, representative of p38 activation and deactivation, and the strong decrease in translocation strength in the presence of p38 inhibitor, confirming the specificity of the KTR (Fig. 1D). The activity of p38 measured using this reporter showed heterogeneity in single cells responding to stimulation in characteristics such as speed, amplitude, and duration of signaling.

As different assays for MAPK activity are known to have different sensitivities, especially at low doses (Gillies et al, 2017), we compared mean (pseudo-bulk) p38 activity reported by KTR microscopy in response to three doses of the PAMPs LPS (lipopolysaccharide, binds TLR4, signals via MyD88 and TRIF adapters to NFκB, MAPK, and IRF pathways), P3C4 (Pam3CysSerLys4, binds TLR2/1, signals via MyD88 to NFκB and MAPK pathways), and CpG (CpG oligodeoxynucleotide, binds TLR9, signals via MyD88 to NFκB and MAPK pathways) and the host cytokine TNF (tumor necrosis factor, binds TNF receptor 1/2, signals via TRADD adapter to NFκB and MAPK pathways) over 4 h to bulk phospho-p38 levels by Western Blotting (WB) (Figs. 1E and EV1D). The KTR measurement recapitulated many features of stimulus- and dose-specific p38 activity also observed by WB, such as fast activation and deactivation dynamics, the stimulus-specificity of peak amplitudes and of activation and deactivation speed, as well as the dose-specificity of activation strength and speed. Specifically, the dose response and dynamics of CpG matched well between the two assays, with a slightly more pronounced deactivation observed by WB. The P3C4 dose response matched well for the relative peak amplitudes, although the timing of the peaks was slightly delayed using the KTR at lower doses. At low doses of LPS and TNF, the KTR measurements appeared to

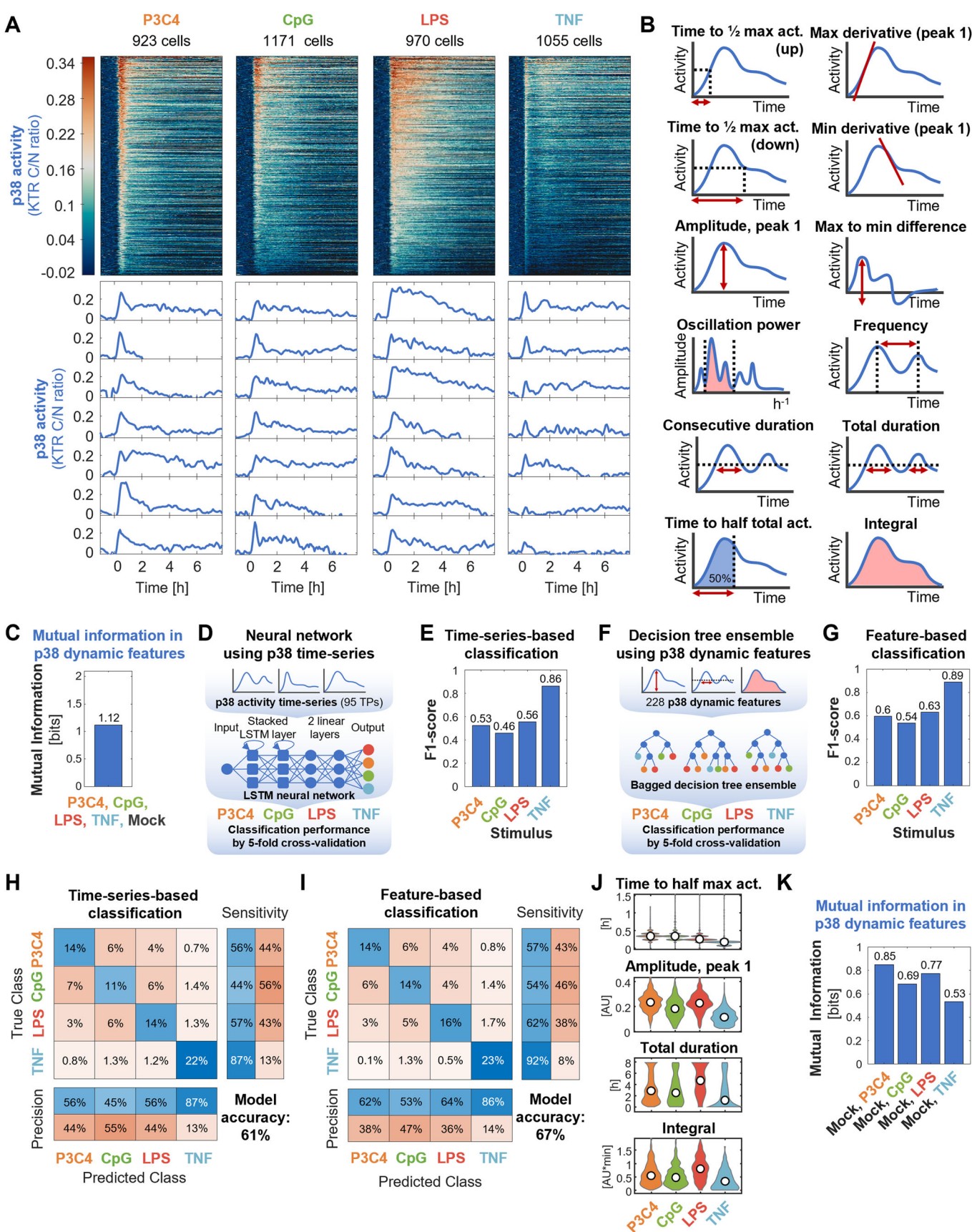

**Figure 2.** **MAPK p38 activity dynamics distinguish TNF from PAMPs with high accuracy.**

(A) MAPK p38 activity dynamics in response to 100 ng/ml P3C4, 1000 nM CpG, 100 ng/ml LPS, and 100 ng/ml TNF stimulation over app. eight hours were measured by fluorescence microscopy of reporter hMPDMs. Each row of the heatmap represents the p38 signaling trajectory of one cell. Trajectories are sorted by maximum amplitude. Example trajectories are shown below. Data from two pooled biological replicates are depicted. Total # of cells: 923, 1171, 970, and 1055 cells for P3C4, CpG, LPS, and TNF. (B) Schematic of 12 (out of 228) dynamic features derived from p38 activity trajectories used for quantitative analysis. (C) Mutual information between mock and high-dose P3C4, CpG, LPS, and TNF stimulations and dynamic features of p38 activity. Data from two pooled biological replicates are used. (D) Schematic of neural network machine learning classifier to test distinguishability of stimuli using p38 signaling time-series. (E) F1 scores by class of neural network classification of p38 time-series in response to the high-dose P3C4, CpG, LPS, and TNF stimulations. Data from two pooled biological replicates are used. (F) Schematic of decision tree ensemble machine learning classifier to test distinguishability of stimuli using p38 signaling dynamic features. (G) F1 scores by class of decision tree ensemble classification of p38 dynamic features in response to the high-dose P3C4, CpG, LPS, and TNF stimulations. Data from two pooled biological replicates are used. (H, I) Confusion matrices of neural network classification of p38 time-series (H) or decision tree ensemble classification of p38 dynamic features (I) in response to high-dose P3C4, CpG, LPS, and TNF stimulations. Data from two pooled biological replicates are used. (J) Violin plots of selected p38 dynamic features in response to the high-dose P3C4, CpG, LPS, and TNF stimulations. Data from two pooled biological replicates are depicted. Total # of cells: 923, 1171, 970, and 1055 cells for P3C4, CpG, LPS, and TNF. (K) Mutual information between mock and high-dose stimulation with P3C4, CpG, LPS, or TNF and dynamic features of p38 activity. Data from two pooled biological replicates are used.

yield less signal than the WB measurements, reflecting different characteristics of these measurement modalities. Similar results were obtained when comparing levels of the phosphorylated p38-substrate MK2 to p38-KTR microscopy results (Fig. EV1D,E).

To gain further confidence that p38-KTR appropriately represents p38 activity dynamics and heterogeneity, we compared p38 phosphorylation measured by flow cytometry to KTR measurements. The mean fluorescence intensity (MFI) within the first hour after stimulation recapitulated many aspects of the dynamics and dose responses observed for mean (pseudo-bulk) p38 activity reported by KTR microscopy, such as the timing, width, and relative amplitudes of peak activation, especially for CpG and LPS (Fig. 1F). In response to low-dose P3C4 and TNF, the KTR measurements showed more and less signal, respectively, than the flow cytometry, potentially within range of experimental variability (Fig. 1F). Importantly, at the single cell level, the fraction of p38-activated cells among all cells showed similar dose responses for P3C4, CpG, and LPS when measured by p-p38 flow cytometry and p38-KTR microscopy (Figs. 1G and EV1F,G), confirming that p38-KTR does not have a systematically lower sensitivity to low stimulus doses compared to p-p38 flow cytometry. However, for TNF, the p38-KTR measurements resulted in a lower fraction of activated cells, especially at lower doses, compared to flow cytometry measurements, suggesting a lower sensitivity of the KTR for low-dose TNF-induced p38 activity (Fig. 1G). The short duration or transience of TNF-mediated p38 activity may play a role. In summary, p38-KTR is a suitable reporter to use in combination with mVenus-RelA in hMPDMs to study the combinatorial signaling of p38 and NFκB in macrophages on a single-cell level.

## Stimulus-specificity of p38 responses: PAMPs can be distinguished from TNF

While it is known that NFκB signaling dynamics are highly stimulus-specific (Adelaja et al, 2021; Luecke et al, 2021), the stimulus-specific information contained in p38 signaling has not yet been determined. Therefore, before studying combinatorial encoding by NFκB and p38, we first compared p38 activity dynamics alone in response to stimulation with high doses of three different PAMPs (P3C4, CpG, and LPS) and with the host cytokine TNF using live cell microscopy of p38-KTR hMPDMs. After automated image analysis and stringent quality filtering of

trajectories (Table EV1), heatmaps displaying p38 activity trajectories of 923–1171 cells obtained in two biological replicates and example trajectories revealed stimulus-specificity of p38 activity (Fig. 2A). The stimulus-induced p38 dynamics differed, for example, in their amplitudes (with P3C4 and LPS having higher, CpG moderate, and TNF lower amplitudes), speed of activation (with TNF activating p38 very quickly and CpG more slowly), the strength of deactivation (with TNF having a very narrow first peak with a steep decline, P3C4 a moderately broad first peak, LPS a slow decline of activity, and CpG very heterogeneous width of the first peak), and duration (with LPS having extended activity times). Heterogeneity of e.g., speed, amplitude, and duration of signal between individual cells stimulated with the same ligand was observed for all ligands (Fig. 2A).

To quantitate the information content of the p38 dynamics, we decomposed the single cell time-series data into 228 dynamic features (Table EV2), including features describing the speed of activation and deactivation, amplitude, oscillatory character, duration, early vs. late activity, and total activity (Fig. 2B). We then determined the mutual information (MI) between the high-dose P3C4, CpG, LPS, TNF and mock stimulations and these dynamic features of p38 activity (a measure of correlation reported in bits, where 1 bit indicates perfect correlation when two conditions are contrasted), and found it to be 1.12 bits (Fig. 2C). This indicates that p38 dynamics distinguish these five stimulus conditions (maximum MI for five conditions being 2.32 bits) to some degree, though imperfectly.

To better understand how these stimuli are distinguishable by p38 activity, we used machine learning to classify the sources of p38 activities, meaning we trained machine learning models to predict the stimulus identity based on the single-cell p38 dynamic activities. First, we trained a Long Short-Term Memory (LSTM) recurrent neural network (RNN) to classify p38 time-series data (using vectors consisting of the 95 trajectory timepoints as input) from the four stimulations (Fig. 2D). The LSTM architecture type of RNN was chosen because it is well suited to directly learn about information in sequences from time-series data. Rather than treating the timepoints as discrete features, it considers the output of prior timepoints in the calculation of the current timepoint's output. Furthermore, the LSTM is an improvement on the standard RNN that can better handle longer sequences (Hochreiter and Schmidhuber, 1997; Van Houdt et al, 2020; Yu et al, 2019). We performed fivefold cross-validation to evaluate model performance.

The by-class F1 scores (harmonic mean of precision and sensitivity of the classification) revealed an especially high performance of the model for TNF (F1 score: 0.86), while CpG, P3C4, and LPS had similar F1 scores between 0.46 and 0.56, with CpG performing worst and LPS performing best among the three (Fig. 2E).

We then characterized p38 dynamical trajectories in terms of dynamical features, using 228 dynamic features to abstract p38 dynamical trajectories from the four stimulation conditions and used the resulting values as input to a decision tree ensemble classifier, which for these types of classification problems often strike a good balance with regards to performance, avoidance of overfitting, and feasibility of implementation (Adelaja et al, 2021). We similarly evaluated this classification using 5-fold cross-validation (Fig. 2F). The by-class F1 scores obtained using this feature-based classifier confirmed that TNF stimulation was particularly distinguishable with an F1 score of 0.89 and that CpG, P3C4, and LPS were more difficult to distinguish with F1 scores between 0.54 and 0.63 (Fig. 2G).

The confusion matrices derived from the 5-fold cross-validations of the two classifiers (which shows how the data in each category was classified) confirmed that TNF-derived p38 activity was classified with high sensitivity (87/92%) and rarely misclassified as another stimulus (Fig. 2H,I). P3C4- and LPS-triggered p38 activities were classified with sensitivities of 56/57% and 57%/62%, respectively, and were both most commonly misclassified as CpG. CpG had the lowest sensitivity of 44/54%.

The similarity in results between these two distinct classification approaches, the LSTM classification of time-series data and the decision tree classifier of dynamic features extracted from the time series, provide confidence in these findings and confirmed that the extracted dynamic features captured aspects of p38 dynamics relevant to the distinguishability of p38-inducing stimuli (Fig. 2E,H vs. 2G,I). The slightly better performance of the feature-based classifier (overall model accuracy: 67%) over the time-series classifier (overall model accuracy: 61%) may reflect the fact that several dynamic features are based on multiple timepoints and may thus neutralize technical noise, although other differences between these very distinct machine learning models may contribute also.

A key finding of these classification analyses is that p38 activation dynamics distinguish the host cytokine TNF from the bacterial and viral PAMPs tested. A closer look at some of the dynamic features revealed that TNF-induced p38 activity is characterized by quick activation speed and low amplitude of the first peak, duration, and total activity (Fig. 2J). One consequence of such low activity might be that TNF is less reliably distinguished from mock stimulation; indeed, we found that 0.53 bits of mutual information separate TNF from the mock condition, while 0.69, 0.77, and 0.85 bits of mutual information separate CpG, LPS, and P3C4 from mock stimulation, respectively (Fig. 2K), though the precise quantitation will be affected by the response speed of the reporter or physiological kinase substrate.

In concordance, when classifying dynamics features from all four stimuli and mock stimulation, compared to the classification of the four stimuli only, the F1 score of the TNF class decreased most (from 0.89 to 0.8) (Fig. EV2A vs. Fig. 2G). The confusion matrix demonstrates that TNF is most often misclassified as mock stimulation and vice-versa (Fig. EV2B). The second most common misclassification for mock stimulation is CpG (Fig. EV2B) and its F1 score decreases from 0.54 to 0.48, while the F1 scores of P3C4

and LPS are less affected by the inclusion of mock stimulation in the classifier (Fig. EV2A vs. Fig. 2G).

## MAPK p38 reinforces NFκB distinctions of ligand identity

Next, we sought to investigate how p38 signaling combines with NFκB signaling, as both pathways are often activated by the same ligands. We used the dual reporter hMDPMs to measure PAMP- and TNF-induced NFκB and p38 dynamics in the same cells (Fig. 3A) and found that NFκB and p38 dynamics were remarkably distinct from each other in single cells. NFκB dynamics showed the previously reported stimulus-specificity, with respect to the speed of response, amplitude, duration, and oscillatory content (Adelaja et al, 2021). For example, TNF-induced NFκB activities were often strongly oscillatory, while bacterial PAMPs elicited primarily non-oscillatory dynamics.

We first asked whether stimulus-specific p38 signaling contributes to stimulus-specific NFκB signaling to improve the cell's ability to distinguish different stimuli. To this end, we first trained the machine learning classifiers using either time-series or dynamic features of NFκB from all four stimulus conditions. Confusion matrices of the classifications showed that while NFκB activity allowed for consistently superior classification accuracy of the PAMP classes compared to p38, identification of TNF was similarly high (slightly higher using time-series, slightly lower using dynamic features) (Fig. 3B,C). We then trained machine learning models with the combined activities NFκB and p38 and found slight improvements in the accuracy of most classifications, with TNF identification again standing out as particularly accurate (Fig. 3B,C). Correspondingly, the overall classification accuracies, which were 61 and 67% for p38 time-series and p38 feature-based classification, respectively, and 74 and 70% when classifying NFκB time-series data and dynamic features, respectively, increased to 79 and 74% when classifying their combined activity, suggesting that p38 contributes to stimulus distinguishability provided by NFκB (Fig. 3D,E). As a control, we trained classifiers using p38 or NFκB activities in combination with incorrectly matched NFκB or p38 activities, respectively, from cells randomly selected from all stimulations ("shuffled"). Using such 'shuffled' NFκB + p38 inputs, neither time-series and nor feature-based classifiers yielded higher overall model accuracies (Fig. 3D,E) or improved patterns of confusion between stimuli (Fig. EV2C,D) compared to using the corresponding single-pathway inputs.

Focusing on the distinguishability of TNF from PAMPs, the by-class F1 scores confirmed that p38 and NFκB both allow for the identification of TNF-induced activity with high sensitivity and precision (F1 scores: 0.85–0.89), with neither providing consistently superior classification in the two classification methods (Fig. 3F,G). Combined p38 + NFκB activity increased TNF's F1 score slightly to 0.94/ 0.91 for time-series and feature-based classifier, respectively. Shuffling either NFκB or p38 activities provided confirmation that the small increase is due to properly matched combined activities (Fig. 3F,G). Thus, both signaling pathways independently distinguish host cytokine TNF from PAMPs and combining their activities improve the reliability only modestly.

We then asked whether their combined activity improves the distinction of TNF from mock. Mutual information calculations revealed that 0.86 bits separated TNF-induced NFκB activity from mock. Considering p38 in combination with NFκB provided no

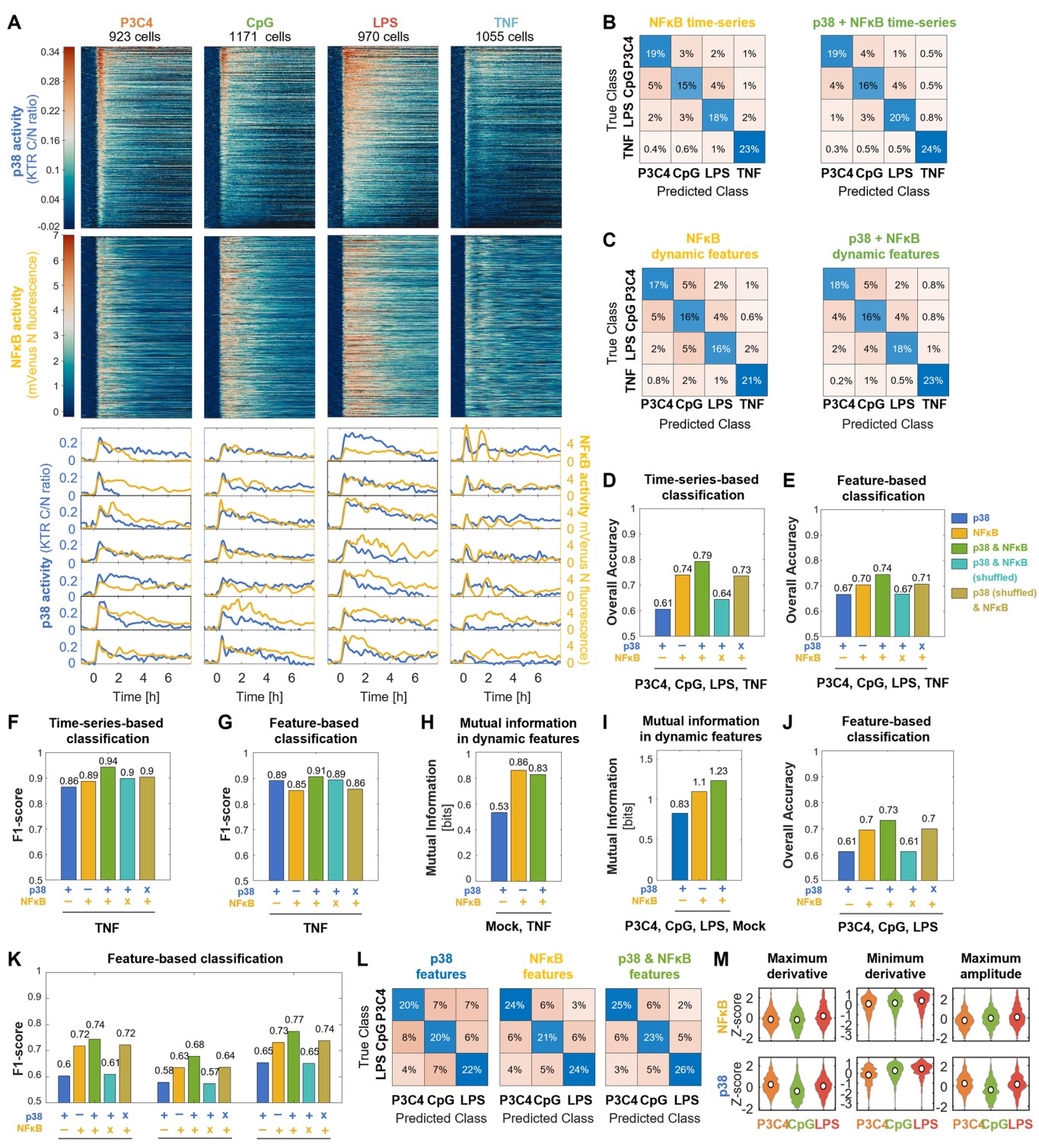

improvement with these calculations, resulting in 0.83 bits for the combined activities and in 0.53 bits for p38 alone (Fig. 3H).

In concordance, when training a classifier on both mock and all four stimuli (Fig. EV2E) or just the four stimuli (Fig. 3C), the overall accuracies using NFκB features or NFκB + p38 features barely increased (0.70/0.74 vs. 0.74/0.77) (Fig. 3E vs. EV2F) and the

F1 scores for the TNF class remained similar (0.85/0.84 vs. 0.91/ 0.89) (Fig. 3G vs. EV2G), while in a classifier using p38 features inclusion of mock stimulation resulted in similar overall accuracies (0.67 vs 0.66) and decreased the TNF F1 score (0.89 to 0.8) (Fig. EV2F, G). Overall, our results support the conclusion that NFκB is primarily responsible for distinguishing host cytokine TNF

◀ **Figure 3.  MAPK p38 contributes little to NFκB dynamics in the distinction of stimulus ligands.**

(A) p38 and NFκB activity dynamics in the same cells in response to 100 ng/ml P3C4, 1000 nM CpG, 100 ng/ml LPS, and 100 ng/ml TNF stimulation over app. eight hours were measured by fluorescence microscopy of reporter hMPDMs. Each row of the heatmap represents the p38 or NFκB signaling trajectory of one cell. Trajectories are sorted by the maximum amplitude of p38 activity. Example trajectories are shown below. Data from two pooled biological replicates are depicted. Total # of cells: 923, 1171, 970, and 1055 cells for P3C4, CpG, LPS, and TNF. p38 data is also shown in Fig. 2A. (B) Confusion matrices of neural network classification of NFκB (left) or NFκB + p38 (right) time-series in response to high-dose P3C4, CpG, LPS, and TNF stimulations. Data from two pooled biological replicates are used. (C) Confusion matrices of decision tree ensemble classification of NFκB (left) or NFκB + p38 (right) dynamic features in response to high-dose P3C4, CpG, LPS, and TNF stimulations. Data from two pooled biological replicates are used. (D, E) Overall classification model accuracy of machine learning classifications using p38 only, NFκB only, NFκB + p38, NFκB ("shuffled": incorrectly matched NFκB activities from cells randomly selected from all classes) + p38, NFκB + p38 ("shuffled": incorrectly matched p38 activities from cells randomly selected from all classes) time-series (D) or dynamic features (E) in response to high-dose P3C4, CpG, LPS, and TNF stimulations. Data from two pooled biological replicates are used. (F, G) F1 score for TNF class in machine learning classifications using p38 only, NFκB only, NFκB + p38, NFκB ("shuffled": incorrectly matched NFκB activities from cells randomly selected from all classes) + p38, NFκB + p38 ("shuffled": incorrectly matched p38 activities from cells randomly selected from all classes) time-series (F) or dynamic features (G) in response to high-dose P3C4, CpG, LPS, and TNF stimulations. Data from two pooled biological replicates are used. (H) Mutual information (MI) between mock and high-dose stimulation with TNF and dynamic features of p38, NFκB, or p38 + NFκB activity. Data from two pooled biological replicates are used. (I) Mutual information between mock and high-dose P3C4, CpG, and LPS stimulations and dynamic features of p38, NFκB, or p38 + NFκB activity. Data from two pooled biological replicates are used. (J, K) Overall classification accuracy (J) and F1 scores for individual classes (K) of machine learning classifications using p38 only, NFκB only, NFκB + p38, NFκB ("shuffled": incorrectly matched NFκB activities from cells randomly selected from all classes) + p38, NFκB + p38 ("shuffled": incorrectly matched p38 activities from cells randomly selected from all classes) dynamic features in response to high-dose P3C4, CpG, and LPS stimulations. Data from two pooled biological replicates are used. (L) Confusion matrices for machine learning classifications using p38 only (left), NFκB only (center), and NFκB + p38 (right) dynamic features in response to high-dose P3C4, CpG, and LPS stimulations. Data from two pooled biological replicates are used. (M) Selected dynamic features of NFκB (top) and p38 (bottom) activity in response to P3C4, CpG, and LPS. Data from two pooled biological replicates are used. Total # of cells: 923, 1171, and 970 cells for P3C4, CpG, and LPS.

stimulation from mock, but that p38 independently reliably distinguishes being exposed to TNF from being exposed to PAMPs.

We next asked whether combinatorial signaling by p38 and NFκB improved the distinction of the three PAMPs. While p38 dynamic features provided 0.83 bits of mutual information for P3C4, CpG, LPS, and mock, NFκB dynamic features provided 1.1 bits, and the combination provided 1.23 bits (Fig. 3I). Overall classification accuracy of time-series and feature-based classifiers was 60%/61% using p38 activity, 70% using NFκB activity, and 75/73% using the combination of activities (Figs. EV2H and 3J). These three separate analyses consistently indicated that p38 dynamics provide limited ability to distinguish PAMPs on their own and contribute little information in combination with NFκB dynamics.

The by-class F1 scores from time-series and feature-based classification revealed that, of the three PAMPs, LPS had the highest F1 scores for p38 only, NFκB only, and p38 + NFκB classifications, making it the most distinguishable of the three (Figs. EV2I and 3K). When considering the combined activity compared to NFκB alone, the classification of CpG and LPS improved more than that of P3C4, which showed only a minor increase. Considering the confusion matrices, all stimuli showed a high degree of confusion using p38 activity alone, with CpG being misclassified as P3C4 slightly more often than as LPS (Fig. EV2J and 3L). With combined activities, misclassification of P3C4 as LPS (or the reverse) was particularly rare. Misclassification of CpG as LPS was reduced when the combined activities were considered. The control classifications using shuffled p38 or NFκB activity performed similarly to single-pathway classifiers with respect to classification accuracy (Figs. EV2H and 3J), F1 scores (Figs. EV2I and 3K), and confusion between classes (Figs. EV2J,K). Visualization of selected dynamic features demonstrates how p38 features may contribute to the distinction of PAMPs: While P3C4- and CpG-induced NFκB trajectories show very similar distributions of "maximum derivatives", these two ligands appear more distinct in the "maximum derivatives" of their p38 dynamics (Fig. 3M, left). Similarly, p38 "minimum derivatives" show more distinction between LPS and P3C4 for p38 than for NFκB (Fig. 3M, center),

and p38 "maximum amplitude" may contribute to the distinction of LPS and CpG (Fig. 3M, right). In summary, p38 contributes stimulus-specific signaling to allow for slightly more accurate identification of difficult-to-distinguish stimuli, such as MyD88-activating PAMPs, than NFκB activity alone.

## MAPK p38 does not improve dose distinction beyond that achieved by NFκB dynamic features

NFκB dynamic features are known to convey some dose-specificity (Adelaja et al, 2021). A previous study had reported a differential dose response of NFκB and MAPK to LipidA (a TLR4-ligand) stimulation in macrophages, with MAPK's switch-like dose response ensuring response distinction between harmless and harmful PAMP levels (Gottschalk et al, 2016). Thus, we next sought to determine whether NFκB and p38 dynamics act combinatorially to distinguish stimulus doses. To this end, we determined p38 and NFκB activity over 8 h for 713–1259 cells over two biological replicates for 5 doses of P3C4, CpG, LPS, or TNF (Fig. 4A; Appendix Fig. S1A–C). The doses spanned a concentration range of $10^5$. The highest dose was chosen to saturate NFκB responses with respect to the proportion of responding cells and the lowest dose to provide NFκB activity similar to mock stimulation. Both p38 and NFκB activity trajectories appeared dose-specific. For example, in response to increasing P3C4 dose, p38 amplitude and speed of response increased while the width of the first peak decreased. NFκB and p38 appeared to have different activation thresholds, e.g., with 0.1 ng/ml P3C4 activating almost NFκB in almost all cells, but p38 in a smaller portion of cells (Fig. 4A).

To quantify the differential activation thresholds, we fit Hill curves to dose responses of the fraction of cells with p38 or NFκB activity upon P3C4 stimulation (Fig. 4B) and determined the ligand concentration that provided the half maximum percentage of NFκB and p38 responder cells (Fig. 4C). Activation of p38 required a 6.7x higher P3C4 dose than NFκB activation, confirming a differential dose response. We hypothesized that the NFκB and p38 differential dose-response behavior does not apply to all stimuli equally.

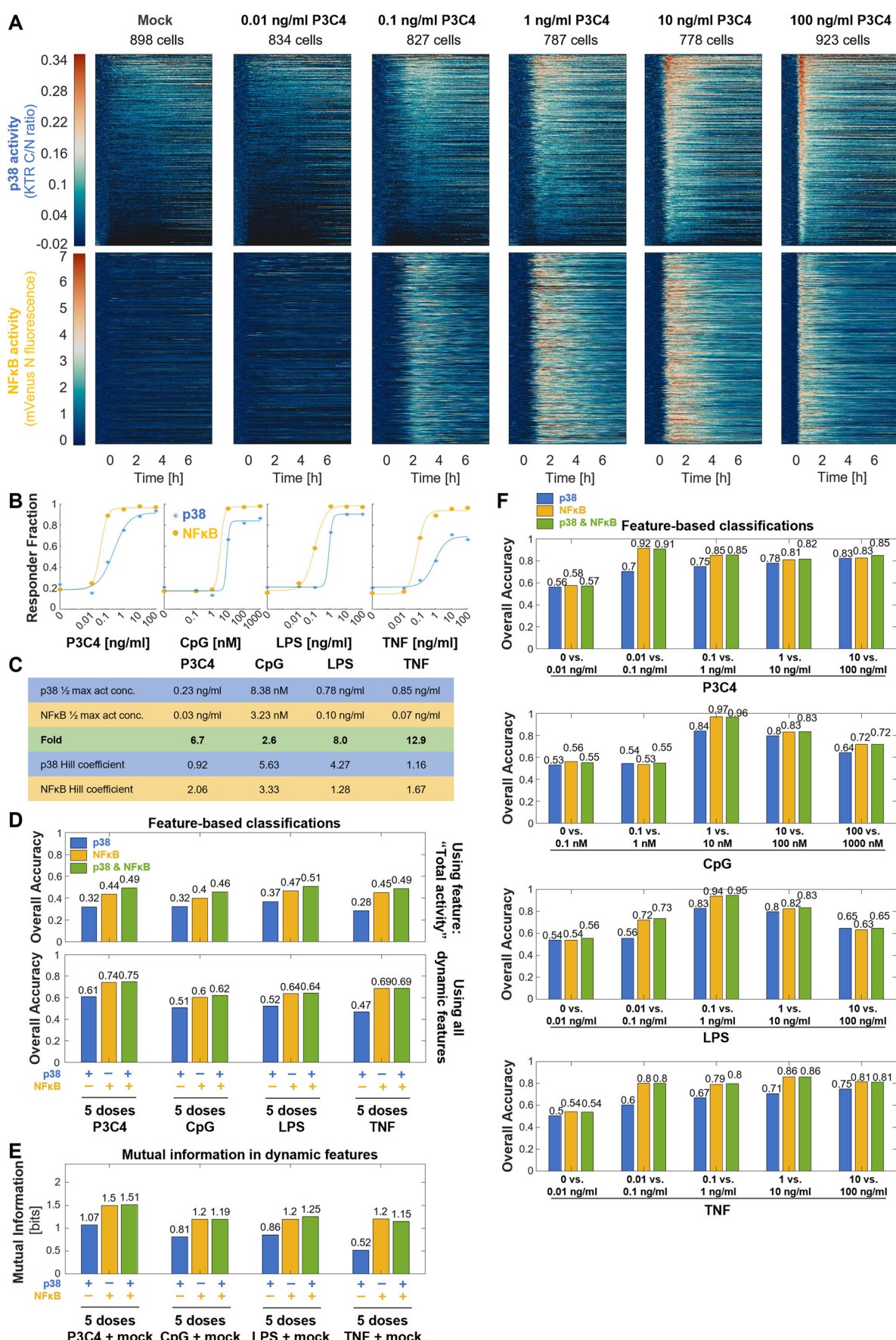

**Figure 4.  MAPK p38 does not improve dose distinction beyond that achieved by NFκB dynamics.**

(A) p38 and NFκB activity dynamics in the same cells in response to 0, 0.01, 0.1, 1, 10, and 100 ng/ml P3C4 stimulation over app. eight hours were measured by fluorescence microscopy of reporter hMPDMs. Each row of the heatmap represents the p38 or NFκB signaling trajectory of one cell. Trajectories are sorted by the maximum amplitude of p38 activity. Data from two pooled biological replicates are depicted. Total # of cells: 898, 834, 827, 787, 778, and 923. Data for 100 ng/ml stimulation is also shown in Figs. 2A, 3A. (B) Dose responses of NFκB and p38 activity, as measured by the fraction of responding cells among all cells, in response to six doses of P3C4 (0, 0.01–100 ng/ml), CpG (0, 0.1–1000 nM), LPS (0, 0.01–100 ng/ml), or TNF (0, 0.01–100 ng/ml), with Hill curve fits. Data from two pooled biological replicates are used. (C) Parameters of the Hill fits of dose-response curves (from Fig. 4B) and fold difference between concentrations of half maximum activity for p38 and NFκB. (D) Overall classification accuracy of decision tree ensemble classifications using p38 only, NFκB only, or NFκB + p38 total activities only (top) or all dynamic features (bottom) in response to 5 doses of P3C4 (0.01–100 ng/ml), CpG (0.1–1000 nM), LPS (0.01–100 ng/ml), or TNF (0.01–100 ng/ml). Data from two pooled biological replicates are used. (E) Mutual information between 6 doses of P3C4 (0, 0.01–100 ng/ml), CpG (0, 0.1–1000 nM), LPS (0, 0.01–100 ng/ml), or TNF (0, 0.01–100 ng/ml) and dynamic features of p38, NFκB, or p38 + NFκB activity. Data from two pooled biological replicates are used. (F) Overall classification accuracies of decision tree ensemble classifications using p38 only, NFκB only, or NFκB + p38 dynamic features in response to the respective two indicated adjacent doses of the indicated stimuli (P3C4, CpG, LPS, or TNF). Data from two pooled biological replicates are used. Source data are available online for this figure.

Indeed, we observed stimulus-specific differential dose responses: The TNF concentration at half maximum activity for p38 was 12.9x that of NFκB; for LPS, it was 8x, and, for CpG 2.6x. (TNF's dose response may be affected by the slightly decreased sensitivity of the KTR for low TNF doses (0.1 ng/ml) compared to flow cytometry or Western blot-based measurements (see Fig. 1E–G)). Thus, we confirmed that the differential dose response between NFκB and p38 previously reported for LipidA can be observed for other immune stimuli as well.

Based on the differential dose response, one might hypothesize that combinatorial signaling between p38 and NFκB would increase the distinguishability among stimulus doses compared to the dose-information contained in p38 or NFκB activity individually. To investigate this, we first trained machine learning classifiers of dose responses of the four stimuli trained only on "total activity" of p38, NFκB, or both. This resulted in low classification accuracy for the p38 activity-based classifiers of the 5 stimulus doses (0.28–0.37) for all four stimuli and slightly higher classification accuracies using NFκB activity (0.4–0.47) (Fig. 4D). Using both p38 and NFκB total activity for classification further increased the classification accuracy (0.46–0.51). However, when using all dynamic features instead of only 'total activity' to classify stimuli doses (Appendix Fig. S2), combining p38 and NFκB features did not consistently result in an increase in classification accuracy across stimuli compared to NFκB features alone or p38 features alone (0.62–0.75 vs. 0.6–0.74 and 0.47–0.61, respectively) (Fig. 4D). Similarly, in a mutual information calculation the combined dynamic features of p38 and NFκB did not provide consistently more dose information than NFκB alone or p38 alone (1.15–1.51 bits vs. 1.2–1.5 bits and 0.52 to 1.07 bits, respectively) (Fig. 4E).

We next asked whether p38 may contribute combinatorially to improve the binary distinction of certain adjacent doses. Similar patterns as in the full dose range classifications were observed in the binary distinction of adjacent stimulus doses across the dose ranges, with NFκB and NFκB + p38 dynamics providing similarly accurate classification (Fig. 4F). p38 dynamic features alone generally provided less accurate classification than either NFκB or NFκB + p38, especially at low and medium doses below or near its activation threshold, but interestingly approached the accuracy of NFκB-based classifications at medium-to-high doses (e.g.,10 vs. 100 ng/ml P3C4) (Fig. 4F). This suggests that p38 can independently provide dose-specific information at higher doses.

Since previous work suggested that p38's role in dose sensing may be to distinguish harmful from harmless PAMP

concentrations (Gottschalk et al, 2016), we next sought to determine how well mock stimulation can be distinguished from individual stimulus doses. Similar to the classification of adjacent doses, in binary classification of mock stimulation vs individual stimulus doses, both NFκB and p38 dynamics achieved highly accurate classifications for doses above their activation threshold, with NFκB generally performing better than p38, with the size of the difference in performance being consistent with their stimulus-specific differential activation thresholds. However, their combined dynamic features did not increase the classification accuracy (Appendix Fig. S3).

Thus, while p38 dynamics on their own are dose-specific to a certain degree and provide accurate identification of medium to high-dose stimulation, they do not increase the total dose information available when considering all dynamic features given the richness of the information contained in NFκB dynamic features. However, when considering only 'total activity', p38 and NFκB pathways combine to slightly increase the accuracy of dose identification.

## Heterogeneous MAPK p38 and NFκB dynamic features are poorly correlated across cells

The activation of p38 is mediated by signaling pathways that are branched off a receptor-proximal signaling module that is shared with NFκB. We sought to understand whether the heterogeneous dynamic features of NFκB and p38 are correlated across the population of single cells. Spearman correlation coefficients of corresponding p38 and NFκB dynamic features were generally low, even for high-dose ligands (Fig. 5A). Interestingly, features determined early in the time course, such as 'time to half maximum activity' showed the highest, but still modest, positive correlations for LPS, P3C4, and CpG (correlation coefficients of 0.59 to 0.74) (but weak correlations for TNF, 0.12), while features determined a little later in the time course, such as 'maximum amplitude', showed very weak or no correlations (−0.08 to 0.11), and features determined late in the time course, such as "duration" and "total activity", showed weakly negative correlations (−0.12 to −0.23) (Fig. EV3A). Following shuffling the cell identities for the NFκB trajectories within a stimulation condition, correlation coefficients were all close to 0 and rarely statistically significant (Fig. EV3B). This suggests that the weak correlations in the correctly matched data may represent relevant relationships between the features. We examined the correlation along the time course of stimulation by

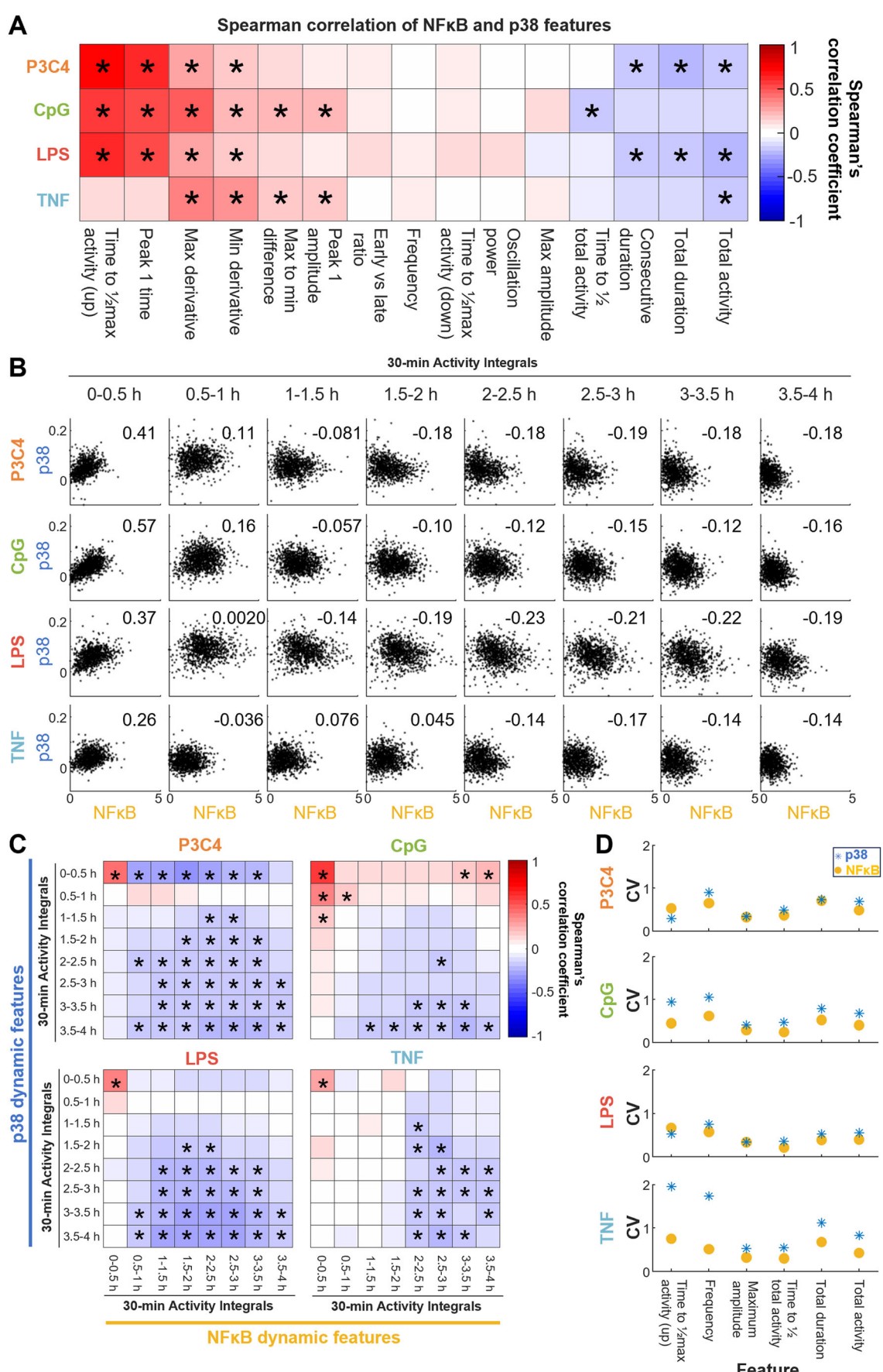

**A** Spearman correlation of NFκB and p38 features

**B** 30-min Activity Integrals

**C**

**D**

Figure 5.   Single-cell MAPK p38 and NFκB dynamic features are poorly correlated.

(A) Spearman correlation coefficients (CCs) between indicated p38 dynamic features and corresponding NFκB dynamic features upon stimulation of hMPDMs with high-dose P3C4, CpG, LPS, or TNF. Features are sorted by means of CCs across stimuli. Asterisk indicates statistically significant correlation ($p < 0.05$) with $|CC| > 0.15$. Data from two pooled biological replicates are used. (B) Scatter plots of 30-min p38 activity integrals from 0 to 4 h and corresponding NFκB activity integrals upon stimulation with high-dose P3C4, CpG, LPS, or TNF. Number indicates the Spearman correlation coefficient. Data from two pooled biological replicates are depicted. (C) Spearman correlation coefficients (CCs) between p38 activity integrals from 0 to 4 h and all NFκB activity integrals from 0 to 4 h upon stimulation with high-dose P3C4, CpG, LPS, or TNF. Asterisk indicates a statistically significant correlation ($p < 0.05$) with $|CC| > 0.15$. Data from two pooled biological replicates are used. (D) Coefficient of variation of indicated p38 and NFκB dynamic features upon stimulation with high-dose P3C4, CpG, LPS, or TNF. Data from two pooled biological replicates are used.

calculating the correlation coefficients for half-hour integrals of stimulus-responsive p38 and NFκB activities. While the integrals from 0–0.5 h were moderately positively correlated (0.26 to 0.57), correlation coefficients of the following interval integrals were smaller or close to 0 (e.g., −0.14 to 0.076 for the 1–1.5 h activity integral), and then weakly negative (−0.14 to −0.19 for the 3.5–4 h activity integral) (Figs. 5B and EV3C).

The correlations between all p38 30-min activity integrals with all NFκB 30-min activity integrals confirmed this dynamical pattern with some stimulus-specific differences (Fig. 5C). The only consistent moderately positive correlation across all four stimuli was the correlation of the 30-min integrals of NFκB and p38 activity between 0–0.5 h (Fig. 5C, see also Fig. 5B), while there tended to be weak negative correlations almost consistently between all 30-min integrals from 2–4 h (Fig. 5C). Similarly, correlations across select p38 and NFκB dynamic features confirmed that there are largely weak correlations between the features, which are often weakly negative for features determined later in the time course (Fig. EV3D). As seen in Fig. 5A, p38 and NFκB "time to ½ max activity", representing the speed of the response, was modestly positively correlated for all stimuli, except TNF. Other, weaker, positive correlations include p38 "time to ½ max activity" with NFκB "maximum amplitude" and NFκB "total activity" for P3C4 and LPS and NFκB "frequency" for CpG. NFκB "time to ½ max activity" is weakly negatively correlated with p38 'maximum amplitude' in all four stimuli and with p38 "total duration" and p38 "total activity". Especially for P3C4 and LPS, p38 "time to ½ total activity", "duration" and "total activity" are weakly negatively correlated with NFκB "max amplitude" as well as NFκB "total duration" and "total activity" (Fig. EV3D). In summary, the heterogeneous p38 and NFκB activities in single cells are positively correlated only very shortly after stimulation with different stimuli, with most dynamic features showing poor correlation, and later activities showing weakly negative correlations, suggesting the presence of negative cross-regulation mechanism(s).

We then quantified the heterogeneity of select p38 and NFκB dynamic features for high-dose stimulation using the coefficient of variation (CV), a mean-normalized measure of variability. Comparisons of the LPS-induced signal distributions produced by the p38 KTR-reporter and p-p38 flow cytometry showed similar widths, suggesting that the KTR represents biological p38 heterogeneity well (Fig. EV3E). Across stimuli, the CV for both p38 and NFκB for most features was generally below or near 1, with p38 and NFκB generally having similar CVs. Notable exceptions were "time to ½ max activity" and "frequency" of TNF stimulation, for which p38's CVs were close to 2, thus much higher than for NFκB, which were below 1 (Fig. 5D). Quantifying the heterogeneity of the dynamic features 'time to ½ max activity', 'maximum amplitude', and "total activity" across doses revealed that the CVs generally decreased with increasing dose and that p38 and NFκB dynamic features had similar CVs, indicating that stimulus-responsive dynamics had lower mean-normalized heterogeneity than mock-induced dynamics (Fig. EV3F). TNF-induced dynamics generally had larger differences between p38 and NFκB across doses than other stimuli (Fig. EV3F). The CV of p38 and NFκB 30-min activity integrals from 0 to 4 h for high-dose stimulations showed a similar pattern, with p38 activity CVs being similar or slightly higher than NFκB CVs. The difference between p38 and NFκB CVs increased towards later timepoints because the increase in activity CVs generally seen between 0.5 and 1 h to 3.5–4 h was more pronounced for p38 than for NFκB. TNF-induced activities had the largest differences between p38 and NFκB CVs, due to larger CVs for p38 activity integrals, LPS the smallest (Fig. EV3G). Thus, there are comparable levels of heterogeneity in stimulus-induced p38 and NFκB dynamic features, but these are poorly correlated in a heterogenous single cell population.

## Mathematically modeling single-cell p38 and NFκB dynamics suggests sources of heterogeneity

Given the complex heterogeneity patterns in the two pathways, we sought to investigate whether the known mechanisms of these signaling pathways might account for the single cell p38 and NFκB trajectories using a mathematical modeling approach. To this end, we integrated a newly built model of the p38 activation pathways via IKK/Tpl2/MKK3/6 and via TAK1/MKK4 (Pattison et al, 2016; Luecke et al, 2021) with a detailed, established model of stimulus-induced NFκB activation (Figs. 6A and EV4A) (Adelaja et al, 2021; Luecke et al, 2023). The p38 module of the model was parameterized to fit representative p38 activity trajectories after high-dose P3C4, CpG, LPS, and TNF stimulation. The resulting model was able to correctly simulate many aspects of the experimental p38 dynamics of cells chosen for their representative stimulus-specific dynamics (Fig. 6B, top). The experimentally measured NFκB dynamics of those representative hMPDMs showed a good match with the simulated 'representative cell' NFκB trajectories (Fig. 6B, bottom).

The parameterized model suggested that the two pathway branches activating p38 may act with different speeds, with the fast-activated MKK6 branch mediating the initial phase of p38 activity and the MKK4 branch, which takes longer to reach its maximal simulated activation, extending the p38 activity to later timepoints (Fig. 6C). As a result, the two pathway branches were predicted to be activated with stimulus-specific dynamics, mediating the stimulus-specificity of p38.

Cell-to-cell heterogeneity of signaling may often be accounted for by heterogeneity in a handful of parameters (Cheng et al, 2015;

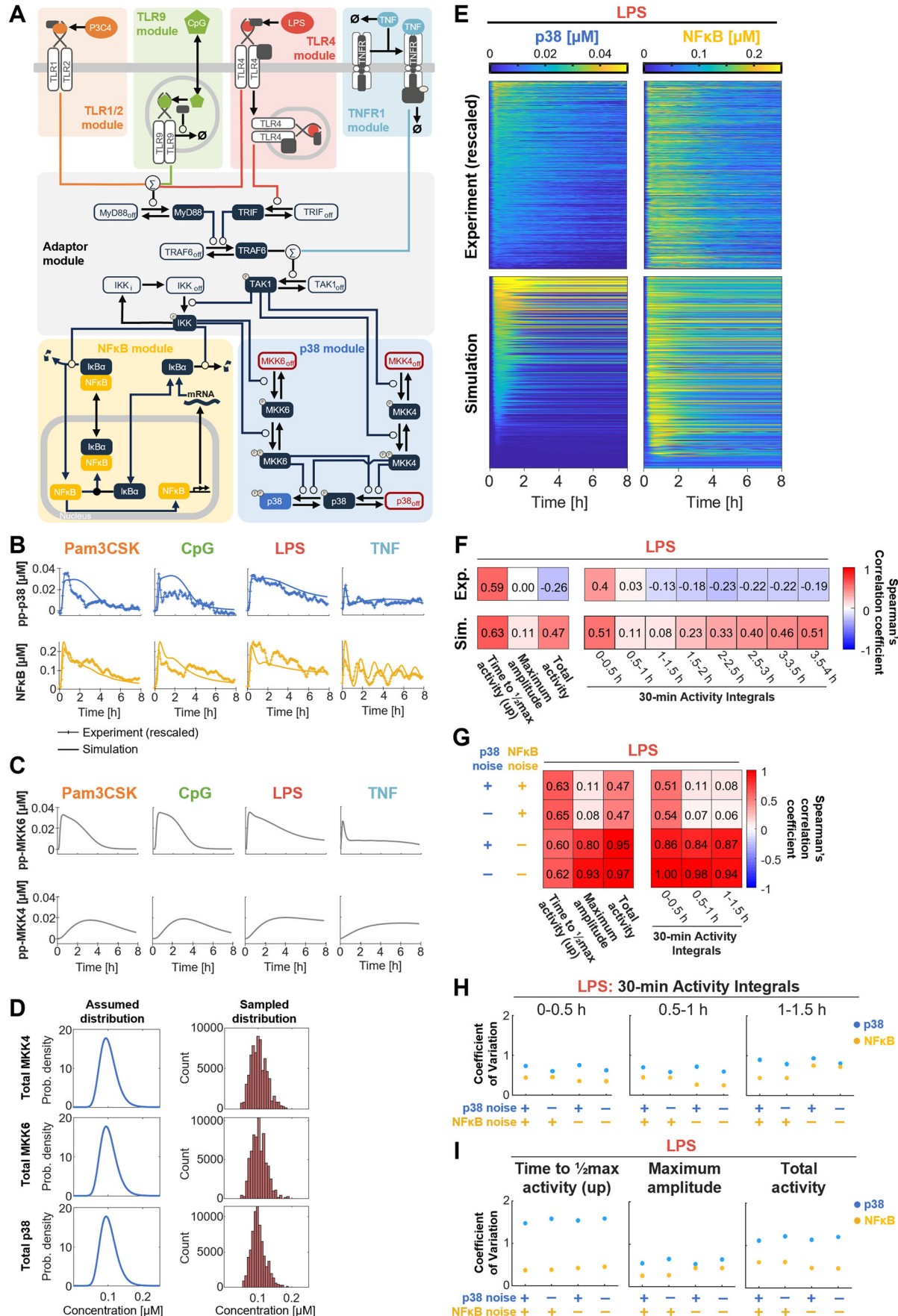

**Figure 6.  Integrating a mathematical model of single-cell p38 activation with a model of NFκB dynamics suggests sources of heterogeneity.**

(A) Schematic of mathematical model structure. A model of p38 activation via IKK/Tpl2/MKK3/6 and TAK1/MKK4 (blue background) is integrated with established models of NFκB activation (yellow) downstream of TLR1/2 (orange), TLR9 (green), TLR4 (red), and TNF receptor (light blue) signaling (Adelaja et al, 2021). Within the p38 module, parameters in red indicate the parameters distributed to simulate heterogeneous single cell trajectories. All model modules except the p38 module are simplified. The full model schematic can be found in Fig. EV4. (B) Simulation of 'representative cell' p38 and NFκB activity trajectories downstream of high-dose P3C4, CpG, LPS, and TNF stimulation after parameter fitting to representative p38 trajectories overlaid with the experimental p38 and NFκB trajectories from the representative cells. Example trajectories were scaled to model units as described in Materials and Methods. (C) Simulation of "representative cell" MKK4 and MKK6 activity trajectories downstream of P3C4, CpG, LPS, and TNF stimulation. (D) Assumed distributions of total MKK4, MKK6, and p38 concentrations and parameter values sampled from those distributions for simulation of cell-cell-heterogeneity of p38 activity in (E). (E) Simulated LPS-induced p38 and NFκB trajectories in corresponding cells compared to experimentally determined p38 and NFκB trajectories rescaled to model units as described in Materials and Methods. Cell-cell-heterogeneity is simulated by distributing starting concentrations of total p38, MKK4, and MKK6 (as in D), in addition to the distribution of receptor- and NFκB-related parameters. Trajectories are sorted by the maximum amplitude of p38 activity. 500 cells are shown. (F) Spearman correlation coefficients between indicated p38 dynamic features and corresponding NFκB dynamic feature in simulated (sim.) or experimental (exp.) stimulation with LPS. For experimental data, two pooled biological replicates were used. Values are rounded to two decimals. (G) Spearman correlation coefficients between indicated p38 dynamic features and corresponding NFκB dynamic feature in simulated LPS stimulation with or without simulated molecular noise (i.e., using parameter distribution or single parameter) in the p38 and NFκB modules. An average of ten simulations is reported. (H, I) Coefficient of variation of 30-min activity integrals (H) or indicated dynamic features (I) of p38 and NFκB activity upon simulated LPS stimulation with or without simulated molecular noise (i.e., using parameter distributions or single parameters) in the p38 and NFκB modules. Results from one simulation are reported. Source data are available online for this figure.

Luecke et al, 2023). To explore whether such assumptions may also account for the heterogeneity of MAPK signaling, we explored the response to LPS. To simulate single cell heterogeneity, the total concentrations of the kinases MKK4, MKK6, and p38 were sampled from log-normal distributions with mean values matching the 'representative cell' model established above (Fig. 6A,D), in addition to distributing receptor- and NFκB-related parameters. This indeed resulted in highly heterogeneous simulated single cell p38 trajectories similar to experimental data (Fig. 6E).

Next, we sought to understand the effects of varying individual parameters in the p38 module on LPS-induced p38 dynamics using a parameter sensitivity analysis. In a simulated 'representative cell', increasing or decreasing the total p38 concentration led to a strong increase or decrease, respectively, in "maximum amplitude" and "total activity" while the speed of the response ("time to ½ max activity") was not affected (Fig. EV4B). Varying the MKK6 concentration affected the "time to ½ max activity", as the max amplitude was reached earlier with increasing concentration; the "maximum amplitude" and "total activity", although the increase in these with increasing MKK6 concentration was not as strong as with p38 and leveled off towards higher concentrations; and the first phase of p38 activity became more pronounced with increasing MKK6 concentration (Fig. EV4B). In contrast, increasing the MKK4 concentration mainly affected the second phase of p38 activity, which became more pronounced and resulted in a modest increase in "maximum amplitude" and an increase "total activity" due to a slower decline of activity. A decrease in MKK4 concentration affected the maximum amplitude only slightly, as the less affected first phase of activity defines the 'maximum amplitude' in those parameter regimes. "Time to ½ max activity" was delayed only by strong increases of MKK4 activity as these caused the second, rather than the first phase of activity to represent the maximum amplitude (Fig. EV4B). Increasing MKK4 and MKK6 at the same time resulted in p38 trajectories increased in amplitude and with slow decline with the first and second phase of activity strengthened proportionally. Increasing p38 concentration in combination with MKK4, MKK6, or both strongly multiplied the overall amplitude and correspondingly "total activity" of the response trajectories while maintaining the shape of the response determined by the other two kinase concentrations (Fig. EV4B). A

similar pattern is observed in the simulation of heterogenous single cell p38 trajectories when the means of the sampling distributions of kinase concentrations are changed (Fig. EV4B). Overall, this parameter sensitivity analysis supports the idea that the MKK6 and MKK4 pathway branches control the first and second phase of p38 activity, while p38 concentration has powerful control over the overall amplitude of response.

Examining the heterogeneous responses of NFκB and p38 in detail, we found that the molecular network heterogeneity introduced by distributing a few parameters resulted in weak correlations between p38 and NFκB activity dynamics in the same simulated cells. Determination of Spearman correlation coefficients for selected dynamic features confirmed this impression: In an excellent match with experimental results, simulated p38 and NFκB activity are moderately positively correlated for "time to ½ maximum activity" (correlation coefficient (CC): 0.63) and not correlated for 'maximum amplitude' (CC: 0.11) (Fig. 6F). Accordingly, half-hour integrals of simulated activity were moderately positively correlated in the first half-hour (CC: 0.51), but not correlated for the following two half-hour intervals (CC: 0.11, 0.08) (Fig. 6F). Our modeling results suggest that the experimentally observed poor correlation of single cell NFκB and p38 activities is, in fact, an inherent feature of the branched pathway architecture. Interestingly, "total activity" and "30-min activity integrals" after 1.5 h, which were weakly negatively correlated in experimental results, were moderately positively correlated in the simulated data (Fig. 6F). This suggests the existence of an unknown negative cross-regulation mechanism between NFκB and p38 that is not represented in the mathematical model.

To locate the sources of molecular heterogeneity that result in poor correlation between p38 and NFκB activity, we removed the noise from the p38 and/or NFκB module by using constant instead of sampled parameter values and calculated the correlations between p38 and NFκB dynamic features. Denoising the p38 module had little effect on the correlations (Figs. 6G and EV4C). In contrast, denoising the NFκB module resulted in a strong increase in correlations of later dynamic features ("maximum amplitude" and "total activity", CCs: 0.80, 0.95) and the 30-min activity integrals from 0 to 1.5 h (CCs: 0.84–0.87), while the correlation of "time to ½ max activity" was not affected (Figs. 6G and EV4C).

Denoising both modules resulted in a further increase in the correlation coefficients, except "time to ½ max activity", which remained unaffected (Figs. 6G and EV4C). This suggests that the lack of strong correlations between the heterogeneous NFκB and p38 dynamics may result mainly from molecular heterogeneity in the IκB-NFκB signaling module.

Analyzing the heterogeneity, measured as coefficient of variation (CV, a mean-normalized measure of variability), in the 30-min activity integrals between 0 and 1.5 h of simulated LPS-responsive p38 and NFκB activity shows CVs below 1, with p38 having higher CVs than NFκB (Fig. 6H). In experimental results, these integrals had CVs below 1 which were similar for p38 and NFκB (Fig. EV3G). We next analyzed the effect of parameter denoising on the heterogeneity. Denoising either p38 or NFκB modules reduced the range of NFκB and p38 activity values as expected (Fig. EV4C), but had only small effects on the CVs (Fig. 6H). Denoising the p38 module resulted only in a small decrease of the CVs of p38 activity integrals, while denoising the NFκB module resulted in small decreases of the CVs of early NFκB activity integrals, but an increase in the 1–1.5 h integral. Similarly, the CV of dynamic features such as "time to ½ max activity", "maximum amplitude", and "total activity" were higher for p38 and only slightly affected by the denoising of either module (Fig. 6I). This suggests that heterogeneity in p38 and NFκB dynamic features can result from the parameter distributions in the shared upstream modules, while their lack of correlation is largely due to parameter distributions in the IκB-NFκB signaling module.

## Uncorrelated heterogeneity of MAPK p38 and NFκB signaling may result in bimodality in the expression of AND gate target genes

Signaling dynamics of both NFκB and p38 showed poorly correlated single-cell heterogeneity (Fig. 5). We sought to explore the functional consequences for the expression of cytokines, which are known as NFκB and p38 targets. Based on the weak correlations between p38 and NFκB dynamic features, we hypothesized that p38, while contributing only little to the stimulus-specificity of the response, contributes heterogeneity to macrophage gene expression responses. Therefore, genes controlled by p38 and NFκB were hypothesized to have more variability in single-cell mRNA levels than those controlled by NFκB only.

To test the plausibility of this hypothesis, we developed a mathematical model of stimulus-induced, NFκB&p38 (AND gate) or NFκB-only controlled gene expression, with NFκB activity promoting the mRNA transcription rate and p38 activity inhibiting the mRNA-degradation rate (representative of p38's role in inhibition of mRNA-degradation promoting protein TTP (O'Neil et al, 2018)) (Figs. 7A and EV5A; Table EV3). Using the experimentally determined p38 and NFκB activities induced by high-dose stimulation with the four stimuli as inputs, we simulated the mRNA concentration over 8 h for a gene controlled by NFκB-only or by NFκB&p38 (Figs. 7B and EV5B). NFκB&p38-controlled gene expression was stronger than NFκB-only from 1 h onwards for all four stimuli and had a larger range of mRNA values at many of the timepoints (Figs. 7B and EV5B). Interestingly, the distributions of mRNA concentrations appeared more bimodal for NFκB&p38-controlled gene expression in some cases, e.g., P3C4-induced gene expression at 1 h, CpG- at 3 h, and TNF at 1 h (Fig. 7B). Thus, the

mathematical gene expression model predicted higher variability, and potentially bimodality, of NFκB&p38-controlled genes than of genes controlled by NFκB only.

To test this prediction, we determined mRNA levels in single cells stimulated with P3C4, CpG, LPS, or TNF for up to 8 h by targeted scRNAseq for a panel of immune response genes that include several cytokines. The cytokine genes *Il1a* and *Il1b*, which are categorized to be controlled by p38 and NFκB (see Materials & Methods), showed stimulus-specific expression time courses with a wide spread of expression in single cells ranging from undetectable to maximally detectable expression (Fig. 7C). In contrast, two genes controlled by NFκB only, *Nfkbia* and *Nfkbie*, had much narrower distributions of expression levels in single cells. Across stimuli, the Fano factor, a mean-normalized measure of the variance of a distribution, for *Il1a* and *Il1b* expression was consistently higher than 1 at all timepoints, while the Fano factor for *Nfkbia* and *Nfkbie* was lower or around 1 at timepoints 1, 3, and 8 h (Fig. 7D, left). The bimodality coefficient, a metric of the degree of bimodality of a distribution, was near 0.5 or higher for *Il1a* and *Il1b* expression at the 1, and 3 h timepoint for all four stimuli, but consistently below 0.5 at those timepoints for *Nfkbia* and *Nfkbie* (Fig. 7D, right).

Considering all genes in the NFκB&p38 *vs.* NFκB-only gene regulatory strategy (GRS) (categorized through literature-based curation of previously published gene assignments obtained through knockout cell lines and quantitative modeling (Cheng et al, 2017; Tong et al, 2016; Sen et al, 2020; Wang et al, 2021; Sheu et al, 2023), Table EV4), the Fano factors of the single cell gene expression at 8 h were higher on average in the NFκB&p38 GRS for P3C4 (NFκB: 1.27; NFκB&p38: 1.59, $p = 0.011$), CpG (NFκB: 1.31; NFκB&p38: 1.58, $p = 0.033$), and LPS (NFκB: 1.33; NFκB&p38: 1.59, $p = 0.045$), with Fano factors of gene expression induced by TNF, which induces lower p38 activity, having a smaller and statistically not significant difference in means (NFκB: 1.26; NFκB&p38: 1.39, $p = 0.23$) between the two GRS (Fig. 7E). Fano Factors of gene expression distributions were not statistically significantly different at 1 h (Fig. 7E). Genes in the NFκB&p38 GRS had higher bimodality coefficients at 1 h of stimulation with P3C4 ($p = 0.031$), with CpG stimulation resulting in a difference in bimodality coefficients at 1 h that was close to statistical significance ($p = 0.058$), but not at 8 h (Fig. EV5C).

Categorizing genes by the presence or absence of ARE elements in their mRNA, which are binding sites for the p38-controlled mRNA stability regulator TTP (O'Neil et al, 2018), resulted in higher bimodality coefficients for ARE-containing genes than for genes not containing ARE at 1 h, with this difference being statistically significant only in CpG stimulation ($p = 0.015$) (Fig. EV5D). No statistically significant differences in Fano Factors were detected in genes with and without ARE elements at 1 or 8 h (Fig. EV5E). Focusing on genes that contain ARE elements and have been reported to be p38 regulated, revealed clear differences in bimodality coefficient at 1 h for P3C4 ($p = 0.035$), CpG ($p = 0.0026$), and TNF ($p = 0.057$) stimulations, but not for LPS ($p = 0.43$) (Fig. 7F, top). No such differences were observed for NFκB-only controlled, ARE-containing genes (Fig. 7F, bottom). Across stimuli, the single cell 30-min integrated signaling activities of both p38 and NFκB over 4 h consistently had bimodality coefficients of below 0.5, with p38 and NFκB activities generally having very similarly low degrees of bimodality (Fig. EV5F). This suggests that non-bimodal p38 and NFκB signaling inputs combine to increase the bimodality of NFκB&p38 AND gate-controlled gene expression.

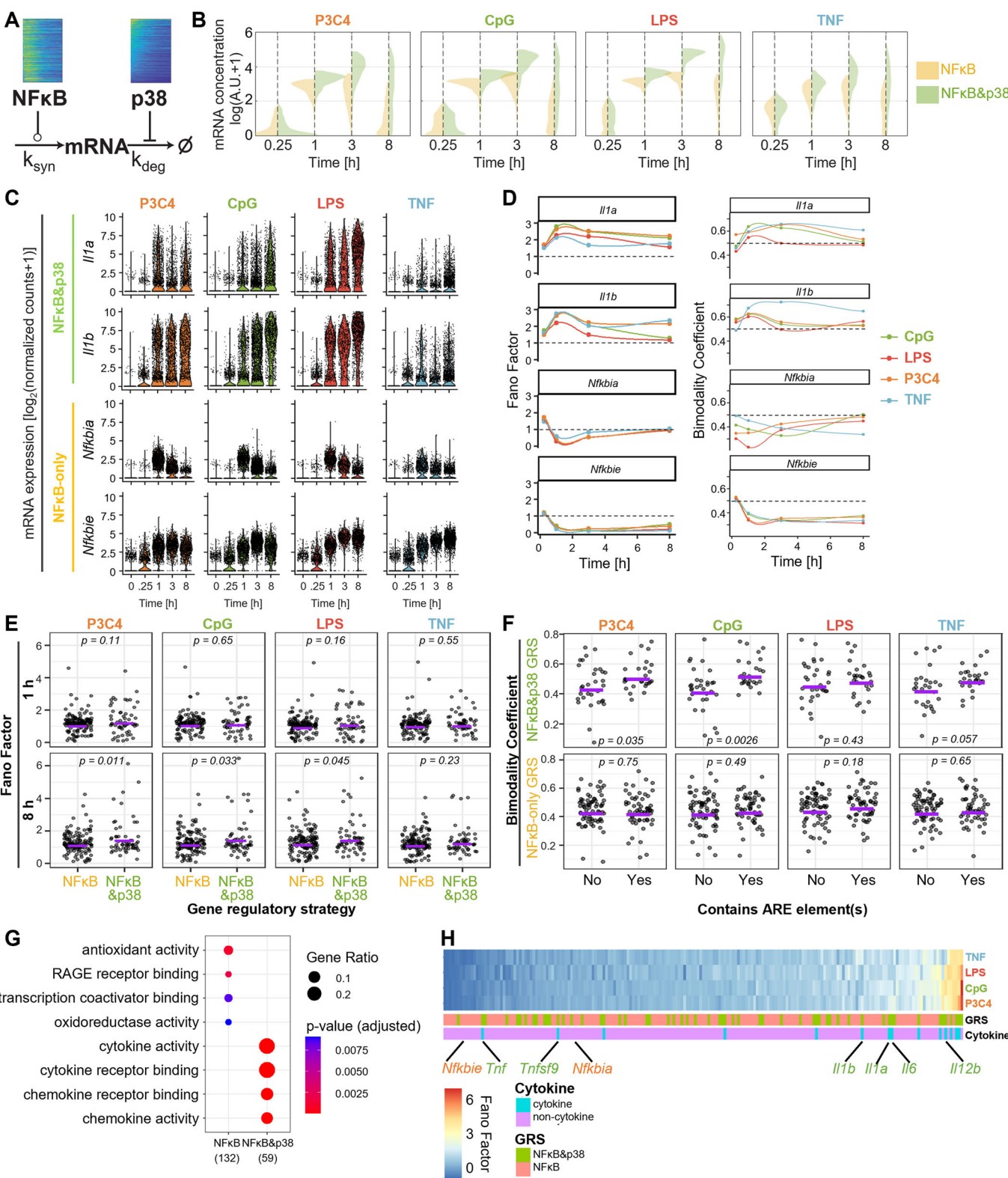

◀ **Figure 7. NFκB&p38 AND gate control of immune response genes increases heterogeneity and bimodality of expression.**

(**A**) Schematic of a mathematical model for simulating heterogeneous single cell gene expression controlled by NFκB or NFκB&p38 using experimental NFκB and p38 activity dynamics over 8 h as input and mRNA abundance over time as output; with NFκB enhancing transcription and p38 decreasing degradation of mRNA. (**B**) Distributions of simulated single-cell mRNA expression over 8 h when controlled by NFκB only or by NFκB&p38 using the model described in Panel (**A**). (**C**) Single cell gene expression distributions ($\log_2$(normalized counts+1)) of known NFκB&p38 or NFκB-only target genes in hMDPMs upon stimulation with 100 ng/ml P3C4, 100 nM CpG, 100 ng/ml LPS, or 10 ng/ml TNF over 8 h, measured by BD Rhapsody. Data from one experiment are depicted. Number of cells analyzed at each timepoint: Unstimulated: 1415 cells; P3C4: 835, 1840, 1251, 1153 cells; CpG: 992, 1229, 2235, 1236 cells; LPS: 941, 1045, 663, 1338 cells; TNF: 980, 776, 994, 2412 cells. (**D**) Fano factor (left, variance/mean) and bimodality coefficient (right) of single cell expression of indicated genes over 8 h (as in **C**). The dashed lines indicate a Fano factor of 1 (left) and a bimodality coefficient of 0.5 (right). (**E**) Fano factor of all immune response genes regulated by NFκB-only (132 genes) or NFκB&p38 (59 genes) upon stimulation with P3C4, CpG, LPS, or TNF at 1 h and 8 h. Purple line indicates the mean of Fano factors. Statistical significance was determined using a permutation test for differences in means using 10,000 permutations. (**F**) Bimodality coefficient of all immune response genes regulated by NFκB&p38 (top, 59 genes total) or NFκB-only (bottom, 132 genes total) categorized by absence or presence of ARE-element(s) upon stimulation with P3C4, CpG, LPS, or TNF at 1 h. The purple line indicates the mean. Statistical significance was determined using a permutation test for difference in means using 10000 permutations. (**G**) Gene ontology results of NFκB-only or NFκB&p38 regulated genes. Benjamini–Hochberg adjustment was applied to $p$ values. (**H**) Heatmap of Fano factors of single cell expression of NFκB-only or NFκB&p38 regulated genes upon stimulation with P3C4, CpG, LPS, or TNF at 8 h, sorted by average Fano factor across the four stimuli. Cytokine vs. non-cytokine identity and NFκB-only vs NFκB&p38 regulation are indicated by a color map. Positions of selected genes of interest are labeled. GRS gene regulatory strategy.

Gene ontology analysis of the NFκB&p38 target genes revealed enrichment for terms related to cytokine and chemokine activity and receptor binding, indicative of the fact that many inflammatory cytokines are NFκB&p38-dependent genes (Fig. 7G). Sorting the gene expression results for all four ligands and both GRS by average Fano factor revealed that cytokine genes tended to have higher Fano factors and were usually controlled by the NFκB&p38 GRS, rather than NFκB only (Fig. 7H). In summary, NFκB&p38-dependent gene expression, including cytokine genes, shows higher bimodality (mainly at earlier timepoints) and heterogeneity (mainly at later timepoints) than NFκB-only dependent gene expression.

## Discussion

Here we report how the combined action of MAPK p38 and NFκB signaling dynamics in the same single cells determine stimulus-specificity and target gene expression responses. The studies were enabled by an experimental system that allowed us to endow primary macrophage cells with fluorescent reporters for both pathways for live cell microscopy (Fig. 1). Specifically, we examined the often quoted hypothesis that a combinatorial code of NFκB and MAPK signaling ensures the stimulus-specificity of immune responses. Using information-theoretic analysis and machine learning classification, we showed that p38 dynamics convey some ligand- and dose-specific information (Figs. 2, 4), but actually contribute little to the information-rich dynamics of NFκB signaling for distinguishing ligand identity (Fig. 3) or dose (Fig. 4). We found that the heterogeneous dynamic features between the two pathways are poorly correlated in single cells (Fig. 5), a characteristic derived from molecular noise in the branched pathway architecture and the IκB-NFκB signaling module (Fig. 6). Thus, a mathematical model of p38- and NFκB-controlled gene expression suggested a different role of MAPK p38 in innate immune responses: to contribute variability to gene expression responses. We confirmed this hypothesis with scRNAseq, observing higher cell-cell heterogeneity and bimodality in mRNA expression of p38&NFκB-dependent genes than NFκB-responsive genes (Fig. 7). Together, this study suggests that one biological function of p38&NFκB combinatorial control of gene expression is to amplify signaling noise, resulting in heterogeneous gene expression responses, and thus limit the number of cells expressing the target cytokine genes.

This study demonstrates the importance of multiplexing live-cell reporters to understand combinatorial and dynamical signaling in stimulus responses of single cells. Here, we combined the endogenous mVenus-RelA knock-in reporter (Adelaja et al, 2021) with a non-effector kinase translocation reporter for MAPK p38 (Regot et al, 2014) that was ectopically expressed. To ensure that our results are reflective of true macrophage functions, we avoided the use of cell lines that are commonly used in live cell microscopy studies. Instead, we turned to myeloid precursor cells harvested from bone marrow whose lifespan was extended with HoxB4, but that may reconstitute a complete hematopoietic system upon transplantation (Ruedl et al, 2008). Here we differentiated them into macrophages whose stimulus-response behavior was largely indistinguishable from replicate cultures of traditional bone marrow-derived macrophages (Singh et al, 2024). The extended lifespan of precursors allowed us to introduce the KTR via a lentiviral vector that is resistant to epigenomic silencing during differentiation due to the presence of a ubiquitous chromatin opening element (UCOE) (Müller-Kuller et al, 2015). We anticipate that this experimental system will find applicability in studies of other signaling pathways regulating macrophage responses, be they MAPK or IRF family members.

We found that p38 activity dynamics alone contain information regarding the molecular identity of the stimulus, allowing for an accurate distinction of PAMP-induced signaling from signaling induced by the host cytokine TNF (Fig. 2). Distinguishing the three different PAMPs from each other is less accurate based on p38 dynamics alone (Fig. 3). This is not surprising as all three PAMPs use the same signal transducer MyD88, while TNF activates p38 via TRAFs and associated ubiquitin chains (Luecke et al, 2021). Further, p38 dynamics do not perform well in finely graded dose distinctions, but do distinguish low from high doses in binary classifications that leverage the thresholded dose response curves (Fig. 4) (Gottschalk et al, 2016) and may be relevant for distinguishing harmful vs harmless PAMP exposure. We can think of two reason for the high-dose threshold: (1) for PAMP-induced p38 activation, Myddosome oligomerization generates a non-linear dose response (Cheng et al, 2015) and (2) the MAPK cascade itself has a thresholded dose response curve (Huang and Ferrell, 1996) that is regulated by phosphatase activity levels (Altan-Bonnet and Germain, 2005). In response to TNF we observed a multi-phase activity, but not oscillations previously reported in IL1β-stimulated

HeLa cells (Tomida et al, 2015). This multi-phase activity is unlikely the result of a delayed negative feedback loop but may result from the branched pathways (MKK3/6 and MKK4) activating p38 rapidly and delayed-and-sustained (Pattison et al, 2016) (Fig. 6).

We found that NFκB dynamics generally contain more stimulus-specific information than p38 dynamics (Fig. 3). Information-rich NFκB dynamics have been noted previously (Adelaja et al, 2021; Selimkhanov et al, 2014) and are encoded by a variety of mechanisms, including several nested negative feedback loops (Basak et al, 2012). Further, when p38 dynamics are combined with NFκB dynamics, there is only little improvement in ligand distinction (Fig. 3). Similarly, p38 activity dynamics show a poorer distinction of stimulus doses than NFκB, and combining p38 and NFκB dynamic features does not increase the dose information available in rich NFκB dynamics (Fig. 4). Thus, the redundancy-synergy index (Timme et al, 2014) indicates that there is redundancy between p38 and NFκB with regards to the mutual information of their combined dynamics. However, when only considering the "total activity" of each signaling activity, p38 does contribute more substantially to the NFκB-mediated dose classification of all four tested stimuli tested (Fig. 4).

What these carefully documented quantifications with information-theoretic and machine learning classification algorithms suggest is that immune response genes gain stimulus-specificity if they contain gene regulatory mechanisms that are capable of decoding the dynamics of signaling—high specificity from decoding NFκB dynamics, moderate specificity from decoding p38 dynamics—or if they contain gene regulatory mechanisms that are capable of decoding NFκB&p38 combinatorics, but they gain little more from being able to decode both dynamics and combinatorics. The combinatorial and dynamical signaling codes are not, in principle, mutually exclusive, but we find here that they are redundant. This suggests that decoding mechanisms that are sensitive to both are less likely to be found. Indeed, the sequential AND gate control previously described (Cheng et al, 2017) and used in our gene expression simulations (Fig. 7) is largely insensitive to intricate dynamics.

While p38's function in immune response genes is tied to NFκB, its biological functions are, of course, broader with numerous other substrates, including histone tails, regulators of protein trafficking and secretion, as well as cell growth and cell cycle and beyond (Caldwell et al, 2014; Mahtani et al, 2001; Luecke et al, 2021; Andersson and Sundler, 2006; Xu and Derynck, 2010; Scott et al, 2011). Additionally, regulation of p38 signaling may involve aspects that are not fully captured by the whole-cell p38-KTR measurements, such as intricate context-specific modulation of subcellular localization (Maik-Rachline et al, 2020). Similar considerations may apply to MAPK JNK which was reported to allow macrophages to distinguish between different threat levels of pathogen exposure (Lane et al, 2019).

But how is it possible that p38 KTR and NFκB signaling trajectories appear to be so redundant when ascertained by mutual information and machine learning classification? These are population-level metrics, which therefore are affected by the poor correlation between heterogeneous p38 and NFκB dynamic features within single cells (Fig. 5). While very early features, such as speed of activation and early integrated activity show some correlation, later features, such as amplitude of activation, do not. Mathematical

modeling suggested that this lack of strong correlations to be an inherent feature of the branched pathways that can result from variable expression of molecular components in the signaling modules downstream of pathway branching (Fig. 6). In this model, noise in the NFκB module, in which the IκBα negative feedback loop provides highly non-linear transformation of signaling activity, had the largest effect on the correlations. Commonly assumed distribution widths for these expression values were sufficient to recapitulate the rapid decline in correlations within the time course and removing the noise in the NFκB module restored strong correlations between p38 and NFκB activities.

In addition, the experimental data showed that late features such as duration or integral were slightly anti-correlated between the two pathways. This suggests mechanisms of negative cross-regulation between the pathways, including limited availability of molecular mechanisms required by both pathways, resulting in competition, or more active mechanisms such as an NFκB target gene that inhibits sustained MAPK p38 activity, or a p38 target that negatively regulates sustained NFκB signaling. For example, it is known that p38 phosphorylates TAB1, causing an inhibitory effect on TAK1 (Cheung et al, 2003), which could diminish NFκB signaling. Alternatively, the p105/Tpl2 node could mediate negative cross-regulation, as p105 binds and inhibits Tpl2 activity (Beinke et al, 2003; Waterfield et al, 2003), and NFκB activity induces p105 expression. Our results should prompt further studies to characterize these cross-regulatory mechanisms quantitatively.

The uncorrelated heterogeneity of NFκB and p38 signaling suggests a new biological role of AND gate gene expression control. Since all signaling pathways inherently have some degree of noise, it makes intuitive sense that when placing a process, such as gene expression, under the control of multiple pathways, as occurs in combinatorial signaling, the heterogeneity of the outcome may increase. Using mathematical models of NFκB- and NFκB&p38-controlled genes, we further find not only higher variability but also higher bimodality in mRNA expression as a result of AND gate control (Fig. 7). Indeed, we show with single cell RNAseq data that NFκB&p38 AND gate target genes have a greater Fano factor, a measure of dispersion, and a greater bimodality coefficient than genes regulated by NFκB alone (Fig. 7). For NFκB&p38 target genes, there was an especially significant difference in bimodality coefficient when comparing genes with and without ARE elements, which mediate p38's mRNA stabilization effect. In sum, while prior bulk assays of the population and theoretical modeling of such scenarios emphasized the increased stimulus-specificity of AND gate genes (Cheng et al, 2017; Buchler et al, 2003; Ten et al, 1992) and the role of AND gates in threshold regulation (Nguyen et al, 2015), the results presented here suggest that such stimulus-specific expression may be driven by only a few cells in the population. Based on these results, we suggest the new hypothesis, that AND gate control of gene expression may have evolved for those genes that must be expressed only in a minority of cells.

Why might the organism benefit from particular genes being expressed noisily? We note that many NFκB&p38 AND gate genes encode pro-inflammatory cytokines. Furthermore, prior micro-ELISA assays of single cells revealed a remarkable cell heterogeneity of TNF secretion that showed little correlation with NFκB activity (Junkin et al, 2016). This single-cell heterogeneity of cytokine expression has also been observed in PBMCs on mRNA and protein level (Pal et al, 2020), and even in clonal T cell population (Bucy

et al, 1994). It appears to be a hallmark of immune stimulus-induced gene expression responses that may have the physiological function of limiting population-level cytokine secretion to avoid harmful cytokine storms (Talemi and Höfer, 2018). Future studies may address how altered MAPK AND NFκB combinatorial control of cytokine expression due to environmental exposure history or genetic predisposition contributes to inflammatory disease.

# Methods

## CBX3-p38-KTR-mCerulean3 construct generation

p38-KTR-mCerulean3 insert was amplified by Q5 polymerase PCR from pENTR-p38-KTR-mCerulean3 (Addgene #59149) (Regot et al, 2014) using primers containing restriction sites for AgeI and SalI (Fwd: 5'-gatccaccggtcgccaccATGCGTAAGCCAGATCTC, Rev: 5'-gttgattgtcgacgcggccgctttacttgtacagctcgtccatgc). After PCR clean up using Qiagen QuickExtract kit, the insert and the vector backbone (CBX3-SEW GFP), a lentiviral vector for silencing-resistant transgene expression (Müller-Kuller et al, 2015), were restriction digested using AgeI and SalI, which excised the GFP from the vector backbone. After gel purification, the insert was ligated into the backbone using T4 ligase and replicated by transformation of Stbl3 *E. coli* (Invitrogen, # C737303) and plasmid purification. Correct insertion and absence of mutations in insert were verified by sequencing.

## Generation of dual reporter macrophages

Mice were housed and handled according to guidelines established by the UCLA Animal Research Committee under an approved protocol. UCLA's current Animal Welfare Assurance identification number is D16-00124 (A3196-01). HoxB4-transduced myleoid precursors (hMPs) were generated from male mVenus-RelA (*Rela*$^{V/V}$) C57BL/6 J mice (Adelaja et al, 2021) as described (Singh et al, 2024). hMPs were maintained in IMDM media (Gibco #12440061) supplemented with 10% ES cell FBS (Gibco #10439024), 100 IU Penicillin, 100 μg/ml Streptomycin, 2 mM L-Glutamine, 55 μM 2-mercaptoethanol, 10 ng/ml IL-6 (Peprotech, #216-16), and 100 ng/ml m-SCF (Peprotech, #250-03). Cells were cultured at 37 °C in a humidified atmosphere with 5% $CO_2$. mCerulean3-p38-KTR *Rela*$^{V/V}$ hMPs were generated by lentiviral transduction of *Rela*$^{V/V}$ hMPs with CBX3-p38-KTR-mCerulean3 as follows: HEK cells were PEI-transfected with CBX3-p38-KTR-mCerulean3, pMDL, pRSV, and pVSV (third generation lentivirus). Media was changed 24 h after transfection and lentivirus-containing supernatant was collected 48 h after transfection. Supernatant was cleared by centrifugation at $300 \times g$ for 5 min and filtering through 0.45 μm filter. Virus was then concentrated using Amicon filter (100 μm pore size). *Rela*$^{V/V}$ hMPs were transduced with concentrated virus with polybrene (10 μg/ml). Media was changed 24 h after transduction. After passaging, cells that were mVenus$^+$ (100% of single cell gate) and mCerulean$^+$ (17% of single cell gate) were collected by fluorescence-activated cell sorting, expanded, and frozen for future culture. HoxB4-transduced myleoid precursor-derived macrophages (hMPDMs) were differentiated from $0.5 \times 10^6$ hMPs in 10 ml of IMDM media (Gibco 12440061) supplemented with 10% ES cell FBS (Gibco 10439024), 100 IU Penicillin, 100 μg/ml Streptomycin, 2 mM L-glutamine, 55 μM 2-mercaptoethanol, and 30%

L929-cell supernatant in 15-cm TC-treated dishes for 10 or 11 days, with media replenishment on Day 3 and replating into experimental dishes on Day 7.

## Cell stimulation

On Day 7 of differentiation, cells were detached using a cell lifter and replated into experimental culture dishes. For determination of protein levels in by Western Blotting or flow cytometry, cells were replated at a density of 20000–22000/cm$^2$ in a six-well format. For live cell microscopy, cells were plated into eight-well μ-slides (Ibidi #80826) at 20000–26000 cells/cm$^2$. For single-cell gene expression measurements, cells were plated into 6-cm plates at a density of 20000/cm$^2$. Cells were stimulated on Day 10 or 11 at the indicated concentrations with recombinant mouse TNF (aa 80–235) protein (410-MT-010, R&D Systems), Pam3CSK4 (tlrl-pms, Invivogen), CpG ODN 1668 (tlrl-1668, Invivogen), LPS (L6529-1MG, Sigma-Aldrich), or media only control for the indicated times. Inhibition of p38 was performed by incubation of cell culture with SB203580 (S-3400, LC Laboratories) for 1 h before stimulation. In cell stimulation experiments, a numbering code for sample tubes was used such that samples could be processed with decreased awareness of their identities.

## Western blotting

For determination of protein levels in whole-cell lysates, cells were stimulated as indicated and placed on ice at harvest timepoints, washed in the plate with cold PBS, lysed directly with β-mercaptoethanol-containing Laemmli sample buffer (Bio-Rad). Lysates were incubated at 95 °C for 5 min. Proteins were separated by SDS–PAGE (Criterion TGX 4–5% gel, Bio-Rad) and transferred to PVDF membranes using wet transfer. Blocking of membranes was carried out in 5% bovine serum albumin (BSA) for 1 h at room temperature (RT), followed by incubation with primary antibodies at 4 °C for 16 h. Membranes were incubated with appropriate HRP-conjugated secondary antibodies (Cell Signaling Technology) for 1 h at RT and developed using Super Signal West Femto and Super Signal West Pico Plus chemiluminescent substrate (Thermo Scientific). The following primary antibodies were used: α-phospho-p38 (Thr180/Tyr182) (CST4511, 1:3000), α-phospho-MK2 (Thr334) (CST3007, 1:1000), α-phospho-CREB (Ser133) (CST9198, 1:2000), α-IκBα (SC371, 1:2000), and α-tubulin (T5201, Sigma-Aldrich, 1:5000). Protein levels were quantified by measuring mean gray value using ImageJ, deducting background value per lane, and dividing by background-corrected β-tubulin bands. When applicable, for comparisons of samples on separate membranes, protein levels were normalized to internal control stimulation samples included on every membrane.

## Flow cytometry

To quantify intracellular p-p38 levels by flow cytometry, hMPDMs were stimulated as indicated. At harvest timepoint, cells were placed on ice, washed in the plate with cold PBS, fixed in 1.6% paraformaldehyde for 10 min at RT, washed with FACS buffer (2% FBS in PBS with 2 mM sodium orthovanadate), lifted, and placed in a 96-well conical plate. After two more washes, cells were permeabilized in ice-cold methanol (100%) for 1 h on ice and

washed twice. After Fc-blocking for 10 at RT min using α-CD16/32 antibody (BioLegend 101301, 1:100), samples were incubated with Alexa647-α-p-p38 (BD 612595, 1:5) for 1 h at RT. Flow cytometry was performed on CytoFLEX flow cytometer (Beckman Coulter). Data were analyzed using FlowJo v10. To determine the fraction of cells with activated p38, a cutoff between p-p38 negative and p-p38 positive cells was determined using the fluorescence signal in an unstained, unstimulated sample (to account for MAPK activation at baseline) and applied to all samples.

## Live-cell fluorescence microscopy

For measurement of signaling dynamics, hMPDMs were subjected to live cell imaging using a previously established workflow (Adelaja et al, 2021). Before imaging, 0.1 μM SiR-DNA dye with 5 μM Verapamil (SC007, Spirochrome) was added to the cells for nuclear staining. Cells were placed on an Axio Observer.Z1 inverted microscope (Zeiss) with live-cell incubation for at least 60 min to equilibrate to environmental conditions. Using automated image acquisition in Zen 2.3 software, at least 5 positions per condition (maximum eight conditions per experiment) were imaged at 1 frame per 5 min using a Plan-Apochromat 20x/0.8 NA M27 air objective. Images were collected sequentially in three channels, for mCerulean3 fluorescence (filter set: Zeiss 47 HE, ex.: 436/25 nm, beam splitter: 455 nm, em.: 480/40 nm; Colibri.2 445 nm, 90% power; exposure: 300 ms), mVenus fluorescence (filter set: Zeiss 46 HE, ex.: 500/25 nm, beam splitter: 515 nm, em.: 535/30 nm; Colibri.2 505 nm, 90% power; exposure: 160 ms), for SiR-DNA fluorescence (filter set: Chroma CY5NX, ex.: 640/30 nm, beam splitter: 660 nm, em.: 690/50 nm; Colibri.2 590 nm, 100% power; exposure: 600 ms), and for differential interference contrast (DIC) (HAL 100 lamp; 2.5 V, exposure: 100 ms). Images were recorded on a Hamamatsu Orca Flash4.0 CMOS camera with $2 \times 2$ binning. The number of cells imaged was determined by the cell density and the total number of positions that could be imaged within the 5 min framerate. After the collection of baseline images for 0.5–1 h, the indicated stimulus diluted in conditioned media was applied using syringe injection into the chamber in situ and images were acquired for an additional 8 h.

## Automated image analysis and feature calculations

Individual channel images were exported in Zen lite 3.3 software. Single cells were segmented into cytoplasm and nucleus and tracked across the time course. SiR-DNA fluorescence was used to segment nuclei. mVenus-RelA median nuclear fluorescence, as well as p38-KTR-mCerulean nuclear and cytosolic median fluorescence, using an annulus around the nucleus to represent the cytoplasm, were quantified using MACKtrack in MATLAB 2015a. MACKtrack was first published here: (Adelaja et al, 2021) and updated for the described studies.

To represent NFκB activity trajectories, background-corrected, baseline-deducted median nuclear mVenus fluorescence was used. MAPK p38 trajectories were represented by a baseline-deducted ratio of median cytosolic/nuclear fluorescence. Trajectories were further corrected for systemic variations in fluorescence over the microscopy time course using a correction trajectory generated using eight mock trajectories. Trajectories underwent stringent quality filtering: cells were filtered out if they did not exist in the field of view at the experiment start, if they did not exist in the field of view for the duration of the time course (app. 8 h), if their area was very small, if their nuclear staining was extremely high, if nuclear mVenus fluorescence was very dim, if KTR expression was extremely low or high, and if they had extreme NFκB or p38-KTR activities. For details, see MACKtrack function *filter_nfkb_ktr_ratio*. Results of the quality filter for all experimental conditions are available here: Table EV1. Dynamic features were extracted from single cell activity trajectories MACKtrack function *nfkb_ktr_ratio_metrics* using MATLAB 2022a. A description of dynamics features is available here: Table EV2.

## Mutual information calculation

Mutual information (MI) was calculated from p38, NFκB, or p38 and NFκB dynamic features from two pooled biological replicates from the indicated sets of stimulation conditions. Prior to calculation, dynamic features containing NaN values were removed, and the features were z-scored. MI was extrapolated using jackknife resampling to control for different sample sizes with 12 sample subsets, each containing 65–90% of the total samples (Adelaja et al, 2021). MI was calculated using a published method optimized for determining mutual information between discrete and continuous datasets (Ross, 2014).

## Neural network classification of signaling dynamics based on time series

Model implementation: Time-series data were classified using a recurrent neural network (RNN) with a Long Short-Term Memory (LSTM)-based machine learning classifier (Hochreiter and Schmidhuber, 1997), which was especially suited for use on time-series data without transformation into dynamic features due to its ability to learn long-term dependencies in input sequences by considering timepoints in sequence rather than as independent features (Van Houdt et al, 2020; Yu et al, 2019). The following architecture was employed: a stacked LSTM layer with two recurrent layers with a hidden dimension of 95 (equal to the number of timepoints), and variable input dimension (1 or 2), followed by two fully connected linear layers with a hidden dimension of 512 and the number of output classes respectively. ReLU nonlinearities were placed between all layers except the final output layer. All models were trained using the Adam optimizer with an initial learning rate of 1e-3 and beta values of 0.9 and 0.999, respectively, and a batch size of 64. During training, we optimized our objective using a weighted categorical cross-entropy loss to account for class imbalances. All models were trained for a maximum of 100 epochs, saving the best checkpoint for each respective model at its lowest validation loss to use for inference. Model logits were converted to class probabilities using the softmax operator and final classifications were determined through the argmax operator.

Training and evaluation: The input data were the NFκB, p38, or both trajectory data stimulated with the ligand and doses corresponding to each specific classification problem. Input data were transformed by a standard scaler (mean 0, variance 1), with scaling being applied individually across the different signaling inputs (p38 and NFκB), before re-concatenation for the two-dimensional case. Trajectories with missing (NaN) values were

removed. Performance was evaluated through a fivefold cross-validation, holding the number of cells per class consistent across all folds. The model performance metrics reported (F1 score, confusion matrix) combine the results across the fivefolds. Training and experiments were run using an NVIDIA 2070 RTX with an approximate training time of 10 min per fold. Model implementation and training were done through the PyTorch framework (1.12.1) (Paszke et al, 2019; Pedregosa et al, 2011).

## Decision tree classification of signaling dynamics based on dynamic features

For the classification of p38 and NFκB dynamic features, machine learning models consisting of ensembles of 30 decision trees were constructed according to similar considerations as described in (Adelaja et al, 2021) in Matlab 2022a. Bootstrap aggregation (bag) was used to construct the decision tree ensembles. Default parameters were used for model training parameters; "MaxNumSplits" was set to 20. As class labels, the molecular identity (P3C4, CpG, LPS, TNF) or dose (Dose 1/5–Dose 5/5) of the stimulus were used in accordance with the specific classification task. As predictors, either all 228 p38, all 228 NFκB, or the combination of all p38 and NFκB dynamic features (Table EV2) derived from two pooled biological replicates each of the indicated stimulation conditions were used. As controls, combinations of p38 and NFκB dynamic features, where either the NFκB or p38 cell assignments were randomly scrambled across all cells were used as predictors. The dynamic features were z-scored across all classes before each model training. In each model, equal cell numbers across all classes were used by a uniform random selection of cells to reach the cell number equal to the number of cells in the class with the fewest cells. 5-fold cross-validation was used to evaluate model performance. Confusion matrices, overall model accuracies, and by-class F1 scores (harmonic mean of precision and sensitivity) were used as performance metrics.

## Dose-response analysis using Hill equation fits

For analysis of dose responses, the fraction of cells with p38 and NFκB activity ('responder fraction') from two pooled biological replicates of stimulations with mock and five doses of indicated stimuli were plotted over the stimulus concentration. To be included on the logarithmic axis, mock stimulation was represented as a 100x lower concentration than the lowest stimulation condition.

Hill equation fits for the dose response analysis were generated using *fit* function in MATLAB 2022a using the following equation:

$$A = \frac{\max A * \mathrm{conc}^{\mathrm{HillC}}}{\mathrm{half}A\_\mathrm{conc}^{\mathrm{HillC}} + \mathrm{conc}^{\mathrm{HillC}}} + \mathrm{intersect}$$

where: *A* is the activity (represented by respective dynamic feature), conc is the stimulus concentration, max*A* is the maximum activity, half*A*_conc is the stimulus concentration at which half maximum activity is reached, Hill*C* is the Hill coefficient, setting [0.1,10] for *HillC* and [min(dose), max(dose)] for *halfA_conc* as lower and upper bounds, respectively, as well as [0,1] for *maxA* and [0,0.9] for *intersect* as lower and upper bounds.

## Calculations of Spearman correlations coefficients between p38 and NFκB features

Spearman correlation coefficients between corresponding p38 and NFκB features were calculated from two pooled replicates of high-dose ligand stimulations as indicated using *corrplot* ("Type": "Spearman") function in MATLAB R2022a.

## Mathematical modeling of single cell p38 and NFκB dynamics

Rescaling of experimental data to SI Units: We assumed a nuclear NFκB concentration range of 0.04 μM to 0.30 μM, in line with prior models (Basak et al, 2012). Specifically, we scaled the 90th percentile response to 100 ng/mL LPS (7.62 A.U.) to 0.30 μM nuclear NFκB. This resulted in an NFκB A.U. to S.I. scaling factor of 0.0383. For p38, we assumed the pp-p38 concentration to maximally reach 0.08 μM. The p38 activity is represented by the baseline-corrected KTR C/N ratio, which reaches a maximal value of 0.73 in this dataset (the 99th percentile of maximum amplitudes in 100 ng/ml P3C4-stimulated cells), resulting in a rescaling factor of 0.1096.

Modeling NFκB and p38 trajectories for a representative cell: To simulate NFκB and p38 activity in response to innate immune stimuli in the same cell, we expanded an established mathematical model of NFκB signaling downstream of P3C4, CpG, LPS, or TNF (Adelaja et al, 2021; Luecke et al, 2023) by incorporating a p38 module accounting for p38 regulation through two pathway branches, via IKK and TAK1 (Figs. 6A and EV4A). IKK, via p105 degradation and Tpl2 activation, instigates MKK3/6 phosphorylation, which we simplified into two steps (single and double phosphorylation) via a Hill equation. This equation factors in Tpl2's effects on the $v_\mathrm{max}$ and $K_\mathrm{m}$ parameters. TAK1's facilitation of MKK4 phosphorylation was similarly modeled, but with unique parameters accounting for molecular differences. MKK3/6 and MKK4 then phosphorylate p38 in two steps. While the parameters for the receptor and NFκB modules of the representative cell were derived from previous models (Adelaja et al, 2021), p38 parameters were ascertained by fitting our model to representative experimental cell responses to four stimuli (100 ng/ml P3C4, 1000 nM CpG, 100 ng/ml LPS, and 100 ng/ml TNF). Using the model for the representative cell, we derived the response curves of all species, including phosphorylated MKK6 and phosphorylated MKK4, to different stimuli (Pam3CSK, CpG, LPS, and TNF). The model species and reaction parameters can be found in Dataset EV1. The differential equations model was computed in MATLAB utilizing the ode15s function. The simulation process consisted of an initial phase to achieve a stable state, followed by a phase involving ligand stimulation, with outcomes visualized in MATLAB.

Simulation of heterogeneous single cell trajectories: To simulate the single-cell heterogeneous LPS-induced NFκB and p38 activities, we distributed ten parameters of the model. Within the p38 module, p38, MKK4, and MKK6 starting concentrations were sampled from a log-normal distribution centered around the "representative cell" parameters, ensuring 99% of parameters varied no more than twofold from the representative. TLR4 and NFκB module parameters were sampled from distributions determined from NFκB single cell data. For "denoising" simulations,

heterogeneous single cell trajectories were simulated using single parameter values (as for the 'representative cell' simulations) instead of sampled parameter distributions for the "denoised" modules. Model simulations were implemented in MATLAB 2020a using the ode15s solver.

## Mathematical modeling of heterogenous single cell AND gate-controlled gene expression

We employed a kinetic model for the gene regulatory network described by the following equation adapted from (Cheng et al, 2017):

$$\frac{d[mRNA]}{dt} = k_{syn,\max} \frac{(NF\kappa B)^{n_{nfkb}}}{\left(K_{d,NF\kappa B}\right)^{n_{nfkb}} + (NF\kappa B)^{n_{nfkb}}}$$
$$- k_{deg,\max} \frac{\left(K_{d,p38}\right)^{n_{p38}}}{\left(K_{d,p38}\right)^{n_{p38}} + (p38)^{n_{p38}}} [mRNA]$$

In this formula, NFκB enhances promoter activity, thereby increasing transcription rates, while p38 stabilizes mRNA, reducing its degradation. As model inputs, the experimentally determined heterogeneous single-cell dynamics of NFκB and p38 activity in response to the highest dose of LPS, TNF, P3C, CpG were used. They affect the rates of mRNA transcription (NFκB) and degradation (p38) over an 8-h-time course. The model outputs the dynamics of mRNA abundance over 8 h. To evaluate gene expression heterogeneity, we collected predicted single-cell mRNA abundances at 0.25, 1, 3, and 8 h. The parameters of the model are listed in Table EV3. Model simulations were implemented in MATLAB 2020a using the ode15s solver.

## Single cell gene expression by BD Rhapsody and analysis

### Sample preparation
For single-cell gene expression analysis, hMPDMs were stimulated as indicated and collected at 15 min, 1, 3, and 8 h post stimulation. Macrophages were sequenced using the BD Rhapsody scRNAseq platform as previously described (Sheu et al, 2023). Briefly, to collect adherent macrophages, cells were washed 1x with cold PBS, then lifted into suspension by incubating at 37 °C for 5 min with Accutase (Thermo Fisher Scientific, #A1110501), which resulted in cell viability of typically >85%. Cells were centrifuged at 4 °C, $400 \times g$ for 5 min, and resuspended in PBS + 2% FBS. Cells were hash-tagged with anti-CD45-hashtags (BD Rhapsody #633793) and loaded onto the cartridge (BD Rhapsody #633771). Libraries were prepared according to the manufacturer's instructions (BD Rhapsody #633771) and sequenced $2 \times 100$ on Novaseq 6000.

### scRNAseq data processing
Raw fastq files were processed using the BD Rhapsody™ Targeted Analysis Pipeline (version v1.0) (Shum et al, 2019) hosted on Seven Bridges Genomics. Distribution-Based Error Correction (DBEC)-adjusted UMI counts (molecules per cell) were used in the downstream analysis. Multiplets, cells with undetermined barcodes, and cells with less than 80 features were removed from the analysis. Due to the selected 500 gene panel comprised of largely inducible genes, the assumption that the total number of RNAs per cell is

constant does not hold. Counts were therefore normalized using the package ISnorm (Lin et al, 2020), and these normalized counts were plotted.

### Gene expression analysis
Fano factor for each gene was calculated by dividing the gene variance by its mean, for each stimulus condition or timepoint. Classification of genes into NFκB-only-regulated or NFκB&p38-regulated was based on literature-based refinement of previously published gene assignments obtained through knockout cell lines or quantitative modeling refined throughout multiple publications (Cheng et al, 2017; Tong et al, 2016; Sen et al, 2020; Wang et al, 2021; Sheu et al, 2023) (Table EV4). Genes were categorized as containing or not containing ARE elements using the 'AREScore' algorithm (Spasic et al, 2012). Bimodality coefficients of gene expression were calculated using the R function *bimodality_coefficient* according to (Pfister et al, 2013). Bimodality coefficients were calculated on all non-zero gene expression values for each gene. Bimodality coefficients of NFκB and p38 activity integrals were calculated using the function *bimodalitycoeff* in MATLAB 2022a (Zhivomirov, 2024). Gene ontology was performed using the R package *clusterProfiler* and searching for enrichment in the "Molecular Function" ontology database. Adjusted $p$ values were calculated via Benjamini–Hochberg correction. Gene ontology terms were collapsed if the genes they contained had more than 90% overlap, and the top four terms for either NFκB-only genes or NFκB&p38-genes were displayed.

# Data availability

The datasets and computer code produced in this study are available in the following databases:

- Code for MACKtrack for automated cell segmentation, tracking, quantification, and dynamic feature calculation:
  - Code optimized for this manuscript, including KTR C/N ratio quantitation: GitHub (github.com/signalingsystemslab/MACKtrack-for-NFkappaB-and-p38-dynamics).
  - Original code: GitHub (github.com/signalingsystemslab/MACKtrack).
- Experimental single-cell NFκB and p38-KTR trajectories and dynamic features: Zenodo (https://doi.org/10.5281/zenodo.8274567).
- Code for classification of p38 and NFκB signaling trajectories using a neural network: GitHub (github.com/signalingsystemslab/nfkbktr).
- Code for mutual information calculations, for classification of dynamic features of p38 and NFκB signaling using decision tree ensembles, for the dose response analysis, and for the feature correlation analysis: GitHub (github.com/signalingsystemslab/CombinatorialSignalingMacrophages_p38NFkB).
- Feature-based classification model outputs: Zenodo (https://doi.org/10.5281/zenodo.8274744).
- Code for mathematical modeling of single-cell p38 and NFκB trajectories and modeling of single-cell gene expression: GitHub (github.com/signalingsystemslab/p38_NFkb_single_cell_simulation).
- Input for mathematical modeling of single-cell p38 and NFκB trajectories and gene expression: Zenodo (https://doi.org/10.5281/zenodo.11518084).
- scRNA-seq datasets: Gene Expression Omnibus GSE224518 (ncbi.nlm.nih.gov/geo/query/acc.cgi?acc=GSE224518).

Other data are available upon request.

The source data of this paper are collected in the following database record: biostudies:S-SCDT-10_1038-S44320-024-00047-4.

## Peer review information

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

## Acknowledgements

We thank Stefan Stein (University of Frankfurt) for the CBX3-SEW plasmid, Supriya Sen for reagents and technical advice, and Sven Kaehler for the helpful discussion of dynamic features. Flow cytometry-assisted cell sorting was performed at the UCLA Janis V Giorgi Flow Cytometry Core Laboratory. Sequencing was performed at the UCLA Technology Center for Genomics and Bioinformatics Sequencing Core. Schematic figures were generated using components from Servier Medical Art, provided by Servier, licensed under a Creative Commons Attribution 3.0 unported license. SL was funded by the Deutsche Forschungsgemeinschaft (DFG, German Research Foundation) (419234150). AS and KS were funded by the UCLA Medical Scientist Training Program (T32-GM008042) and Systems and Integrative Biology Training Grant (T32-GM008185). SCL was funded by the UCLA Gastroenterology Training Grant (T32- DK007180). FL was funded by the Conselho Nacional de Desenvolvimento Científico e Tecnológico (Brazilian Council for Scientific and Technological Development) (200176/2022-6). The work was funded by National Institutes of Health (NIH) grants R01AI127864 (to AH) and R01AI173214 (to AH and RW).

## Author contributions

**Stefanie Luecke**: Conceptualization; Data curation; Formal analysis; Validation; Investigation; Visualization; Methodology; Writing—original draft; Writing—review and editing. **Xiaolu Guo**: Software; Formal analysis; Investigation; Visualization; Methodology; Writing—review and editing. **Katherine M Sheu**: Software; Formal analysis; Validation; Investigation; Visualization; Writing—review and editing. **Apeksha Singh**: Software; Validation; Methodology; Writing—review and editing. **Sarina C Lowe**: Validation; Investigation; Visualization; Writing—review and editing. **Minhao Han**: Investigation; Methodology. **Jessica Diaz**: Investigation. **Francisco Lopes**: Formal analysis; Investigation. **Roy Wollman**: Formal analysis; Writing—review and editing. **Alexander Hoffmann**: Conceptualization; Supervision; Funding acquisition; Writing—original draft; Project administration; Writing—review and editing.

Source data underlying figure panels in this paper may have individual authorship assigned. Where available, figure panel/source data authorship is listed in the following database record: biostudies:S-SCDT-10_1038-S44320-024-00047-4.

## Disclosure and competing interests statement

The authors declare no competing interests. Alexander Hoffmann is a member of the Advisory Editorial Board of Molecular Systems Biology. This has no bearing on the editorial consideration of this article for publication.

# Expanded View Figures

**Figure EV1.  Quality control of the experimental system for quantifying p38 and NFκB activity dynamics in murine macrophages.**

(**A**) Gating strategy for FACS of mVenus-RelA+ p38-KTR-mCerulean+ hMPs. (**B**) mCerulean and mVenus expression by flow cytometry in WT hMPs, mVenus-Rela hMPs, and mVenus-Rela hMPs transduced with p38-KTR-mCerulean before FACS. (**C**) Signaling responses to three LPS doses over 2 h in mVenus-Rela p38-KTR-mCerulean hMPDMs compared to parent cell line by Western Blotting for phospho-p38, phospho -MK2, phospho-CREB, and IκBα protein levels. Quantification in Fig. 1B. (**D**) Bulk p-p38 and p-MK2 protein levels in hMPDMs measured by Western Blotting in response to indicated stimulations over 4 h. Quantification in Figs. 1E and EV1D. (**E**) Comparison of p38 activity over 4 h in hMPDMs measured by p38-KTR microscopy (mean of trajectories) and by bulk phospho-MK2 levels measured by Western Blotting in response to three doses of P3C4, CpG, LPS, or TNF (0.1, 1, 100 ng/ml for P3C4, LPS, TNF; 1, 10, 1000 nM for CpG). Western Blotting quantification: band intensities were background corrected, normalized to tubulin control, normalized across multiple membranes using an internal control sample, and baseline-deducted; depicting data from a single experiment (Western Blot membrane shown in Fig. EV1E). For microscopy, the mean of means of trajectories from two biological replicates is shown (in total: 923, 1171, 970, and 1055 cells included in the analysis for P3C4, CpG, LPS, and TNF, respectively; same data as in Fig. 1E). (**F**) Gating strategy for flow cytometry measurements of intracellular p-p38 levels in hMPDMs. (**G**) Example quantification of p-p38+ fraction of cells stimulated with indicated doses of LPS for 30 min (blue). The fluorescence signal of an unstimulated, unstained sample is used to define a cutoff between p-p38+ and p-p38- cells (gray). Data from one experiment are displayed.

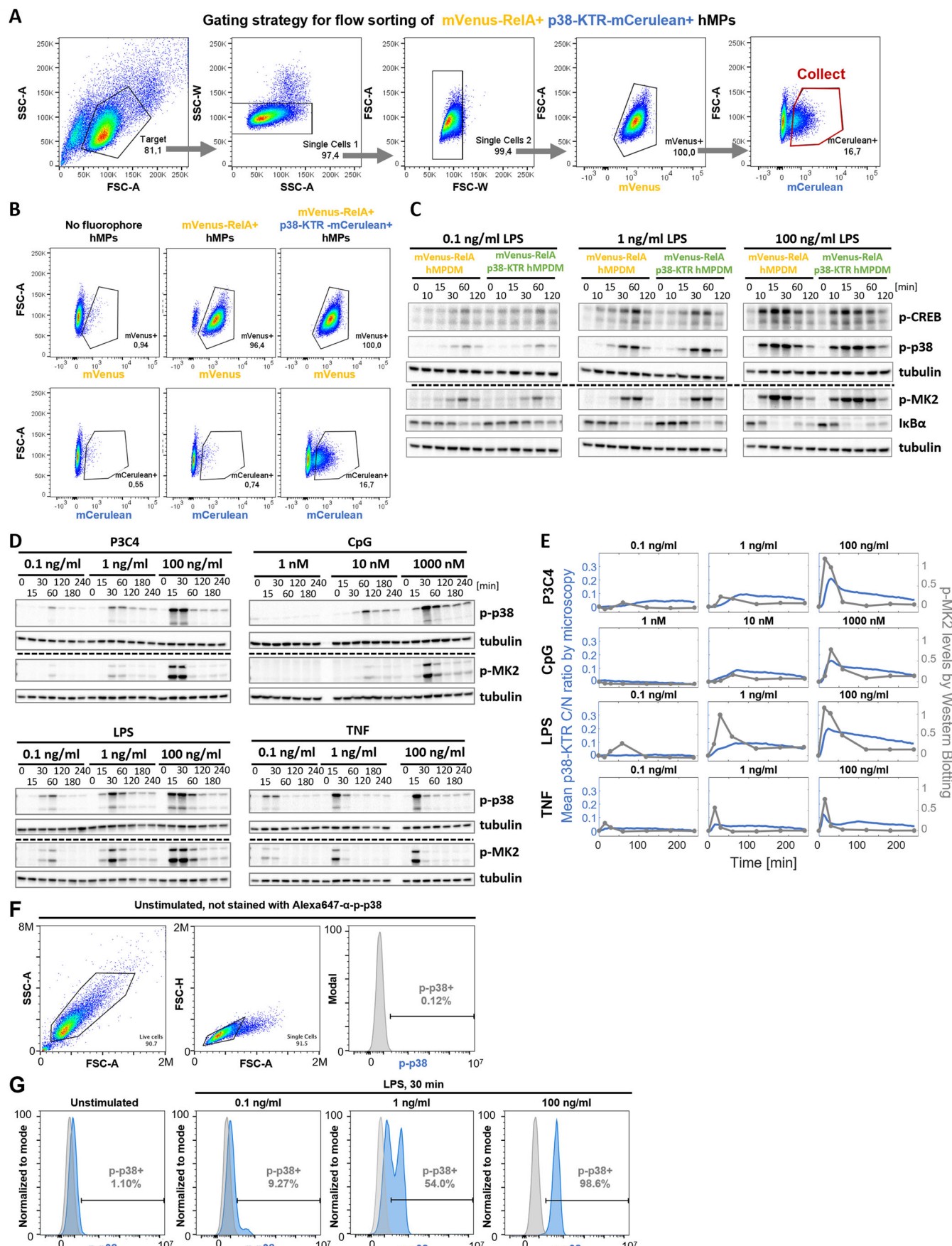

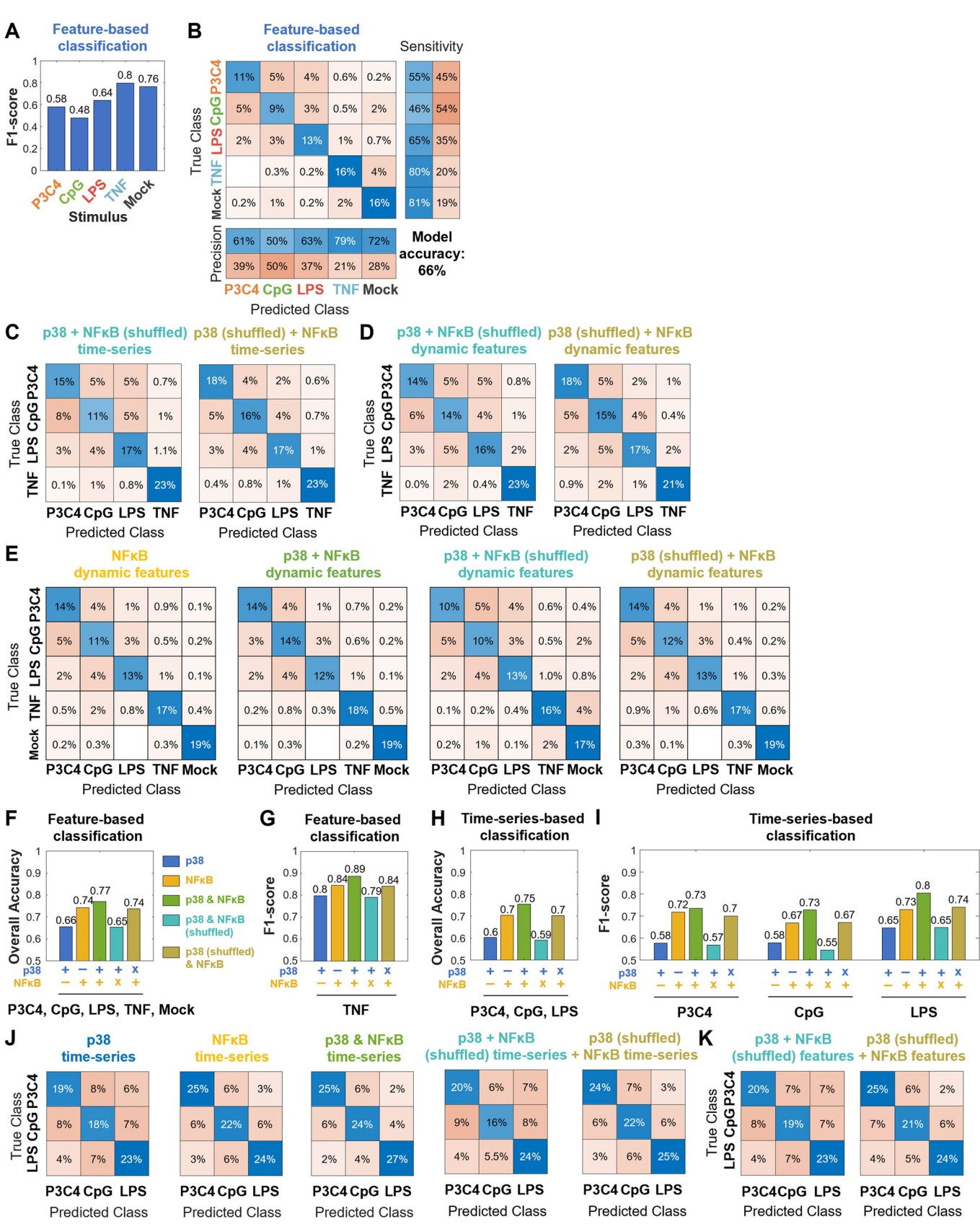

◀ **Figure EV2. Machine learning classifications of p38 and NFκB activities in response to four high-dose ligand stimulations or PAMP stimulations using shuffled activities as negative controls.**

(**A**, **B**) F1 scores by class and confusion matrix of decision tree ensemble classification of p38 dynamic features in response to mock and high-dose P3C4, CpG, LPS, and TNF stimulations. Data from two pooled biological replicates are used. (**C**, **D**) Confusion matrices of machine learning classification of NFκB ("shuffled": incorrectly matched NFκB activities from cells randomly selected from all classes) + p38 (left) or NFκB + p38 ("shuffled": incorrectly matched p38 activities from cells randomly selected from all classes) (right) time-series (**C**) or dynamic features (**D**) in response to high-dose P3C4, CpG, LPS, and TNF stimulations. Data from two pooled biological replicates are used. (**E**) Confusion matrices of machine learning classification of decision tree ensemble classification of NFκB only, NFκB + p38, NFκB ("shuffled": incorrectly matched NFκB activities from cells randomly selected from all classes) + p38, or NFκB + p38 ("shuffled": incorrectly matched p38 activities from cells randomly selected from all classes) dynamic features in response to mock and high-dose P3C4, CpG, LPS, and TNF stimulations. Data from two pooled biological replicates are used. (**F**, **G**) Overall classification accuracy (**F**) and F1 score for TNF class (**G**) in machine learning classifications using p38 only, NFκB only, NFκB + p38, NFκB ("shuffled": incorrectly matched NFκB activities from cells randomly selected from all classes) + p38, NFκB + p38 ("shuffled": incorrectly matched p38 activities from cells randomly selected from all classes) dynamic features in response to mock and high-dose P3C4, CpG, LPS, and TNF stimulations. Data from two pooled biological replicates are used. (**H–J**) Overall classification accuracy (**H**), F1 scores for individual classes (**I**), and confusion matrices (**J**) for machine learning classifications of time series using p38 only, NFκB only, NFκB + p38, NFκB ("shuffled": incorrectly matched NFκB activities from cells randomly selected from all classes) + p38, or NFκB + p38 ("shuffled": incorrectly matched p38 activities from cells randomly selected from all classes) in response to high-dose P3C4, CpG, and LPS stimulations. Data from two pooled biological replicates are used. (**K**) Confusion matrices of machine learning classification of NFκB (shuffled among all cells in all classes) + p38 (left) or NFκB + p38 (shuffled among all cells in all classes) (right) dynamic features in response to high-dose P3C4, CpG, and LPS stimulations. Data from two pooled biological replicates are used.

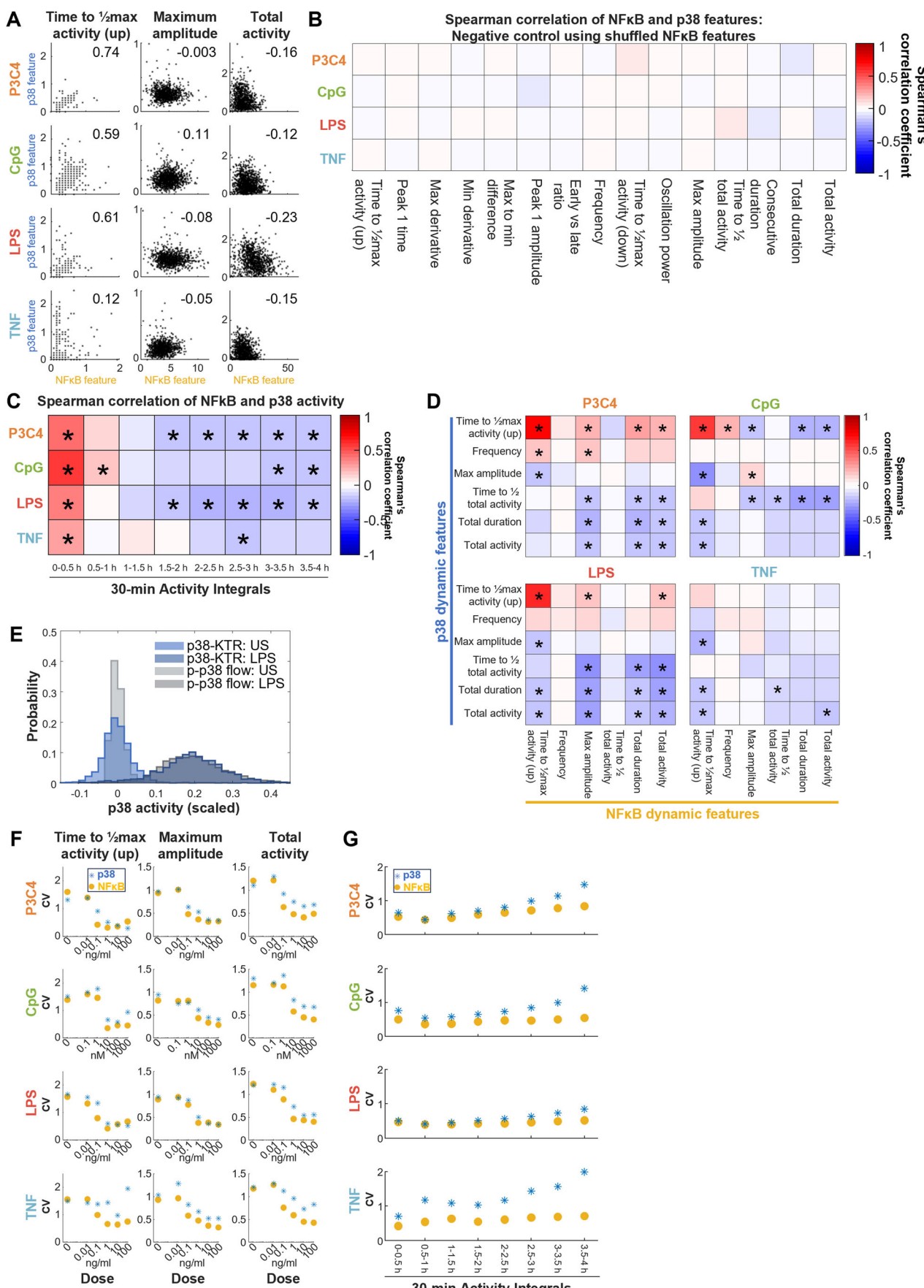

◀ **Figure EV3.   MAPK p38 and NFκB dynamic features are poorly correlated.**

(**A**) Scatter plots of correlations of selected p38 vs. NFκB dynamic features (as in Fig. 5A) upon stimulation with high-dose P3C4, CpG, LPS, or TNF. Number indicates the Spearman correlation coefficient. Data from two pooled biological replicates are depicted. (**B**) Negative control: Spearman correlation coefficients (CCs) between indicated p38 dynamic features and corresponding NFκB dynamic feature using NFκB features randomly shuffled with respect to p38 features among cells within a stimulation condition for stimulation with high-dose P3C4, CpG, LPS, or TNF. The asterisk indicates a statistically significant correlation ($p < 0.05$) with |CC | > 0.15. Data from two pooled biological replicates are used. (**C**) Spearman correlation coefficients (CC) between integrals of p38 activity over 30 min from 0 to 4 h and corresponding NFκB integrals upon stimulation with high-dose P3C4, CpG, LPS, or TNF. The asterisk indicates a statistically significant correlation ($p < 0.05$) with |CC | > 0.15. Data from two pooled biological replicates are used. (**D**) Spearman correlation coefficients (CCs) between indicated p38 dynamic features and all indicated NFκB dynamic features upon stimulation with high-dose P3C4, CpG, LPS, or TNF. The asterisk indicates a statistically significant correlation ($p < 0.05$) with |CC | > 0.15. Data from two pooled biological replicates are used. (**E**) Comparison of heterogeneity of p38 activity measured by flow cytometry (gray) and p38-KTR microscopy (blue) in unstimulated cells and upon 30 min 100 ng/ml LPS stimulation. For each assay, the mean of the unstimulated sample is deducted from each distribution. The flow cytometry distributions are scaled using the distance between the peaks of lognormal fits to the unstimulated and the stimulated distributions. Microscopy: Data from two pooled biological replicates are used. Flow cytometry: Data from a single experiment are displayed. (**F**) Coefficient of variation of indicated p38 and NFκB dynamic features upon stimulation across dose range of P3C4, CpG, LPS, or TNF. Data from two pooled biological replicates are used. (**G**) Coefficient of variation of 30-min p38 and NFκB activity integrals upon stimulation with high-dose P3C4, CpG, LPS, or TNF. Data from two pooled biological replicates are used.

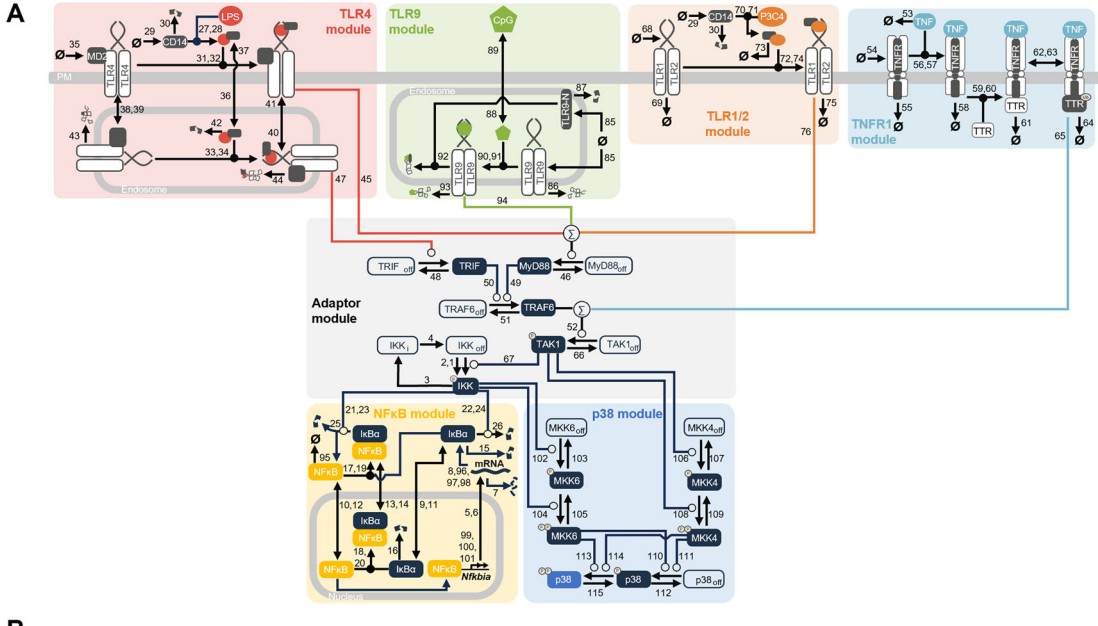

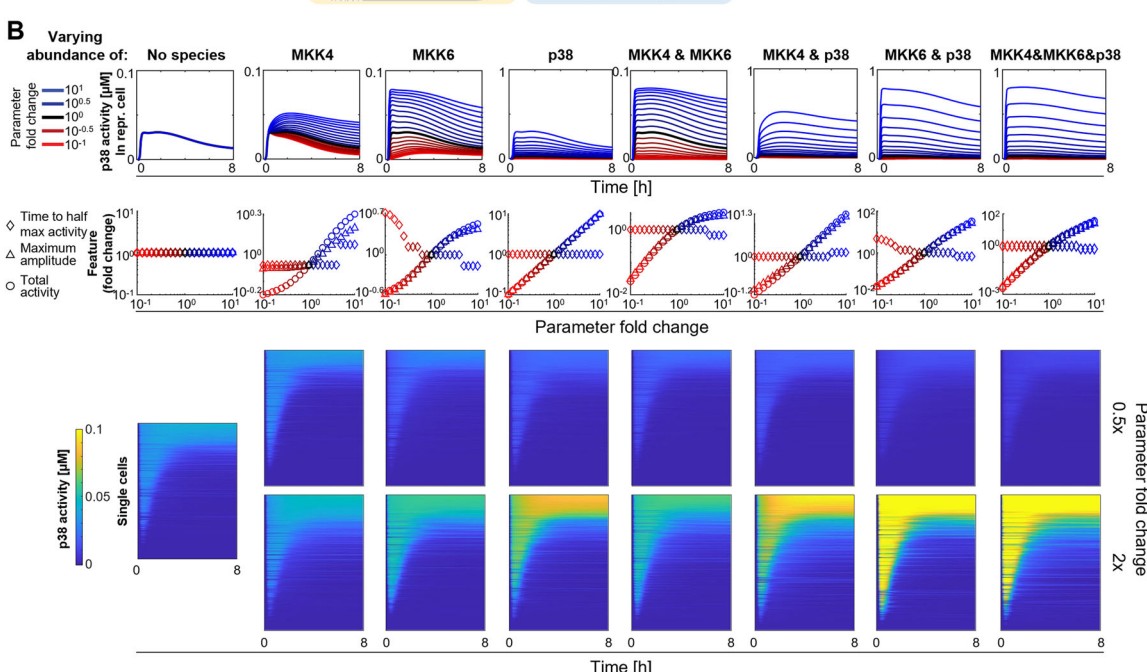

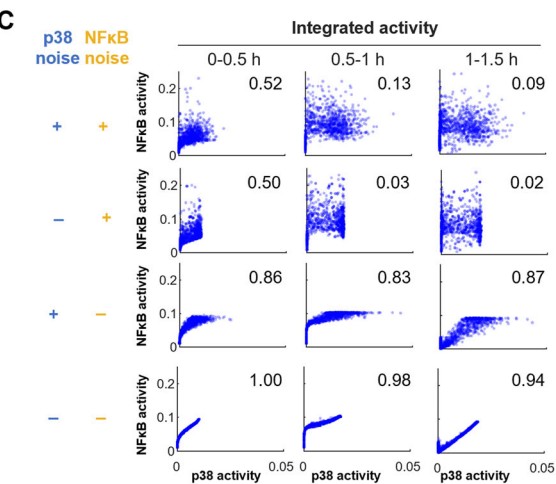

**Figure EV4.  An integrated mathematical model of MAPK p38 and NFκB activation suggests sources of heterogeneity.**

(A) Detailed schematic of mathematical model structure including reaction numbers (see also Dataset EV1). A model of p38 activation via IKK/Tpl2/MKK3/6 and TAK1/MKK4 (blue background) is integrated with established models of NFκB activation (yellow) downstream of TLR1/2 (orange), TLR9 (green), TLR4 (red), and TNF receptor (light blue) signaling (Adelaja et al, 2021; Luecke et al, 2023). (B) Parameter sensitivity analysis probes the effect of kinase abundances within the p38 module. Effect of variations of the abundance of indicated kinase(s) on p38 activity of a representative cell (top), p38 dynamic features in a representative cell (middle), and heterogeneous p38 activity trajectories in single cells upon simulated LPS stimulation (bottom). (C) Scatter plots of p38 vs. NFκB integrated activity over indicated 30 min intervals in simulated LPS stimulation with or without simulated molecular noise (i.e., using parameter distributions or fixed parameter values) in p38 and NFκB modules.

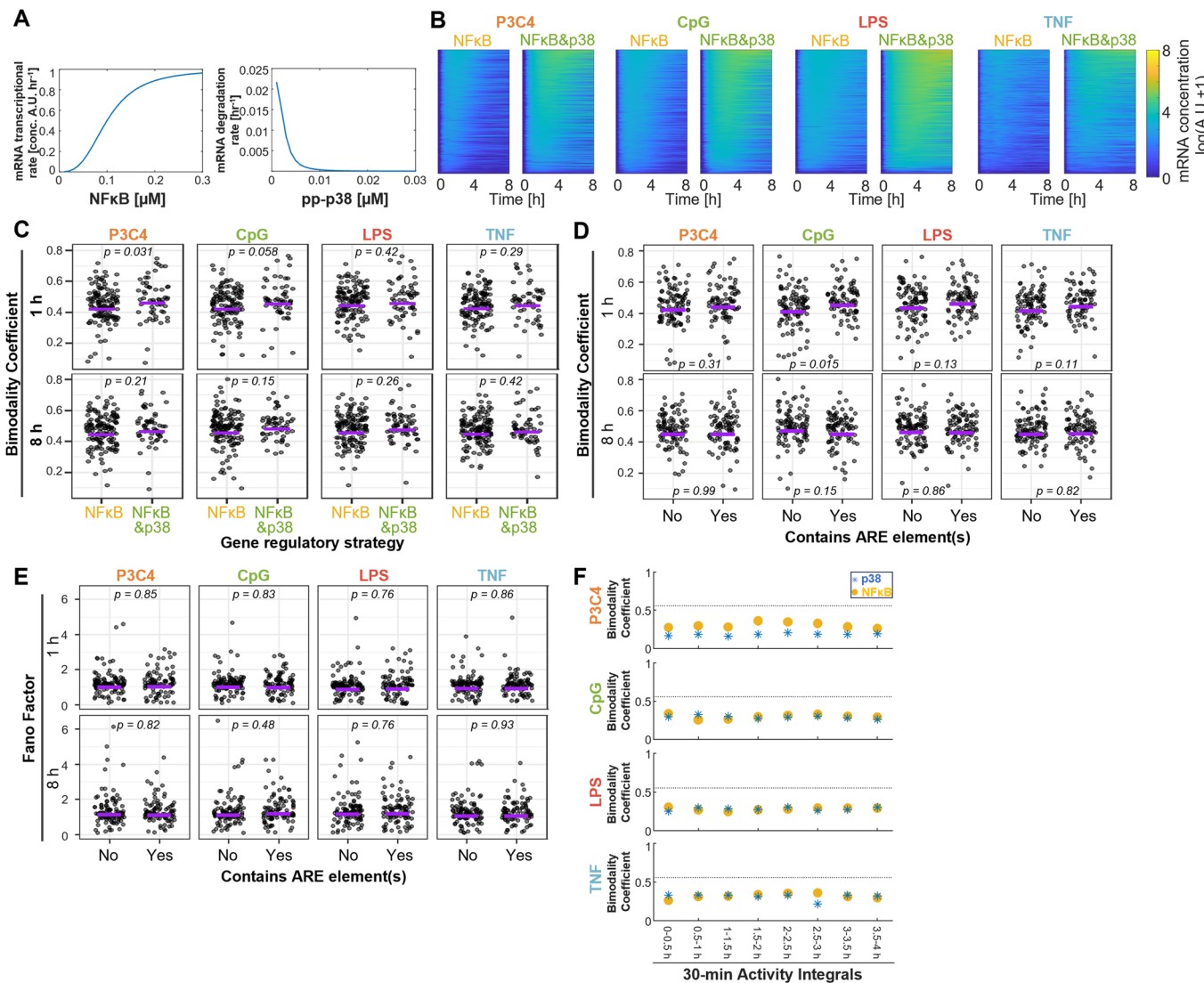

**Figure EV5.   MAPK p38 signaling contributes heterogeneity to macrophage gene expression responses.**

(A) Effect of NFκB and p38 activity levels on simulated mRNA transcription and mRNA-degradation, respectively. (B) Simulated mRNA expression under control of NFκB only or NFκB&p38 over 8 h using experimentally determined NFκB and p38 activity upon high-dose P3C4, CpG, LPS, and TNF stimulation as input to the mathematical model. (C) Bimodality coefficient of all immune response genes regulated by NFκB-only (132 genes) or NFκB&p38 (59 genes) in single cells upon stimulation with P3C4, CpG, LPS, or TNF at 1 and 8 h. Purple line indicates mean. Statistical significance was determined using a permutation test for difference in means using 10000 permutations. Data from one experiment are depicted. (D, E) Bimodality coefficient (D) and Fano Factor (E) of all immune response genes in single cells categorized by absence or presence of ARE-element(s) upon stimulation with P3C4, CpG, LPS, or TNF at 1 and 8 h. Purple line indicates mean. Statistical significance was determined using a permutation test for difference in means using 10000 permutations. Data from one experiment are depicted. (F) Bimodality Coefficients of 30-min p38 and NFκB activity integrals upon stimulation with high-dose P3C4, CpG, LPS, or TNF determined by dual reporter macrophage imaging. Data from two pooled biological replicates are used.

