## [Peer Review File · Molecular Systems Biology]

Dynamical and combinatorial coding by MAPK p38 and NF κ B in the inflammatory response of macrophages

Stefanie Luecke, Xiaolu Guo, Katherine Sheu, Apeksha Singh, Sarina Lowe, Minhao Han, Jessica Diaz, Francisco Lopez, Roy Wollman, and Alexander Hoffmann

Corresponding author(s): Alexander Hoffmann (ahoffmann@ucla.edu)

Review Timeline:

Submission Date:	22nd Oct 23
Editorial Decision:	29th Nov 23
Revision Received:	22nd Apr 24
Editorial Decision:	21st May 24
Revision Received:	27th May 24
Accepted:	28th May 24

Editor: Maria Polychronidou

Transaction Report:

29th Nov 2023

Manuscript Number: MSB-2023-12076

Title: Temporal and combinatorial coding by MAPK p38 and NFkB in the inflammatory response of macrophages

Dear Alex,

Thank you again for submitting your work to Molecular Systems Biology. We have now heard back from the three reviewers who agreed to evaluate your study. The reviewers appreciate that the study addresses a relevant topic. However, as you will see below, they raise a series of concerns, which we would ask you to address in a major revision.

Without repeating all the issues listed below, some of the more fundamental issues are the following:

- The reviewers mention that the findings described in the last two figures (ie. that that "and" gate regulation results in greater variability in gene regulation) remain somewhat preliminary and need to be better supported. Given that the reviewers point out that these findings are important for the overall advance provided by the study, we would strongly encourage you to follow the recommendations of the reviewers and perform some follow up analyses to strengthen this part of the study.
- The reviewers recommend some additional analyses to better support several of the conclusions. They make constrictive suggestions in this regard. Some specific points were highlighted during our cross-commenting process (in which the reviewers can make additional comments based on each other's reports): reviewer #1 mentioned that further explorations related to the questions listed in the major point #2 of reviewer #3 would increase the biological insight gained from the analyses shown in Figure 1-5. Reviewer #1 also indicated that the point raised by reviewer #3 regarding assessing heterogenous vs bimodal gene expression seems important and it is related to their own point about the complexity of p38 dependent gene regulatory mechanisms and the need to better link p38 dynamics (Figure 1-6) to the RNAseq (Figure 7).
- Reviewer #3 recommends editing the manuscript to make sure that it is easily accessible to immunologists and a more general audience beyond modeling experts.

All issues raised by the referees would need to be satisfactorily addressed. Please let me know in case you would like to discuss in further detail any of the issues raised, I would be happy to schedule a call.

On a more editorial level, we would ask you to address the following points:

- Please provide a .doc version of the manuscript text (including legends for the main figures) and individual production quality figure files for the main Figures (one file per figure).
- Please include 5 keywords.
- We have replaced Supplementary Information by the Expanded View (EV format). 5 additional figures can be provided as EV Figures. Please provide one file per EV Figure. Their legends should be included in the manuscript text. For detailed instructions regarding expanded view please refer to our Author Guidelines: . Further figures can be included in a PDF called Appendix. Appendix figures should be labeled and called out as: "Appendix Figure S1, Appendix Figure S2... Appendix Table S1..." etc. Each legend should be below the corresponding Figure/Table in the Appendix. Please include a Table of Contents in the beginning of the Appendix. For detailed instructions regarding expanded view please refer to our Author Guidelines: .
- Tables EV1-EV4 should be provided as separate .xls files. If they are long and/or complex (ie. do not fit in a single page) they should be provide and called out in the text as Datasets EV1-EV4. Please provide one file per EV Table or EV Dataset. Each Please include the description of each EV Table or Dataset in the file itself, i.e. in a separate tab for .xls files or as a README.txt file in .zip folders for .csv files.
- Please provide a "standfirst text" summarizing the study in one or two sentences (approximately 250 characters), three to four "bullet points" highlighting the main findings and a "synopsis image" (550px width and max 400px height, jpeg format) to highlight the paper on our homepage.
- Please include a "Disclosure and Competing Interests statement" in the main text.
- All Materials and Methods need to be described in the main text. We would encourage you to use 'Structured Methods', our new Materials and Methods format. According to this format, the Material and Methods section should include a Reagents and Tools Table (listing key reagents, experimental models, software and relevant equipment and including their sources and relevant identifiers) followed by a Methods and Protocols section in which we encourage the authors to describe their methods using a step-by-step protocol format with bullet points, to facilitate the adoption of the methodologies across labs. More information on how to adhere to this format as well as downloadable templates (.doc or .xls) for the Reagents and Tools Table can be found in our author guidelines: . An example of a Method paper with Structured Methods can be found here:

- Please include a Data availability section describing how the data, code etc. have been made available. This section needs to be formatted according to the example below:

The datasets and computer code produced in this study are available in the following databases:

- Chip-Seq data: Gene Expression Omnibus GSE46748 (<https://www.ncbi.nlm.nih.gov/geo/query/acc.cgi?acc=GSE46748>)

- Modeling computer scripts: GitHub (<https://github.com/SysBioChalmers/GECKO/releases/tag/v1.0>)

- [data type]: [full name of the resource] [accession number/identifier] ([doi or URL or identifiers.org/DATABASE:ACCESSION])

- For data quantification: please specify the name of the statistical test used to generate error bars and P values, the number (n) of independent experiments (specify technical or biological replicates) underlying each data point and the test used to calculate p-values in each figure legend. The figure legends should contain a basic description of n, P and the test applied. Graphs must include a description of the bars and the error bars (s.d., s.e.m.).

- The References should be formatted according to the Molecular Systems Biology reference style (i.e., ordered alphabetically and listing the first 10 authors followed by et al).

- When you resubmit your manuscript, please download our CHECKLIST (<https://bit.ly/EMBOPressAuthorChecklist>) and include the completed form in your submission.

Please note that the Author Checklist will be published alongside the paper as part of the transparent process (<https://www.embopress.org/page/journal/17444292/authorguide#transparentprocess>).

If you feel you can satisfactorily deal with these points and those listed by the referees, you may wish to submit a revised version of your manuscript. Please attach a covering letter giving details of the way in which you have handled each of the points raised by the referees. A revised manuscript will be once again subject to review and you probably understand that we can give you no guarantee at this stage that the eventual outcome will be favorable.

Best wishes,

Maria

Maria Polychronidou, PhD
Senior Editor
Molecular Systems Biology

We realize that it is difficult to revise to a specific deadline. In the interest of protecting the conceptual advance provided by the work, we recommend a revision within 3 months (27th Feb 2024). Please discuss the revision progress ahead of this time with the editor if you require more time to complete the revisions. Use the link below to submit your revision:

IMPORTANT: When you send your revision, we will require the following items:

1. the manuscript text in LaTeX, RTF or MS Word format
 2. a letter with a detailed description of the changes made in response to the referees. Please specify clearly the exact places in the text (pages and paragraphs) where each change has been made in response to each specific comment given
 3. three to four 'bullet points' highlighting the main findings of your study
 4. a short 'blurb' text summarizing in two sentences the study (max. 250 characters)
 5. a 'thumbnail image' (550px width and max 400px height, Illustrator, PowerPoint or jpeg format), which can be used as 'visual title' for the synopsis section of your paper.
 6. Please include an author contributions statement after the Acknowledgements section (see <https://www.embopress.org/page/journal/17444292/authorguide>)
 7. Please complete the CHECKLIST available at (<https://bit.ly/EMBOPressAuthorChecklist>).
- Please note that the Author Checklist will be published alongside the paper as part of the transparent process (<https://www.embopress.org/page/journal/17444292/authorguide#transparentprocess>).
8. When assembling figures, please refer to our figure preparation guideline in order to ensure proper formatting and readability in print as well as on screen:
<https://bit.ly/EMBOPressFigurePreparationGuideline>

See also figure legend guidelines: <https://www.embopress.org/page/journal/17444292/authorguide#figureformat>

9. Please note that corresponding authors are required to supply an ORCID ID for their name upon submission of a revised manuscript (EMBO Press signed a joint statement to encourage ORCID adoption).

(<https://www.embopress.org/page/journal/17444292/authorguide#editorialprocess>)

Currently, our records indicate that the ORCID for your account is 0000-0002-5607-3845.

Please click the link below to modify this ORCID:
Link Not Available

*** PLEASE NOTE *** As part of the EMBO Press transparent editorial process initiative (see our Editorial at <https://dx.doi.org/10.1038/msb.2010.72>), Molecular Systems Biology publishes online a Review Process File with each accepted manuscripts. This file will be published in conjunction with your paper and will include the anonymous referee reports, your point-by-point response and all pertinent correspondence relating to the manuscript. If you do NOT want this File to be published, please inform the editorial office at msb@embo.org within 14 days upon receipt of the present letter.

Reviewer #1:

Luecke et al. address the important question of how NFkB and MAPK pathways interface to encode stimulus specific information. A strength of the paper is use of multiplex live cell reporters in differentiated myeloid precursor cells that closely resemble primary macrophages. The majority of the manuscript (Fig 1-5) is focused on information theory and machine learning approaches and yields modest advances, in terms of both the computational approach or resulting biological insight. Several potentially impactful results are highlighted in the final two figures, related to mechanisms supporting divergence of NFkB and p38 dynamics and single cell signaling heterogeneity, and how the "AND" NFkB&p38 gene regulatory strategy impacts gene expression heterogeneity. However, these later findings must be explored further before it is possible to evaluate the conclusions.

Major points

1. It is not clear how results presented in Figures 1-5 significantly advance our understanding of signal integration, either fundamental principles or in the specific context of macrophage signaling. Many stated conclusions, such as the "key finding ... that p38 activation dynamics distinguish the host cytokine from the bacterial and viral PAMPs tested," are expected and can be observed without the innovate approaches used (ie. TNF produces a less robust p38 response compared to TLR ligands).

The work shown in Figures 1-5 also do not seem to provide a significant advances regarding computational approach. The approach seems quite similar to previous work from this lab (Adelaja et al Immunity 2021) and the authors did not note any significant advances regarding the computational approach.

2. A major strength of the manuscript is dynamic single cell data. The fact that the NFkB and p38 dynamics are not well correlated and diverge particularly at late timepoints is interesting. This point is first made in Figure 3, yet the correlations are not shown until Figure 6A. It would be help to show the correlations sooner and to show select features as scatterplots to help the reader visualize the relationships. Perturbations of the system would also be interesting (eg. TRIF KO or kinase inhibitors) as a means to understand the uncoupling of NFkB and p38 dynamics. Generally, increased focus on visualizing and understanding the uncoupling of NFkB and p38 responses within single cells, at the expense of many figures focused on mutual information and classification results, would increase the insight that can be gained from the analysis.

3. Quantification of signaling heterogeneity is lacking. For example, the authors state "Heterogeneity between single cells of some features appeared to decrease with increasing dose ...". It would be helpful to quantify heterogeneity over doses and across the ligands, comparing NFkB and p38. This is important given that the connection between signaling (Figures 1-6) and gene expression (Figure 7) is solely based on heterogeneity.

4. How dose P38-KTR heterogeneity compare with phospho-p38 heterogeneity (by immunofluorescence or flow)?

5. The work presenting in Figure 6 has potential for significant impact. The p38 model is new and an exciting addition to the NFkB model. With additional exploration, this could be expanded to multiple figures and greatly enhance our understanding of the system.

How did the authors choose the representative cells in figure 6C? It seems that many cells shown in earlier figures have a shorter duration of p38 activity, which would be more consistent with the Western data. From Figure 6F, it would seem the

model can produce a variety of example cells. However, the authors do not examine what parameters are important to achieve those different trajectories. I am quite enthusiastic about the efforts to sample a distribution of protein levels to simulate single cell heterogeneity, but there should be an effort to connect the varied parameters to the simulation outcomes. For example, what are the distinct impacts of varying MKK4, MKK6, and p38 levels? This could be multiple figures and would greatly increase the novelty and impact of the work.

6. The authors should attempt to link the dynamic imaging studies (Fig1-6) to the gene expression studies. The authors suggest that an NFkB AND P38 gene regulatory strategy would promote more single cell heterogeneity in expression of select genes. It would be interesting for the authors to use their mechanistic model to simulate fraction of cells that would express NFkB&P38 genes (based on the signaling dynamics, not modeling additional gene regulator mechanisms).

7. The assignment of genes to NFkB only and NFkB&P38 gene regulatory strategies should be explained in the results section. Considering there are many p38 dependent gene regulatory mechanisms, some effort should be made to identify subsets of genes that share specific regulatory strategies. This could be done using the timing of their expression, co-expression across single cells, or their dependence known mechanisms. For example, it is well known that post-transcriptional regulation of cytokines and other genes with AU rich elements (ARE) is dependent on p38; focus on these genes (AREscore; PMID: 22242014) may reveal more robust difference in the Fano factor. Alternatively, chromatin accessibility may predict heterogeneous gene expression; the authors could evaluate the accessibility of the NFkB only and NFkB&P38 genes in unstimulated BMDM using published datasets, and compare Fano scores.

8. The authors should select a few NFkB&P38 targets and quantify their expression, mRNA or protein, at the end of live cell imaging studies to demonstrate the link between signaling and expression. This has been done previously using FISH (PMID: 24530305) or could be accomplished by adding Brefeldin A during the late hours of imaging, fixing cells, staining for IL-6 and IL-12b protein, and associating cytokine with the dynamics.

Minor points

1. These papers should be cited in the introduction:

Lee , et al PMID: 24530305

Tong...Smale et al (already cited later in the manuscript).

2. While I understand the results, the stated conclusions that p38 does "not contribute to dose distinction" should be clarified at multiple points in the text. This sort of statement means something very different when talking about information theory and machine learning, versus how these signals are actually decoded by the cells.

Reviewer #2:

This manuscript from Luecke et al. evaluates the contribution of p38 kinase signaling to the ability of macrophages to mount unique responses to different pathogenic stimuli. They establish a nicely engineered cell system with endogenously tagged NF-kB and virally inserted p38-KTR to generate reporter macrophages and pair this system with live cell microscopy and computational analysis. The authors compare hundreds of p38 and NF-kB single-cell activity time series for three stimuli associated with pathogens and the host cytokine TNF. They perform a range of quantitative analyses to address the question of how p38 and NF-kB activities distinguish the different stimuli. A major conclusion from the paper is that while p38 activity dynamics are on their own sufficient to distinguish between the different stimuli reasonably well, combining p38 and NF-kB activities doesn't add substantially to ligand distinction. In fact, somewhat surprisingly, the p38 or NF-kB component of the signal can be scrambled between cells without much affecting the ability to predict the stimulus. The authors also examine correlations between the p38 and NF-kB dynamics within single cells, finding low correlation in general, and they explain this using an ODE model with a distribution of initial conditions, which shows results consistent with the low degree of correlation. Finally, the authors examine the relationship of the signaling reporter results to downstream gene expression. Here, the authors hypothesize that genes that are responsive to both NF-kB and p38 will show higher variability from cell to cell than genes regulated by a single pathway, since the uncorrelated dynamics of the two pathways will both influence the expression levels of jointly regulated genes. Data from single-cell RNAseq, with genes grouped according to their known regulators, are used to confirm a statistically significant difference consistent with this idea.

In general, this paper is a very well executed study of the correlation between two important signaling pathways involved in innate immune signaling and pathogen detection. While a substantial body of literature has investigated NF-kB dynamics at the single-cell level, there is much less known about how multiple pathways interact, and this study goes a long way toward filling this gap. The analyses examining the information content of the two pathways are nicely presented and paint a clear picture of their relationship of stimuli and dynamics. This relationship is somewhat different than what might have been expected (e.g. the two pathways could make strongly cooperative independent contributions to distinguishing the stimuli), and the paper does a good job of portraying the more nuanced conclusions that the data lead to. The idea proposed in the last two figures, that "and" gate regulation results in greater variability in gene regulation, is an interesting one, but it is less well grounded in data than the rest

of the paper. Overall, this is an important study that fits the journal well and will be of interest to readers both in innate immunity specifically and the broader signal transduction and systems biology fields.

Several issues should be addressed:

1. The models relating dynamic features (or time series) to stimuli in figures 2-4 consider the differences between the four stimuli, but don't include the mock treatment as a possible class (though there is some separate consideration of TNF vs. mock in Fig. 3H). From one perspective this is an understandable choice, but it leaves the reader wondering how the models would look if the mock treatment were included along with the four stimuli. Given that cells don't necessarily "know" that they are receiving a stimulus rather than no stimulus, I think this is an important question. Including the mock treatment is important for providing context to the models distinguishing the different stimuli. These alternate models could be included as supplementary figures.
2. The analysis of scRNA-seq gene categories in Fig. 7 feels incomplete. Why aren't p38-only genes included as a category? The details on which genes make up the NF-kB only and NF-kB+p38 classes are not provided, and there is only a brief and very general note in the Methods on how they were chosen. Although the hypothesis is interesting, it is hard to put a lot of stock in the conclusions drawn from this analysis as it is currently presented.
3. The text discussing the correlations shown in Fig. 6A doesn't seem to match the figure well. The text states that all correlations were low, but the coloring of the squares in the figure suggests that some of them are in the 0.7-0.8 range. The ambiguity could be from my misinterpretation of the color scale, or in what is considered "low" correlation. It would be useful to include actual numbers in addition to the heatmap coloring, and/or to indicate in the text what the numerical threshold is for a "low" vs. "high" correlation.
4. I think it is worth adding a section in the Discussion to make clear the general caveats and limitations of looking at the information content of signaling from the human perspective rather than the cell perspective. For example, the activity perceived by reporters from the whole-cell KTR measurement may miss meaningful differences in sub-cellular location.

Reviewer #3:

In this manuscript, the authors explore how the MAPK p38 pathway contributes to temporal encoding of the macrophage inflammatory response. The authors have previously explored temporal encoding by NF-kB. To build on this, they create a p38-NF-kB dual reporter primary murine cell line to measure and analyze signaling dynamics through these two nodes in single cells. They find that p38 dynamics alone can distinguish between TNF vs. PAMP stimulation in macrophages. However, when compared to NF-kB, it did not contribute much additional information to distinguishing between different PAMPs or between doses of the same PAMP. They observed that NF-kB and p38 activation was poorly correlated at the level of single cells, and computational modeling suggested that loss of correlation between the two pathways was due to a branch in the p38 pathway that contributed an early v. late component to its activation. Finally, they suggest that this might contribute to heterogeneity in target genes that require both NF-kB and p38 for expression.

Overall, this is an interesting paper with a technically challenging set up. A primary macrophage live-cell dual signaling reporter is a considerable feat and the number of single cells analyzed across 4 conditions and 5 doses is impressive. This allowed the authors to make some very interesting insights into the differential dynamic encoding of p38 vs. NF-kB. My main questions pertain to the conclusion regarding p38's role in distinguishing doses, both the technical robustness of this conclusion and the interpretation (see points below). Also, this manuscript should be of high interest both to systems biologists and to cell biologists interested in innate immune signaling, however as written it is not very accessible to the latter group.

Major comments

1. I appreciate the extent to which the authors benchmarked the measurements of the p38 KTR reporter with Western blots. In figure 1E, it seems that the p38 dynamics of this reporter cells differ from those of the Western, especially at lower doses. A previous study comparing a FRET reporter versus a KTR reporter for ERK showed that the FRET reporter was sensitive at lower activity levels, and the KTR was more sensitive at higher activity levels (<https://doi.org/10.1016/j.cels.2017.10.019>). Is it possible that there is a loss of biological information in the KTR assay (especially at low doses), and that this might affect the conclusion that p38 does not contribute to discrimination of different PAMPs or doses? On a related note, did the authors compare fractional activation of p38 using their reporter to flow analysis of p-p38?
2. A main conclusion of the paper is that p38 does not contribute to dose distinction for PAMPs. The authors compare their results to Gottschalk et al (Cell Systems 2016). In my reading, the main conclusion from the Gottschalk paper is that the switch-like induction of the MAPKs following KLA stimulation enables macrophages to distinguish homeostatic v. pathogenic doses of PAMPs. Given this, one would not expect p38 to contribute to fine dose distinction for LPS, but rather to distinguish between 'harmless' vs. 'harmful' levels. The authors mention this in the Discussion on p. 13 ("This suggests that p38 mainly provides information about the absence or presence of a high dose stimulus.") but this seems like an important point that could be tested directly:
 - a. How would p38 v. NF-kB perform if tasked with distinguishing sub-threshold vs. high-threshold doses for LPS rather than all doses?
 - b. It's interesting that not all the PAMPs appear to follow this same switch-like induction for p38. How they compare in performing

this simpler task?

3. The observation of heterogeneity (and thus weak correlations) in activation of p38 and NF-kB is interesting. The authors suggest this can be linked to heterogeneity in downstream targets that require both signals, especially cytokines, and show higher Fano factors for those. Bimodal distributions can lead to high Fano factors, and Fig. 7A looks like this is the case for Il1a and Il1b. Again, the authors mention this is in the Discussion ("We therefore suggest a new hypothesis, that AND gate control of gene expression may have evolved for those genes that must be expressed only in a minority of cells."). Combined with the point above that p38 shows thresholding behavior:

a. Do the NF-kB+p38 gene targets show more bimodal behavior relative to NF-kB only targets? Other studies suggest this could be the case.

b. Is there anything in the signaling dynamics of p38 (even at the high doses) that would contribute not only to heterogeneity, but to such bimodal behavior at the single-cell level?

The authors seem to suggest in the Discussion that such analysis would be beyond the scope of this paper, which is fair. However, connecting heterogeneity with bimodality (or fractional expression) is a point worth making in the main text if it's indeed there.

4. The manuscript would benefit from more background explanation. For example, it would be helpful to include the definition of a 'bit', what this measure means, and how to interpret its significance. It would also be helpful to include a brief explanation for why LSTM and decision tree classifiers were chosen. As written, the work is not very accessible to a non-expert.

Minor comments

1. Fig. 1E: it's not clear what is meant by the labels "Dose 2/5, Dose 3/5 etc."

2. In figure 3D-H (and in other similar figures) all the panels share the same color scheme so it might be easier to a legend rather than using the +, - system at the bottom of each graph. Also, it is also not obvious what is meant by "shuffling" of the p38 or NFkB.

3. p5: Include some brief background on the PAMPs used (full name, receptor, pathway)

4. p6: LSTM is not defined.

Editor

Thank you again for submitting your work to Molecular Systems Biology. We have now heard back from the three reviewers who agreed to evaluate your study. The reviewers appreciate that the study addresses a relevant topic. However, as you will see below, they raise a series of concerns, which we would ask you to address in a major revision.

Without repeating all the issues listed below, some of the more fundamental issues are the following:

- The reviewers mention that the findings described in the last two figures (ie. that that "and" gate regulation results in greater variability in gene regulation) remain somewhat preliminary and need to be better supported. Given that the reviewers point out that these findings are important for the overall advance provided by the study, we would strongly encourage you to follow the recommendations of the reviewers and perform some follow up analyses to strengthen this part of the study.

We are pleased that the reviewers appreciate the finding about gene expression variability. In the revision we have expanded on that to dedicate 3 figures to this, thereby contracting the first part of the manuscript to 4 figures. They are detailed below.

In addition, we have clarified what the conclusions are from Figures 2-4, which may have been a little bit muddled in the first version. What these figures show is that, while combinatorial and dynamical coding are not mutually exclusive and in principle can occur together and function synergistically, in the response to inflammatory stimuli they appear to be redundant. Thus, immune response genes gain stimulus-specificity by evolving regulatory mechanisms that can decode dynamical information present in NF κ B trajectories (or to a lesser degree p38 trajectories), or that can decode the combinatorics of NF κ B and p38 activation per se, but they gain little further stimulus-specificity by being able to decode both combinatorics and dynamics of these pathways. This is a new insight that we believe is of broad interest to the signaling and innate immunity research community.

- The reviewers recommend some additional analyses to better support several of the conclusions. They make constrictive suggestions in this regard. Some specific points were highlighted during our cross-commenting process (in which the reviewers can make additional comments based on each other's reports): reviewer #1 mentioned that further explorations related to the questions listed in the major point #2 of reviewer #3 would increase the biological insight gained from the analyses shown in Figure 1-5. Reviewer #1 also indicated that the point raised by reviewer #3 regarding assessing heterogenous vs bimodal gene expression seems important and it is related to their own point about the complexity of p38 dependent gene regulatory mechanisms and the need to better link p38 dynamics (Figure 1-6) to the RNAseq (Figure 7).

We have now significantly restructured the figures: we have added to the dose specificity analysis (now Figure 4) and provide binary classifications of adjacent doses, binary classifications of mock from all individual doses, and a comparison of dose distinction using all vs. a subset of dynamic features. These lead to several interesting and consistent conclusions. We have also included a thorough discussion of the cell's interpretation of dose responses (for details see response to Reviewer #3, point #2, Reviewer #1, minor point #2, Reviewer #2 point #4).

We also now provide a thorough analysis of the question of bimodality as suggested by reviewer #3, also in the context of ARE containing genes (as suggested by reviewer #1) (for details, see response to Reviewer #3,

point #3 and Reviewer #1, point #7). To better connect the signaling dynamics with gene expression, we now include mathematical modeling of AND-gate controlled gene expression and relating that to experimental signaling dynamics and single-cell gene expression analysis (for details, see response to Reviewer #1, point #6).

- Reviewer #3 recommends editing the manuscript to make sure that it is easily accessible to immunologists and a more general audience beyond modeling experts.

We have extended several of the explanations of the quantitative analyses in the intro and early in the Results section (for details response to Reviewer #3, point #4) and hope that the accessibility for researchers with a less quantitative focus is now improved.

Reviewer #1:

Luecke et al. address the important question of how NF κ B and MAPK pathways interface to encode stimulus specific information. A strength of the paper is use of multiplex live cell reporters in differentiated myeloid precursor cells that closely resemble primary macrophages. The majority of the manuscript (Fig 1-5) is focused on information theory and machine learning approaches and yields modest advances, in terms of both the computational approach or resulting biological insight. Several potentially impactful results are highlighted in the final two figures, related to mechanisms supporting divergence of NF κ B and p38 dynamics and single cell signaling heterogeneity, and how the "AND" NF κ B&p38 gene regulatory strategy impacts gene expression heterogeneity. However, these later findings must be explored further before it is possible to evaluate the conclusions.

We thank the reviewer for their assessment of our study, especially for noting the potential relevance of the uncorrelated heterogeneity in p38 and NF κ B dynamics and resulting effects on gene regulation, and for the many helpful suggestions they provide. We address their comments in detail below.

Major points

1. It is not clear how results presented in Figures 1-5 significantly advance our understanding of signal integration, either fundamental principles or in the specific context of macrophage signaling. Many stated conclusions, such as the "key finding ... that p38 activation dynamics distinguish the host cytokine from the bacterial and viral PAMPs tested," are expected and can be observed without the innovate approaches used (ie. TNF produces a less robust p38 response compared to TLR ligands).

The work shown in Figures 1-5 also do not seem to provide a significant advances regarding computational approach. The approach seems quite similar to previous work from this lab (Adelaja et al Immunity 2021) and the authors did not note any significant advances regarding the computational approach.

We now recognize that our focus on thorough documentation may have distracted readers from the key findings. We actually think that the work described in these figures is significant and surprising, running counter to often repeated notions of combinatorial signaling. What these figures show is that while combinatorial and dynamical coding are not mutually exclusive and in principle can occur together and function synergistically, in the response to inflammatory stimuli they appear to be redundant. Thus, immune response genes gain stimulus-specificity by evolving regulatory mechanisms that can decode dynamical information present in NF κ B trajectories (or to a lesser degree p38 trajectories), or that can decode the combinatorics of NF κ B and p38 activation per se, but they gain little further stimulus-specificity by being able to decode both combinatorics and dynamics of these pathways. This is a new insight that we believe is of broad interest to the signaling and innate immunity research community.

In order to focus better on the key message (without losing rigor that is especially required when findings don't meet expectations) we have restructured and streamlined the figures such that we now have Figures 1-4:

Figure 1 establishes a technically challenging new primary-cell-like dual reporter macrophage for simultaneous measurements of p38 and NF κ B dynamics and thoroughly tests the validity of this system for making statements about p38 activity dynamics.

Figure 2 addresses the question of p38 stimulus specificity. We find that p38 dynamic activity provides a surprisingly accurate means of distinguishing TNF from PAMPs, even if responses to PAMPs are less well distinguishable. This is new information in the field. To make this finding we used established computational approaches (information theory, decision tree ensemble classification of features), and also through newly implemented approaches (LSTM classification of time series data), which taken together provide confidence in the conclusions.

Figure 3 addresses whether the combination of dynamic signaling of p38 and NFκB provides better stimulus specificity than when only one pathway is assessed. We find that there is surprisingly little gain. NFκB has such rich dynamics that p38 only contributes very little in distinguishing PAMPs. On the flipside p38 is so good at distinguishing TNF from PAMPs that NFκB is not needed.

Figure 4 extends this analysis by asking whether combinatorial signaling contributes possibly more to the distinction of doses. On the contrary, the answer is “no”, even though p38 dynamics alone are capable of distinguish low from high dose stimuli. We consistently find no evidence of a role of combinatorial signaling increasing the available dose-specific information.

These findings are in contrast to the prior hypothesis long-held by the field that combinatorial signaling between NFκB and p38 MAPK increases available stimulus-specific information. We believe we provide a thorough analysis and nuanced discussion of the biological implications that represents significant advances. Reviewer #2 appears to agree with this: “[...] much less known about how multiple pathways interact, and this study goes a long way toward filling this gap. The analyses examining the information content of the two pathways are nicely presented and paint a clear picture of their relationship of stimuli and dynamics. This relationship is somewhat different than what might have been expected (e.g. the two pathways could make strongly cooperative independent contributions to distinguishing the stimuli), and the paper does a good job of portraying the more nuanced conclusions that the data lead to.”

As we contracted the first 5 figures into 4, we now dedicate the next 3 figures to explore what the combination of NFκB and p38 may be doing, given that they do not combine to provide more information or precision in stimulus specificity.

Figure 5 + EV3 reports on the surprising correlation structure between NFκB and p38, Figure 6 + EV4 identifies mechanistic explanations for this surprising correlation structure by leveraging a mathematical modeling approach, and Figure 7 + EV5 reports the functional consequence of the correlation structure in generating bimodality in the expression of combinatorial AND gate target genes; this is predicted with a math model and tested with single cell RNAseq.

2. A major strength of the manuscript is dynamic single cell data. The fact that the NFκB and p38 dynamics are not well correlated and diverge particularly at late timepoints is interesting. This point is first made in Figure 3, yet the correlations are not shown until Figure 6A. It would be help to show the correlations sooner and to show select features as scatterplots to help the reader visualize the relationships. Perturbations of the system would also be interesting (eg. TRIF KO or kinase inhibitors) as a means to understand the uncoupling of NFκB and p38 dynamics. Generally, increased focus on visualizing and understanding the uncoupling of NFκB and p38 responses within single cells, at the expense of many figures focused on mutual information and classification results, would increase the insight that can be gained from the analysis.

We thank the reviewer for their interest in the single cell dynamics and the surprising lack of strong correlations.

We've streamlined the earlier figures, such that we now arrive at the analysis of the correlations in Figure 5. We'd like to keep the current order of figures as the first few figures focus on the motivating hypothesis of the study: that combinatorial dynamics of NFκB and MAPK enhance stimulus-specificity of macrophage responses. When instead we observe redundancy in the combinatorial dynamics, we then ask why that is the case, which leads us to the correlation analysis in Figure 5 and the functional consequences on gene expression in Figures 6 and 7.

All of Figure 5 and EV3 are now devoted to a thorough analysis of the correlations and the heterogeneity of experimentally determined dynamics. We provide visualization of scatter plots of select dynamic features and 30-min activity integrals. We've also added an exploration of correlations not just between corresponding p38 and NFκB dynamic features, but also between different types of p38 and NFκB features (e.g. correlation between p38 max amplitude with NFκB duration, etc).

We also now use the mathematical model of p38 and NFκB signaling to perturb the system in an attempt to restore correlations and thus identify the source of the lack of correlations by "denoising" the parameters in the NFκB and p38 signaling modules (Figure 6G, EV4C) and find that removing noise in the NFκB module restores much of the correlations, with noise in the p38 module providing a smaller contribution. We also test the effect of denoising of these modules on heterogeneity (as measured by CV) and find that although the range of values is affected by denoising (Figure EV4C), the CV is not much affected – indicating that noise upstream of pathway branching is sufficient to mediate the observed CVs.

Since the reviewer suggests a TRIF knockout, we provide results of simulated LPS-stimulation in TRIF and also MyD88 knockouts below (Figure RL1 A and B). In a TRIF knockout the correlations between 2 h activity integrals of p38 and NFκB are very similar from WT, and a MyD88 knockout reduces the degree of NFκB activation substantially, which renders the increase in calculated correlation less meaningful. Thus, it appears that while TRIF contributes to heterogeneity, it is not the reason for lack of correlations – which makes sense because 1) the lack of correlation is seen in multiple immune response pathways in which TRIF does not participate, and 2) the source for the lack of correlation is more likely downstream of the signaling branch, i.e. TAK1 and IKK, which is also documented in (Figure RL1 C and D).

Figure RL1. Uncoupling noise in different genotype and modules

(A) Scatter plots of p38 vs. NFkB integrated activity over indicated 2-hour intervals in simulated LPS stimulation with simulated molecular noise (i.e. using parameter distribution) in p38 and NFkB modules for different genotypes, TRIF knockout, MyD88 knockout, and wild type (WT).

(B) Spearman correlation coefficients between p38 and NFkB integrated activity in simulated LPS stimulation with simulated molecular noise in the p38 and NFkB modules.

(C) Scatter plots of p38 vs. NFkB integrated activity over indicated 2-hour intervals in simulated LPS stimulation with different denoise strategies in p38 and NFkB modules. Denoise is achieved by collapsing the parameter distribution to one single value.

(D) Spearman correlation coefficients between p38 and NFkB integrated activity in simulated LPS stimulation with simulated molecular noise in the p38 and NFkB modules.

3. Quantification of signaling heterogeneity is lacking. For example, the authors state "Heterogeneity between single cells of some features appeared to decrease with increasing dose ..." It would be helpful to quantify heterogeneity over doses and across the ligands, comparing NFκB and p38. This is important given that the connection between signaling (Figures 1-6) and gene expression (Figure 7) is solely based on heterogeneity.

We thank the reviewer for this suggestion. We have now devoted Figure 5 to the question of heterogeneity and correlations. We include quantifications of the coefficient of variation, a mean-normalized measure of the variability of a distribution, of a selection of dynamic features at high doses (Figure 5D) and across dose responses (Figure EV3F), and of 30-min activity integrals across 4 h (Figure EV3G), allowing for comparisons of heterogeneity between stimuli, between NFκB and p38, and across doses. We find that p38 heterogeneity is generally slightly higher or similar to NFκB, that heterogeneity indeed decreases with increasing dose, that p38's heterogeneity increases across the time course while this increase is less pronounced for NFκB, and that TNF-induced p38 activity has more heterogeneity than the other stimuli. We also provide quantification of the simulated LPS-induced p38 and NFκB dynamics and analyze the effect of denoising of p38 and NFκB modules on the coefficient of variation, determining noise in parameters upstream of NFκB and p38 modules as sufficient for generating the CVs of the dynamics (Figure 6 H, I). We hypothesize that the uncorrelated heterogeneity in p38 and NFκB dynamics mediates increased heterogeneity and bimodality for AND gate regulated genes, using a mathematical model of gene expression to connect Figures 5+6 to Figure 7. This is tested experimentally in Figure 7.

4. How dose P38-KTR heterogeneity compare with phospho-p38 heterogeneity (by immunofluorescence or flow)?

This is an important question that in part overlaps with that posed by Reviewer #2 point #1. We now provide a comparison of fractional activation of p38 across doses for all 4 stimuli measured by p38-KTR and p-p38 flow cytometry (Figure 1G) and, as a control before quantification of heterogeneity of the signaling dynamics, of the heterogeneity of p38 activity upon LPS stimulation at 30 min (Figure EV3E) and find that stimulus-induced heterogeneity is very similar between KTR and flow measurements.

5. The work presenting in Figure 6 has potential for significant impact. The p38 model is new and an exciting addition to the NFκB model. With additional exploration, this could be expanded to multiple figures and greatly enhance our understanding of the system.

How did the authors choose the representative cells in figure 6C? It seems that many cells shown in earlier figures have a shorter duration of p38 activity, which would be more consistent with the Western data. From Figure 6F, it would seem the model can produce a variety of example cells. However, the authors do not examine what parameters are important to achieve those different trajectories. I am quite enthusiastic about the efforts to sample a distribution of protein levels to simulate single cell heterogeneity, but there should be an effort to connect the varied parameters to the simulation outcomes. For example, what are the distinct impacts of varying MKK4, MKK6, and p38 levels? This could be multiple figures and would greatly increase the novelty and impact of the work.

We thank the reviewer for their appreciation of the model of p38 signaling in macrophage immune responses. We now devote the entire Figure 6 and Figure EV4 to the model and provide additional explorations and hope that the reviewer finds the impact of this analysis increased.

For the initial parameterization, we chose representative cells that capture the median of characteristic dynamic features (p38 signaling codons) induced each ligand. As can be seen in the heatmaps of p38 trajectories and in the example trajectories (which include the representative cells) (Figure 2A), many cells have extended activity duration. Also, the NFκB trajectories were not considered when choosing the representative cells for the model – thus the excellent match of the experimental with the simulated NFκB trajectories for all four stimuli indicates that the chosen representative experimental cells have quite typical signaling in upstream and NFκB pathway components.

The model indeed allows for exploring the sources of the heterogeneity of p38 signaling dynamics. Prompted by the reviewer's excellent suggestion, we now provide a detailed sensitivity analysis investigating the effects of changes to MKK4, MKK6, and p38 concentrations and combinations thereof on the simulated representative cell trajectories, on select dynamic features, and on the single cell trajectories (Figure EV3B). We describe in detail in the Results section how this allows us to differentiate the effects of concentrations the different kinase on overall signaling strength, and early and late phase activity.

Other additional panels in the math modeling figure now include an investigation of the sources of noise resulting in the observed lack of correlation between p38 and NFκB activity via denoising of the signaling modules (Figure 6G, EV4C) and quantification of heterogeneity in simulated signaling dynamics (Figure 6H, I). We find that noise in the NFκB module is the most relevant contributor to the weak correlations between p38 and NFκB activity, while the heterogeneity (measured as coefficient of variation) is mainly determined by upstream modules.

6. The authors should attempt to link the dynamic imaging studies (Fig1-6) to the gene expression studies. The authors suggest that an NFκB AND P38 gene regulatory strategy would promote more single cell heterogeneity in expression of select genes. It would be interesting for the authors to use their mechanistic model to simulate fraction of cells that would express NFκB&P38 genes (based on the signaling dynamics, not modeling additional gene regulator mechanisms).

Thank you for this suggestion. We now link the heterogeneous uncorrelated signaling dynamics to the higher variability in AND gate controlled genes with mathematical model of AND gate gene control by NFκB (positively controlling mRNA transcription) and p38 (negatively controlling mRNA degradation), using experimentally determined signaling dynamics induced by all four stimuli as input, and predict larger variability under NFκB&p38 control than under NFκB-only control (Figure 7A, B).

7. The assignment of genes to NFκB only and NFκB&P38 gene regulatory strategies should be explained in the results section. Considering there are many p38 dependent gene regulatory mechanisms, some effort should be made to identify subsets of genes that share specific regulatory strategies. This could be done using using the timing of their expression, co-expression across single cells, or their dependence known mechanisms. For example, is well known that post-transcriptional regulation of cytokines and other genes with AU rich elements (ARE) is dependent on p38; focus on these genes (AREscore; PMID: 22242014) may reveal more robust difference in the Fano factor. Alternatively, chromatin accessibility may predict heterogeneous gene expression; the authors could evaluate the accessibility of the NFκB only and NFκB&P38 genes in unstimulated BMDM using published datasets, and compare Fano scores.

We've now added the explanation of gene assignments to the Results section, in addition to providing it in Materials & Methods. All the details of which genes make up the NFκB-only and the NFκB+p38 categories and

how they were assigned are given in Table EV5 (to which we now refer in Results and Materials & Methods section). As stated in the Materials & Methods, category assignment was performed based on thorough literature-based manual curation of previously published gene assignments that were obtained using knockout cell lines as well as quantitative modeling of gene expression data refined in multiple publications (Cheng *et al*, 2017; Tong *et al*, 2016; Sen *et al*, 2020; Wang *et al*, 2021; Sheu *et al*, 2023). We apologize that Table EV5 was not provided in the original submission.

Thank you for the great suggestion regarding the focus on shared regulatory strategies. We now include an analysis of Fano Factor and Bimodality Coefficient for genes with and without ARE (Figure EV5D, E). While this did not reveal robust differences in Fano Factors, when focusing on a comparison of genes with and without ARE among genes classified as p38&NFκB dependent, we indeed observed a pronounced difference in Bimodality coefficients that was not observed among genes classified as NFκB dependent (Figure 7F).

8. The authors should select a few NFκB&P38 targets and quantify their expression, mRNA or protein, at the end of live cell imaging studies to demonstrate the link between signaling and expression. This has been done previously using FISH (PMID: 24530305) or could be accomplished by adding Brefeldin A during the late hours of imaging, fixing cells, staining for IL-6 and IL-12b protein, and associating cytokine with the dynamics.

While Reviewer 3 suggests a certainly important experiment and we have made some progress towards achieving multiplexed quantification of gene expression after imaging, this is currently beyond our available personnel resources to implement at a scale and with the technical accuracy required given the heterogeneity of signaling and predicted gene-specificity of decoding. We are committed to performing these experiments in the future, but it is beyond the scope of the current study, which already provides significant advances. Further, the finding that inflammatory AND gate genes may interpret two uncorrelated distributions into bimodal gene expression distributions does not necessarily require linked signaling and mRNA data.

Minor points

1. These papers should be cited in the introduction:

Lee , et al PMID: 24530305

We apologize for this oversight and now cite this excellent paper in the introduction in the context of stimulus-specific NFκB signaling codons and their decoding by gene-specific mechanisms.

Tong...Smale et al (already cited later in the manuscript).

This paper does not address mechanisms that mediate stimulus-specific gene expression (whereas Cheng et al 2017, and Sen et al 2020 do), but we now cite this in the introduction in the context of gene-specific regulatory mechanisms.

2. While I understand the results, the stated conclusions that p38 does "not contribute to dose distinction" should be clarified at multiple points in the text. This sort of statement means something very different when talking about information theory and machine learning, versus how these signals are actually decoded by the cells.

We agree with the reviewer. The established methods in the field to quantify information contained in signaling use information theoretic and machine learning algorithms. By those measures, p38 dynamics alone

provides some ability to distinguish doses, NFκB dynamics has good capacity to distinguish doses. But the combination of the two does not perform better than NFκB alone.

We have now added new machine learning classification of doses to Figure 4 (dose specificity) comparing use of all dynamic features to use of 'total activity' only, demonstrating that, when using 'total activity' only, the intuitive prediction holds true that p38 contributes to improved distinguishability of doses in combination with NFκB compared to either NFκB or p38 activities alone (Figure 4D). However, when all dynamic features are used for dose classification, we consistently see no improvement using the combination of p38 and NFκB dynamics for classification of 5 doses, adjacent doses, and mock from individual doses. This is therefore due to the richness of information provided by the full set of NFκB dynamics alone. This also means that target genes that can only distinguish total activity, and not all the dynamic features, may in fact benefit from both NFκB and p38 combined activities to distinguish doses.

We added a paragraph to the discussion on a) how gene-specific regulatory mechanisms affect which information can be decoded and thus whether p38 dynamics may contribute information and b) the limitations of population-based measures such as information theory in describing signal encoding decoded by gene-specific mechanisms.

We have also clarified when stating this conclusion in different parts of the manuscript that p38 does not contribute to dose distinction with the 'full-set of information-rich NFκB dynamics', to emphasize that this applies to target genes that are capable of decoding the available dynamic information.

Reviewer #2:

This manuscript from Luecke et al. evaluates the contribution of p38 kinase signaling to the ability of macrophages to mount unique responses to different pathogenic stimuli. They establish a nicely engineered cell system with endogenously tagged NF-κB and virally inserted p38-KTR to generate reporter macrophages and pair this system with live cell microscopy and computational analysis. The authors compare hundreds of p38 and NF-κB single-cell activity time series for three stimuli associated with pathogens and the host cytokine TNF. They perform a range of quantitative analyses to address the question of how p38 and NF-κB activities distinguish the different stimuli. A major conclusion from the paper is that while p38 activity dynamics are on their own sufficient to distinguish between the different stimuli reasonably well, combining p38 and NF-κB activities doesn't add substantially to ligand distinction. In fact, somewhat surprisingly, the p38 or NF-κB component of the signal can be scrambled between cells without much affecting the ability to predict the stimulus. The authors also examine correlations between the p38 and NF-κB dynamics within single cells, finding low correlation in general, and they explain this using an ODE model with a distribution of initial conditions, which shows results consistent with the low degree of correlation. Finally, the authors examine the relationship of the signaling reporter results to downstream gene expression. Here, the authors hypothesize that genes that are responsive to both NF-κB and p38 will show higher variability from cell to cell than genes regulated by a single pathway, since the uncorrelated dynamics of the two pathways will both influence the expression levels of jointly regulated genes. Data from single-cell RNAseq, with genes grouped according to their known regulators, are used to confirm a statistically significant difference consistent with this idea.

We thank the reviewer for reading the paper carefully and the brief summary.

In general, this paper is a very well executed study of the correlation between two important signaling pathways involved in innate immune signaling and pathogen detection. While a substantial body of literature has investigated NF- κ B dynamics at the single-cell level, there is much less known about how multiple pathways interact, and this study goes a long way toward filling this gap. The analyses examining the information content of the two pathways are nicely presented and paint a clear picture of their relationship of stimuli and dynamics. This relationship is somewhat different than what might have been expected (e.g. the two pathways could make strongly cooperative independent contributions to distinguishing the stimuli), and the paper does a good job of portraying the more nuanced conclusions that the data lead to. The idea proposed in the last two figures, that "and" gate regulation results in greater variability in gene regulation, is an interesting one, but it is less well grounded in data than the rest of the paper. Overall, this is an important study that fits the journal well and will be of interest to readers both in innate immunity specifically and the broader signal transduction and systems biology fields.

We thank the reviewer for this appreciative assessment of our study, especially for noting our thorough computational analysis of the nuanced conclusions that are in contrast with the previously held hypothesis in the field. We hope that the reviewer will now find the gene regulation analysis much improved (outlined below).

Several issues should be addressed:

1. The models relating dynamic features (or time series) to stimuli in figures 2-4 consider the differences between the four stimuli, but don't include the mock treatment as a possible class (though there is some separate consideration of TNF vs. mock in Fig. 3H). From one perspective this is an understandable choice, but it leaves the reader wondering how the models would look if the mock treatment were included along with the four stimuli. Given that cells don't necessarily "know" that they are receiving a stimulus rather than no stimulus, I think this is an important question. Including the mock treatment is important for providing context to the models distinguishing the different stimuli. These alternate models could be included as supplementary figures.

Thank you for this suggestion! We agree that both classifications of the four immune stimuli with and without additionally including mock are relevant, as they address slightly different biological questions. In the information theoretic calculations on stimulus distinction we now always include mock with the other four or three stimulations (Figure 2C, 3I). We also provide machine learning classifications using dynamic features of the four stimuli + mock in Figure EV2 A, B, E-G. These show in p38 feature based classification, the F1 score of TNF was most affected by the inclusion of the mock and was most commonly misclassified mock, while the accuracy of identification of the PAMPs was less affected. In the NF κ B-based or p38&NF κ B based-classifications, inclusion of mock as a category had little effect on the TNF F1 score, supporting the previously made conclusions that NF κ B provides reliable distinction of TNF from mock, while p38 is less reliable in this regard.

For the distinction of the three PAMPs from each other, we believe that the biological question of whether p38 contributes to stimulus-specificity among these three is answered best when not including mock stimulation and thus do not provide additional classification models for this analysis.

2. The analysis of scRNA-seq gene categories in Fig. 7 feels incomplete. Why aren't p38-only genes included as a category? The details on which genes make up the NF- κ B only and NF- κ B+p38 classes are not provided, and

there is only a brief and very general note in the Methods on how they were chosen. Although the hypothesis is interesting, it is hard to put a lot of stock in the conclusions drawn from this analysis as it is currently presented.

We do not include a category of p38-only genes, because we are actually unaware of immune response genes that are exclusively regulated by p38.

All the details of which genes make up the NFkB-only and the NFkB+p38 categories and how they were assigned are given in Table EV5. As stated in the Materials&Methods, category assignment was performed based on thorough literature-based manual curation of previously published gene assignments that were obtained using knockout cell lines as well as quantitative modeling of gene expression data refined in multiple publications (Cheng *et al*, 2017; Tong *et al*, 2016; Sen *et al*, 2020; Wang *et al*, 2021; Sheu *et al*, 2023). We've now added this explanation to the Results section, in addition to providing it in Materials & Methods. We apologize that Table EV5 was not provided in the original submission.

We hope that the Reviewer will now find our newly expanded analysis more complete. It now additionally includes a) a mathematical model of AND gate gene control by NFkB (positively controlling mRNA transcription) and p38 (negatively controlling mRNA degradation), using experimentally determined signaling dynamics as input and predicting larger variability under NFkB&p38 control than under NFkB-only control (Figure 7A,B), b) an exploration of bimodality of scRNAseq-measured gene expression (and signaling codons) in addition to the previously provided analysis of Fano Factors, showing that non-bimodal p38 and NFkB activities can combine in AND gate gene regulation to result in an increased bimodality coefficient of gene expression (Figure 7F, EV5C,D,F) and c) an additional focus on gene-regulatory strategies via presence of ARE elements (required for TTP binding to mRNA, the mediator of p38's indirect effect on mRNA degradation), which reveal more pronounced differences in bimodality (Figure 7F).

3. The text discussing the correlations shown in Fig. 6A doesn't seem to match the figure well. The text states that all correlations were low, but the coloring of the squares in the figure suggests that some of them are in the 0.7-0.8 range. The ambiguity could be from my misinterpretation of the color scale, or in what is considered "low" correlation. It would be useful to include actual numbers in addition to the heatmap coloring, and/or to indicate in the text what the numerical threshold is for a "low" vs. "high" correlation.

The reviewer is correct that correlations of dynamic features determined early the pathway, such as the speed of the response and early 30-min integrals have correlation coefficients going up to 0.74, which we describe in the text as 'modest' or 'moderate'. However, after these very early times of the response the correlations drop dramatically, followed by weak negative correlations at later time points.

To avoid any ambiguity or misunderstanding, we now provide more correlation coefficients of specific dynamic features in the text. We have also added scatter plots of the activity integral correlations and for some of the correlations of the other dynamic features that also directly state the correlation coefficients (Figure 5B, EV3A). We've also updated the heatmaps such that correlations marked with a 'star' pass both a p-value threshold (<0.05) and a Spearman correlation coefficient threshold (>0.15), to provide a more clear idea of what might be considered correlations of relevance.

4. I think it is worth adding a section in the Discussion to make clear the general caveats and limitations of looking at the information content of signaling from the human perspective rather than the cell perspective.

For example, the activity perceived by reporters from the whole-cell KTR measurement may miss meaningful differences in sub-cellular location.

This is an important point that we realize we did not make clearly in the first submission. In brief, what we show is that, while combinatorial and dynamical coding are not mutually exclusive and in principle can occur together and synergistically, in the response to inflammatory stimuli they appear to be redundant. Thus, immune response genes gain stimulus-specificity by evolving regulatory mechanisms that can decode dynamical information present in NF κ B trajectories (or to a lesser degree p38 trajectories), or that can decode the combinatorics of NF κ B and p38 activation per se, but they gain little further stimulus-specificity by being able to decode both combinatorics and dynamics of these pathways. This is a new insight that we believe is of broad interest to the signaling and innate immunity research community. In addition, we describe the caveat that whole-cell KTR measurements may not report on the control sub-cellular localization needed for the diversity of functions.

Reviewer #3:

In this manuscript, the authors explore how the MAPK p38 pathway contributes to temporal encoding of the macrophage inflammatory response. The authors have previously explored temporal encoding by NF- κ B. To build on this, they create a p38-NF- κ B dual reporter primary murine cell line to measure and analyze signaling dynamics through these two nodes in single cells. They find that p38 dynamics alone can distinguish between TNF vs. PAMP stimulation in macrophages. However, when compared to NF- κ B, it did not contribute much additional information to distinguishing between different PAMPs or between doses of the same PAMP. They observed that NF- κ B and p38 activation was poorly correlated at the level of single cells, and computational modeling suggested that loss of correlation between the two pathways was due to a branch in the p38 pathway that contributed an early v. late component to its activation. Finally, they suggest that this might contribute to heterogeneity in target genes that require both NF- κ B and p38 for expression.

Overall, this is an interesting paper with a technically challenging set up. A primary macrophage live-cell dual signaling reporter is a considerable feat and the number of single cells analyzed across 4 conditions and 5 doses is impressive. This allowed the authors to make some very interesting insights into the differential dynamic encoding of p38 vs. NF- κ B. My main questions pertain to the conclusion regarding p38's role in distinguishing doses, both the technical robustness of this conclusion and the interpretation (see points below). Also, this manuscript should be of high interest both to systems biologists and to cell biologists interested in innate immune signaling, however as written it is not very accessible to the latter group.

We thank the reviewer for this appreciative assessment of our work, especially for noting the technical challenges we overcame, allowing us to gather extensive single cell data across stimuli and doses, leading to conclusions regarding the stimulus- and dose-specificity of p38 signaling and the role of combinatorial signaling between p38 and NF κ B. We have revised the analysis of dose distinction thoroughly, adding multiple additional analyses (Figure 4), have added flow cytometry analysis of p38 fractional activation (Figure 1), and have added a thorough discussion on the limitations of using these approaches for analysis of functional dose distinction at the level of individual genes (for details see below). We hope the reviewer will thus find the robustness of our conclusions much improved through implementation of many of their suggestions.

Major comments

1. I appreciate the extent to which the authors benchmarked the measurements of the p38 KTR reporter with Western blots. In figure 1E, it seems that the p38 dynamics of this reporter cells differ from those of the Western, especially at lower doses. A previous study comparing a FRET reporter versus a KTR reporter for ERK showed that the FRET reporter was sensitive at lower activity levels, and the KTR was more sensitive at higher activity levels (<https://doi.org/10.1016/j.cels.2017.10.019>). Is it possible that there is a loss of biological information in the KTR assay (especially at low doses), and that this might affect the conclusion that p38 does not contribute to discrimination of different PAMPs or doses? On a related note, did the authors compare fractional activation of p38 using their reporter to flow analysis of p-p38?

We thank the reviewer for this important note regarding the sensitivity of the KTR assay at low doses and for this interesting reference, which we now include in the manuscript. Given that these comparisons across measurement modalities are always affected by both the sensitivity and the dynamic range of the assays, it is indeed possible that the quantitative analysis are affected by imperfections of the assay. We have included this caveat when discussing the Results.

We've now also included flow cytometry measurements of p-p38 over 1 h of stimulation with the 4 stimuli at 3 doses and show the mean fluorescence intensity as well as fractional activation across doses (Figure 1F, G, EV1F, G). We generally find a good match in p38 activation between both measurements. The main noticeable discrepancy was a decreased sensitivity to low doses of TNF, compared to flow cytometry. The decreased sensitivity to low doses of TNF may be due to the brevity of TNF-induced p38 activity.

This may affect the interpretation of how different the p38 and NF κ B TNF dose responses are and the differences in Mutual Information in the dose responses between TNF and the other stimuli, which we now note in the Results description of Figure 4. Apart from these, the observed patterns and conclusions for TNF dose distinctions (all doses, adjacent doses, mock vs individual doses) and effect of combinatorial signaling on them are very similar between TNF and the other stimuli – We therefore consider our overall conclusions on dose specificity of p38 dynamics and combinatorial signaling with NF κ B robust. We'd also like to point out that the overall shape of the dose responses for p38 and conclusions of differential response thresholds between NF κ B and p38 across stimuli are in good agreement with what was previously observed for LipidA dose responses using flow cytometry (Gottschalk *et al*, 2016).

Many TNF-specific aspects of p38 signaling observed via KTR are also observed via flow cytometry at high TNF dose, such as the brief peak of activation and the low fractional activation. Thus, the decreased sensitivity of the KTR is less of a concern at the high TNF dose which is used for analysis of stimulus-specific signaling. Therefore, we consider the analysis of stimulus-specificity of p38 dynamics at high doses to not be significantly affected by sensitivity issues of the assay.

2. A main conclusion of the paper is that p38 does not contribute to dose distinction for PAMPs. The authors compare their results to Gottschalk *et al* (Cell Systems 2016). In my reading, the main conclusion from the Gottschalk paper is that the switch-like induction of the MAPKs following KLA stimulation enables macrophages to distinguish homeostatic v. pathogenic doses of PAMPs. Given this, one would not expect p38 to contribute to fine dose distinction for LPS, but rather to distinguish between 'harmless' vs. 'harmful' levels. The authors mention this in the Discussion on p. 13 ("This suggests that p38 mainly provides information about the absence or presence of a high dose stimulus.") but this seems like an important point that could be tested directly:

a. How would p38 v. NF-kB perform if tasked with distinguishing sub-threshold vs. high-threshold doses for LPS rather than all doses?

b. It's interesting that not all the PAMPs appear to follow this same switch-like induction for p38. How they compare in performing this simpler task?

We agree with the reviewer's reading of the Gottschalk et al., 2016 study with regards to one of p38's roles being distinguishing harmful from harmless PAMP concentrations, but also think that the differences in activation thresholds between NFkB and p38 still warrant the hypothesis of an improved distinguishability of doses around those thresholds (i.e. NFkB being activated for 2 adjacent intermediate doses, and p38 only being activated for the higher of the 2 doses might be hypothesized to improve distinguishability of the doses).

In any case, binary dose analyses are a great suggestion! We have significantly restructured the dose-specificity figure and now added three new types of machine learning classification of doses (for all 4 stimuli) to Figure 4 (dose specificity), including the suggested binary classification of below-and above-threshold doses:

1) A comparison of using all dynamic features for classification compared to use of a limited dynamic feature ('total activity') only (Figure 4D): Given the differential dose response thresholds between NFkB and p38, our intuitive hypothesis had been that combinatorial signaling should indeed increase the amount of information available about fine dose distinction. When using 'total activity' only, the intuitive prediction holds true, with combined activities performing slightly better than individual activities. However, as reported in the original submission, when all dynamic features are used for dose classification, we consistently see no improvement using the combination of p38 and NFkB dynamics for classification of 5 doses. This is likely due to the richness of information provided by the full set of NFkB dynamics alone. Based on this, we added a paragraph to the discussion on a) how gene-specific regulatory mechanisms affect which information can be decoded and thus whether p38 dynamics may functionally contribute information and b) the limitations of population-based measures such as information theory in describing signal encoding decoded by gene-specific mechanisms.

2) Binary classifications of adjacent doses (e.g. 0.1 ng/ml from 1 ng/ml) (across all doses and stimuli) (Figure 4F): Across stimuli, NFkB and NFkB+p38 dynamics performed provided similarly accurate classifications, while p38 dynamic features generally provided less accurate classification than either NFkB or NFkB+p38, especially at low and medium doses below or near its activation threshold, but interestingly approached the accuracy of NFkB-based classifications at medium-to-high doses (e.g. 10 vs. 100 ng/ml P3C4). This suggests that p38 can independently provide dose specific information for adjacent doses at higher doses, but that there is no evidence of additional information being provided by combinatorial signaling, even for binary distinction of intermediate doses around p38's activation threshold.

3) Binary classifications of mock (sub-threshold) from individual doses (the higher doses being above-threshold) (across all stimuli and doses) (Appendix Figure 3): This analysis shows that, across stimuli, above its activation threshold, p38 is independently capable of providing highly accurate identification of presence or absence of a stimulus, but again no role for combinatorial signaling is identified. For each stimulus whether this accuracy is achieved at a similar or a higher dose than NFkB's ability accurately identify is determined by their stimulus-specific differential dose responses.

Thus, overall, there is robust evidence that when considering all available dynamic features, p38 dynamics on their own have some dose specificity that is highly accurate for some dose distinctions, but inferior to NFκB's dose distinctions and provide no additional information when considered in combination with NFκB dynamics. We discuss how individual gene-specific mechanisms may not be able to interpret the full set of dynamics and thus may still increase their specificity of expression using combinatorial signaling.

3. The observation of heterogeneity (and thus weak correlations) in activation of p38 and NF-κB is interesting. The authors suggest this can be linked to heterogeneity in downstream targets that require both signals, especially cytokines, and show higher Fano factors for those. Bimodal distributions can lead to high Fano factors, and Fig. 7A looks like this is the case for Il1a and Il1b. Again, the authors mention this is in the Discussion ("We therefore suggest a new hypothesis, that AND gate control of gene expression may have evolved for those genes that must be expressed only in a minority of cells."). Combined with the point above that p38 shows thresholding behavior:

- a. Do the NF-κB+p38 gene targets show more bimodal behavior relative to NF-κB only targets? Other studies suggest this could be the case.
- b. Is there anything in the signaling dynamics of p38 (even at the high doses) that would contribute not only to heterogeneity, but to such bimodal behavior at the single-cell level?

The authors seem to suggest in the Discussion that such analysis would be beyond the scope of this paper, which is fair. However, connecting heterogeneity with bimodality (or fractional expression) is a point worth making in the main text if it's indeed there.

a) Thank you for this great suggestion! We've now added a quantification of bimodality coefficients to the scRNAseq analysis (Figure 7F, EV5C, D). Indeed, we find some evidence of increased bimodality coefficients for Il1a, Il1b, and the collection of NFκB&p38-dependent genes compared to NFκB-only genes. Interestingly, in the collection of genes, this difference is statistically significant for some stimuli at 1 h, while the Fano Factors showed statistically significant differences at 8 h. Especially pronounced differences in bimodality coefficients are seen between NFκB&p38 dependent genes with ARE elements (via which p38 exerts its indirect effect on mRNA stabilization) and without ARE elements; these differences are not present for NFκB-only genes. There is stimulus-specificity to the presence and extent of these differences. The Reviewer indicates there may be prior studies that may anticipate or support this conclusion. We are not sure which papers the reviewer is referring to and would be appreciative of pointers.

b) We've now added an analysis of the bimodality coefficients of the 30-min activity integrals over 4 h reveals that NFκB and p38 activities, which show similar, low bimodality coefficients below a threshold for what would be considered a bimodal distribution (Figure EV5F). This suggests that the AND gate control allows generation of bimodal gene expression from a combination of non-bimodal signals.

4. The manuscript would benefit from more background explanation. For example, it would be helpful to include the definition of a 'bit', what this measure means, and how to interpret its significance. It would also be helpful to include a brief explanation for why LSTM and decision tree classifiers were chosen. As written, the work is not very accessible to a non-expert.

We hope that the accessibility of the manuscript for interested readers with a less quantitative background is now improved. The text now includes expanded background explanations:

- Explanation of temporal and combinatorial signaling (paragraph 2 and 3 of intro), with many biology-focused specific examples (already present in initial submission, now slightly extended)
- Slightly expanded the explanation of the 'sequential AND gate' of NFκB and p38 in the introduction (paragraph 4)
- Slightly expanded the explanation of the purpose of information theoretic approaches (paragraph 4)
- Expanded the explanation of mutual information and 'bits' in the Results section description of Figure 2
- Added an explanation of machine learning classifications in general and of why the LSTM and the decision tree ensemble was chosen (Results section of Figure 2)
- We also already provided a visual explanation of the LSTM and the decision tree ensemble classifiers which we hope also adds to the accessibility of these analysis (Figure 2D, F, already in initial submission)
- Slightly expanded the explanation of a confusion matrix (Results section of Figure 2)
- Expanded the explanation of the relevance of the comparison of the two distinct classifiers (Results section of Figure 2)
- We generally start many of the figures with the heatmaps of p38 and NFκB trajectories and with thorough description of the dynamics observable in those, which already preview many of the conclusions then achieved via more quantitative analysis (already in initial submission). We believe this also contributes to the accessibility of the manuscript.

Minor comments

1. Fig. 1E: it's not clear what is meant by the labels "Dose 2/5, Dose 3/5 etc."

We have now spelled out the individual doses in detail in Figure 1E.

2. In figure 3D-H (and in other similar figures) all the panels share the same color scheme so it might be easier to a legend rather than using the +, - system at the bottom of each graph.

Thank you for this advice. We have added a color legend, in some cases in addition to the +/- system, to Figures 3,4, EV2, Appendix 3.

Also, it is also not obvious what is meant by "shuffling" of the p38 or NFκB.

We've now attempted to explain this more thoroughly in each figure legend. In the case of the machine learning classifiers, by 'shuffling' we are referring to matching the 'correct' NFκB or p38 activity with the 'incorrect' p38 or NFκB, respectively, activities from cells randomly selected from all classes.

3. p5: Include some brief background on the PAMPs used (full name, receptor, pathway)

We have now included more background on the PAMPs on p.5.

4. p6: LSTM is not defined.

We apologize for this oversight. The abbreviation is now defined on p.6.

References

Cheng CS, Behar MS, Suryawanshi GW, Feldman KE, Spreafico R & Hoffmann A (2017) Iterative Modeling Reveals Evidence of Sequential Transcriptional Control Mechanisms. *Cell systems* 4: 330-343.e5

- Gottschalk RA, Martins AJ, Angermann BR, Dutta B, Ng CE, Uderhardt S, Tsang JS, Fraser ID, Meier-Schellersheim M & Germain RN (2016) Distinct NF-kappaB and MAPK Activation Thresholds Uncouple Steady-State Microbe Sensing from Anti-pathogen Inflammatory Responses. *Cell systems* 2: 378–90
- Sen S, Cheng Z, Sheu KM, Chen YH & Hoffmann A (2020) Gene Regulatory Strategies that Decode the Duration of NFkB Dynamics Contribute to LPS- versus TNF-Specific Gene Expression. *Cell Syst* 10: 169-182.e5
- Sheu KM, Guru AA & Hoffmann A (2023) Quantifying stimulus-response specificity to probe the functional state of macrophages. *Cell Syst* 14: 180-195.e5
- Tong AJ, Liu X, Thomas BJ, Lissner MM, Baker MR, Senagolage MD, Allred AL, Barish GD & Smale ST (2016) A Stringent Systems Approach Uncovers Gene-Specific Mechanisms Regulating Inflammation. *Cell* 165: 165–179
- Wang N, Lefaudeux D, Mazumder A, Li JJ & Hoffmann A (2021) Identifying the combinatorial control of signal-dependent transcription factors. *PLoS Comput Biol* 17: e1009095

21st May 2024

Manuscript Number: MSB-2023-12076R

Title: Dynamical and combinatorial coding by MAPK p38 and NFκB in the inflammatory response of macrophages

Dear Alex,

Thank you for sending us your revised manuscript. We have now heard back from the three reviewers who were asked to evaluate your revised study. As you will see below, the reviewers are satisfied with the performed revisions and support publication. They only raise a couple of minor points which we would ask you to address in a minor revision. We would also ask you to address some editorial issues listed below.

- Our Data Editors noted that the following needs to be corrected/added in the Figure Legends:

-- Please indicate the statistical test used for data analysis in the legends of figures 7e-f; EV 5c-e.

-- Information related to n is missing in the legends of figures 2j; 3m; 7c.

- The funding information provided in the manuscript text should match the information entered in the online submission system. The information on "the UCLA Medical Scientist Training Program (T32-GM008042) and Systems and Integrative Biology Training Grant (T32-GM008185); UCLA Gastroenterology Training Grant (T32- DK007180); Brazilian Council for Scientific and Technological Development (200176/2022-6)" is missing from the submission system.

- Please include a callout for Fig. 7F.

- Tables EV3A and EV3B should be combined into a single EV Dataset, with separate sheets for each table, and another for the description of the Dataset. Please update the callouts and EV table labels should be updated accordingly as the numbering will change for the rest of the EV Tables.

- Not all sections of the Author Checklist have been completed: in "Experimental study design and statistics" section, a response needs to be selected for "Include a statement about sample size estimate even if no statistical methods were used."

Please resubmit your revised manuscript online, with a covering letter listing amendments and responses to each point raised by the referees. Please resubmit the paper ****within two weeks**** and ideally as soon as possible. If we do not receive the revised manuscript within this time period, the file might be closed and any subsequent resubmission would be treated as a new manuscript. Please use the Manuscript Number (above) in all correspondence.

Click on the link below to submit your revised paper.

Best wishes,

Maria

Maria Polychronidou, PhD
Senior Editor
Molecular Systems Biology

If you do choose to resubmit, please click on the link below to submit the revision online before 20th Jun 2024.

IMPORTANT:

Please note that corresponding authors are required to supply an ORCID ID for their name upon submission of a revised manuscript (EMBO Press signed a joint statement to encourage ORCID adoption).

(<https://www.embopress.org/page/journal/17444292/authorguide#editorialprocess>)
Currently, our records indicate that the ORCID for your account is 0000-0002-5607-3845.

Please click the link below to modify this ORCID:
Link Not Available

*** PLEASE NOTE *** As part of the EMBO Press transparent editorial process initiative (see our Editorial at <https://dx.doi.org/10.1038/msb.2010.72>), Molecular Systems Biology will publish online a Review Process File to accompany accepted manuscripts. When preparing your letter of response, please be aware that in the event of acceptance, your cover letter/point-by-point document will be included as part of this File, which will be available to the scientific community. More information about this initiative is available in our Instructions to Authors. If you have any questions about this initiative, please contact the editorial office (msb@embo.org).

Reviewer #1:

I am very satisfied with how the authors have addressed my comments in the initial report. The new data analysis and model simulations greatly increase the novelty and impact of the work. The new results are quite interesting. This paper will be appreciated by the molecular systems biology audience.

Reviewer #2:

This revised version quite thoroughly addresses the points raised by the reviewers. Overall, this study provides a deep exploration of the interaction of the NF- κ B and p38 pathways in controlling inflammation-related genes, and it delivers interesting and unexpected insights into the ways that the parallel pathways are related and interpreted. This paper will likely become a model for similar analyses of other pathways that act in parallel downstream of a common receptor, a common motif in signaling networks.

Some small details remain to be addressed. There were a few typos found, such as "Furher" on p. 17 of the PDF or "first speak" on p. 10. Also this sentence on p. 3 deserves some rewording for clarity: "Quantifying coding capacities in NF κ B-p38 combinatorial coding the context of substantial cell-cell heterogeneity within innate immune responses must be examined at the single cell level."

Another point is that in the analysis of dose discrimination described in Fig. 4, it is not acknowledged that that p38 KTR fails to pick up activity that can be observed by WB or flow cytometry for 0.1 ng/ml TNF.

Last, Figures 1E,F and EV3 note WB and flow data that come from single experiments. Could the reliability of these data be strengthened, perhaps by noting typical ranges for replicates observed within the authors' labs for these techniques?

Reviewer #3:

In this revision, the authors have more clearly presented their key point: that p38 dynamics distinguish types of stimuli and doses of stimuli, but surprisingly do not contribute much beyond information rich NF- κ B dynamics when both pathways are considered. They have also increased the support for their second important key point: that combinatorial signaling increases heterogeneity, and genes dependent on both pathways show increased variability and bimodality relative to genes dependent only on NF- κ B. The analysis comparing genes with and without ARE elements (as a proxy for p38 regulation) was a clever addition that further supports their conclusions. These findings provide important insights into how combined signaling inputs contribute to stimulus discrimination versus regulation of response.

Very minor comment:

Fig. 7D: it would be easier to compare differences across genes by adding a reference line at 1 (Fano) and 0.5 (Bimodality).

Editorial Requests:

- Our Data Editors noted that the following needs to be corrected/added in the Figure Legends:

-- Please indicate the statistical test used for data analysis in the legends of figures 7e-f; EV 5c-e.

In each of these figure legends, we state: "Statistical significance was determined using a permutation test for difference in means using 10000 permutations." This is a non-ambiguous description of the statistical test used, so we are not sure what other information is expected.

-- Information related to n is missing in the legends of figures 2j; 3m; 7c.

Fig. 2J: We are confused as to why information related to n is thought to be missing from this legend. The legend states "Data from two pooled biological replicates are depicted. Total # of cells: 923, 1171, 970, and 1055 cells for P3C4, CpG, LPS, and TNF."

Fig. 3M: We now reiterate in 3M legend the total number of cells used for this analysis (same throughout the Figure, mentioned in legend 3A): "Total # of cells: 923, 1171, and 970 cells for P3C4, CpG, and LPS."

Fig. 7C: Here we are again confused as to why information related to n is thought to be missing. The figure legend very thoroughly states: "Number of cells analyzed at each time point: Unstimulated: 1415 cells; P3C4: 835, 1840, 1251, 1153; CpG: 992, 1229, 2235, 1236; LPS: 941, 1045, 663, 1338; TNF: 980, 776, 994, 2412 cells." We have now added the word 'cells' for each stimulation, to improve clarity.

- The funding information provided in the manuscript text should match the information entered in the online submission system. The information on "the UCLA Medical Scientist Training Program (T32-GM008042) and Systems and Integrative Biology Training Grant (T32-GM008185); UCLA Gastroenterology Training Grant (T32- DK007180); Brazilian Council for Scientific and Technological Development (200176/2022-6)" is missing from the submission system.

We have corrected this during the re-submission in the system.

- Please include a callout for Fig. 7F.

Thank you. We have corrected the two instances where Fig. 7E was mistakenly called out instead of Fig. 7F.

- Tables EV3A and EV3B should be combined into a single EV Dataset, with separate sheets for each table, and another for the description of the Dataset. Please update the callouts and EV table labels should be updated accordingly as the numbering will change for the rest of the EV Tables.

We have implemented these changes.

- Not all sections of the Author Checklist have been completed: in "Experimental study design and statistics" section, a response needs to be selected for "Include a statement about sample size estimate even if no statistical methods were used."

We apologize if this was overlooked in the submitted version. This is what that part of the table is intended to look like:

Experimental study design and statistics	Information included in the manuscript?	In which section is the information available? (Reagents and Tools Table, Materials and Methods, Figures, Data Availability Section)
Include a statement about sample size estimate even if no statistical methods were used.	Yes	Materials and Methods Section

The statement included in the Materials & Methods section reads: "The number of cells imaged was determined by the cell density and the total number of positions that could be imaged within the 5 min framerate."

Reviewer #1:

I am very satisfied with how the authors have addressed my comments in the initial report. The new data analysis and model simulations greatly increase the novelty and impact of the work. The new results are quite interesting. This paper will be appreciated by the molecular systems biology audience.

We thank the reviewer for their appreciative assessment of our revisions.

Reviewer #2:

This revised version quite thoroughly addresses the points raised by the reviewers. Overall, this study provides a deep exploration of the interaction of the NF-kB and p38 pathways in controlling inflammation-related genes, and it delivers interesting and unexpected insights into the ways that the parallel pathways are related and interpreted. This paper will likely become a model for similar analyses of other pathways that act in parallel downstream of a common receptor, a common motif in signaling networks.

Some small details remain to be addressed. There were a **few typos found**, such as "Furher" on p. 17 of the PDF or "first speak" on p. 10. Also this sentence on p. 3 deserves some rewording for clarity: "Quantifying coding capacities in NFkB-p38 combinatorial coding the context of substantial cell-cell heterogeneity within innate immune responses must be examined at the single cell level."

We thank the reviewer for catching these typos. We have now done our best to find any other typos.

We have reworded that sentence to read "Quantifying coding capacities in NFkB-p38 combinatorial signaling requires examination at the single cell level given the substantial cell-cell-heterogeneity within innate immune responses".

Another point is that in the analysis of dose discrimination described in Fig. 4, it is not acknowledged that that p38 KTR fails to pick up activity that can be observed by WB or flow cytometry for 0.1 ng/ml TNF.

It was our intention to acknowledge this: In our previous submission, when describing Figure 4 results, we had written: “The TNF concentration at half maximum activity for p38 was 12.9x that of NFκB (*potentially affected by the slightly decreased sensitivity of the KTR for low TNF doses*); for LPS it was 8x and for CpG 2.6x. “

To make this more explicit, we have rephrased: “The TNF concentration at half maximum activity for p38 was 12.9x that of NFκB; for LPS it was 8x and for CpG 2.6x. (TNF’s dose response may be affected by the slightly decreased sensitivity of the KTR for low TNF doses (0.1 ng/ml) compared to flow cytometry or Western Blot based measurements (see Figure 1E-G)).”

Last, Figures 1E,F and EV3 note WB and flow data that come from single experiments. Could the reliability of these data be strengthened, perhaps by noting typical ranges for replicates observed within the authors' labs for these techniques?

Since the main goal of these experiments was to evaluate the sensitivity of the KTR to dynamics and dose response of p38 activity, we achieve confidence in the reliability of the data by comparing three different methods of measuring p38 activity (KTR microscopy, Western Blotting, and flow cytometry) in detailed time course experiments and at three doses performed with independent iMPDM differentiations on different days. We note in the manuscript: “In response to low dose P3C4 and TNF, the KTR measurements showed more and less signal, respectively, than the flow cytometry, potentially within range of experimental variability”, based on our lab’s experience of variability of speed and amplitude in cellular responses to stimulations.

As an example, please see Figure RL1 for a flow cytometry replicate experiment to evaluate variability in p-p38 activity measured by flow cytometry upon TNF and CpG stimulation:

Figure RL1: Variability of p38 activity measurements as p-p38 levels by flow cytometry in TNF and CpG stimulated iMPDMs.

Reviewer #3:

In this revision, the authors have more clearly presented their key point: that p38 dynamics distinguish types of stimuli and doses of stimuli, but surprisingly do not contribute much beyond information rich NF-κB dynamics when both pathways are considered. They have also increased the support for their second important key point: that combinatorial signaling increases heterogeneity, and genes dependent on both pathways show increased variability and

bimodality relative to genes dependent only on NF- κ B. The analysis comparing genes with and without ARE elements (as a proxy for p38 regulation) was a clever addition that further supports their conclusions. These findings provide important insights into how combined signaling inputs contribute to stimulus discrimination versus regulation of response.

We are happy to read that the reviewer finds the clarity of our conclusions improved.

Very minor comment:

Fig. 7D: it would be easier to compare differences across genes by adding a reference line at 1 (Fano) and 0.5 (Bimodality).

That is a great suggestion. Figure 7 has been edited accordingly.

28th May 2024

Manuscript number: MSB-2023-12076RR

Title: Dynamical and combinatorial coding by MAPK p38 and NFκB in the inflammatory response of macrophages

Dear Alex,

Thank you again for sending us your revised manuscript. We are now satisfied with the modifications made and I am pleased to inform you that your paper has been accepted for publication.

Best wishes,

Maria

Maria Polychronidou, PhD
Senior Editor
Molecular Systems Biology
